# Visual correspondence-based explanations improve AI robustness and human-AI team accuracy

**Giang Nguyen** *
nguyengiangbkhn@gmail.com

**Mohammad Reza Taesiri** *
mtaesiri@gmail.com

**Anh Nguyen**
anh.ng8@gmail.com

Auburn University

## Abstract

Explaining artificial intelligence (AI) predictions is increasingly important and even imperative in many high-stakes applications where humans are the ultimate decision makers. In this work, we propose two novel architectures of self-interpretable image classifiers that first explain, and then predict (as opposed to post-hoc explanations) by harnessing the visual correspondences between a query image and exemplars. Our models consistently improve (+1 to +4 points) on out-of-distribution (OOD) datasets while performing marginally worse (-1 to -2 points) on in-distribution tests than ResNet-50 and a $k$-nearest neighbor classifier (kNN). Via a large-scale, human study on ImageNet and CUB, our correspondence-based explanations are found to be more useful to users than kNN explanations. Our explanations help users more accurately reject AI's wrong decisions than all other tested methods. Interestingly, for the first time, we show that it is possible to achieve complementary human-AI team accuracy (i.e., that is higher than either AI-alone or human-alone), in ImageNet and CUB image classification tasks.

## 1 Introduction

Comparing the input image with training-set exemplars is the backbone for many applications, such as face identification [29], bird identification [17, 79], and image search [79]. This non-parametric approach may improve classification accuracy on out-of-distribution (OOD) data [29, 75, 79, 57] and enables a class of prototype-based explanations [17, 52, 53, 66, 41] that provide insights into the decision making of Artificial Intelligence (AI) systems. Interestingly, prototype-based explanations are more effective in improving human classification accuracy [55, 24, 42] than attribution maps—a common eXplainable AI (XAI) technique in computer vision. Yet, it remains an open question how to make prototype-based XAI classifiers (1) accurate on in-distribution and OOD data and (2) improve human decisions. For example, in face identification, AIs can be confused by partially occluded, never-seen faces and are unable to explain their decisions to users, causing numerous people falsely arrested [5, 7, 3, 6] or wrongly denied unemployment benefits [1] by the law enforcement.

To address the above questions, we propose two *interpretable* [61], (i.e., first-explain-then-decide) image classifiers that perform three common steps: (1) rank the training-set images based on their distances to the input using *image-level* features; (2) re-rank the top-50 shortlisted candidates by their

---

*Equal contribution. Listing order is random. GN led the development of EMD-Corr and human studies on Gorilla. MRT led the development of CHM-Corr, pilot studies on HuggingFace, and the analysis of human-study data from Gorilla. AN advised the project. MRT's work was done before he joined University of Alberta.

36th Conference on Neural Information Processing Systems (NeurIPS 2022).

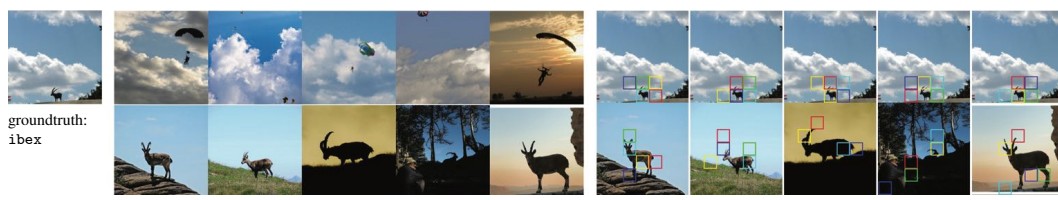

(a) Explanations for kNN's parachute decision (top) and CHM-NN (bottom)  (b) Explanations for CHM-Corr's ibex decision

Figure 1: The ibex image is misclassified into parachute due to its similarity (clouds in blue sky) to parachute scenes (a). In contrast, CHM-Corr correctly labels the input as it matches ibex images mostly using the animal's features, discarding the background information (b).

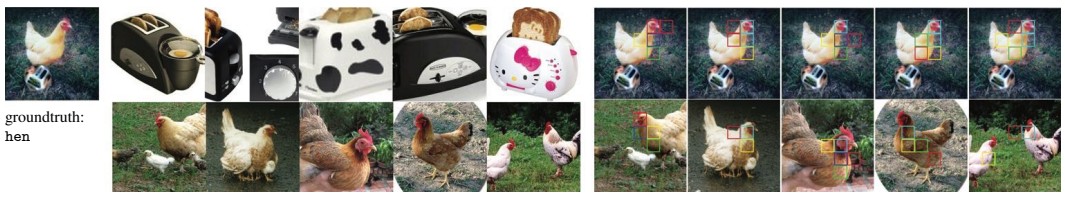

(a) Explanations for kNN's toaster decision (top) and EMD-NN (bottom)  (b) Explanations for EMD-Corr's hen decision

Figure 2: Operating at the image-level visual similarity, kNN incorrectly labels the input toaster due to the adversarial toaster patch (a). EMD-Corr instead ignores the adversarial patch and only uses the head and neck patches of the hen to make decisions (b).

*patch-wise* correspondences w.r.t. the input [51, 29]; and then (3) take the dominant class among the top-20 candidates as the predicted label. That is, our classifiers base their decisions on a set of support image-patch pairs, which also serve as *explanations* to users (Figs. 1b and 2b). Our main findings include: [2]

- On ImageNet, a simple $k$-nearest-neighbor classifier (kNN) based on ResNet-50 features slightly but consistently outperforms ResNet-50 on many OOD datasets (Sec. 3.1). This is further improved after a re-ranking step based on patch-wise similarity (Sec. 3.2).

- Via a large-scale human study, we find visual correspondence-based explanations to improve AI-assisted, human-alone accuracy and human-AI team accuracy on ImageNet and CUB over the baseline kNN explanations (Sec. 3.4).

- Having interpretable AIs label images that they are confident and humans label the rest yields better accuracy than letting AIs or humans alone label all images (Sec. 3.5 and Appendix M).

To the best of our knowledge, our work is the first to demonstrate the utility of correspondence-based explanations to users on ImageNet [63] and CUB [71] classification tasks.

## 2 Methods

### 2.1 Datasets

We test our ImageNet classifiers on the original 50K-image ILSVRC 2012 ImageNet validation set (i.e., in-distribution data) and four common OOD benchmarks below.

**ImageNet-R** [35] contains 30K images in 200 ImageNet categories, mostly artworks – ranging from cartoons to video-game renditions.

**ImageNet-Sketch** [72] consists of 50,889 black-and-white sketches of all 1,000 ImageNet classes.

**DAmageNet** [18] consists of 50K ImageNet validation-set images that contain universal, adversarial perturbations for fooling classifiers.

**Adversarial Patch** [15] are 50K ImageNet validation-set images that are modified to contain an adversarial patch that aims to cause ResNet-50 [31] into labeling every image toaster (see Fig. 2).

---

[2]Code and models are available at https://github.com/anguyen8/visual-correspondence-XAI.

Using the implementation by [2], we generate this dataset, which causes ResNet-50 accuracy to drop from 76.13% to 55.04% (Table 1). See Appendix A.5 for how to download and generate this dataset.

**CUB-200-2011** [71] (hereafter, CUB) is a fine-grained, bird-image classification task chosen to complement ImageNet. CUB contains 11,788 images (5,994/5,794 for train/test) of 200 bird species.

## 2.2 Classifiers

We harness the same ResNet-50 layer4 backbone [8] as the main feature extractor for all four main classifiers, including our two interpretable models. Therefore, to test the effectiveness of our models, we compare them with (1) a vanilla ResNet-50 classifier; and (2) a kNN classifier that uses the same pretrained layer4 features. We report the top-1 accuracy of all classifiers in Table 1.

**ResNet-50** For experiments on ImageNet and its four OOD benchmarks, we use the ImageNet-trained ResNet-50 from TorchVision [8] (top-1 accuracy: 76.13%).

For CUB, we take the ResNet-50 pretrained on iNaturalist [70] from [53] (hereafter, iNaturalist ResNet) and retrain only the last 200-output classification layer (right after avgpool) to create a competitive, baseline ResNet-50 classifier for CUB (top-1 accuracy: 85.83%). See Appendix A.1 for finetuning details.

**kNN** We implement a vanilla kNN classifier that operates at the avgpool of the last convolutional layer of ResNet-50. That is, given a query image $Q$, we sort all training-set images $\{G_i\}$ based on their distance $D(Q, G_i)$, which is the cosine distance between the two corresponding image features $f(Q)$ and $f(G_i) \in \mathbb{R}^{2048}$ where $f(.)$ outputs the avgpool feature of layer4 (see code) of ResNet-50.

The predicted label of $Q$ is the dominant class among the top-$k$ nearest neighbors. We choose $k = 20$ as it performs the best among the tested values of $k \in \{10, 20, 50, 100\}$.

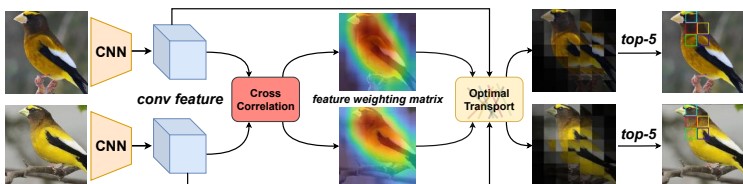

(a) EMD-Corr: First compute patch-wise similarity, and then find correspondences via solving EMD [29, 79].

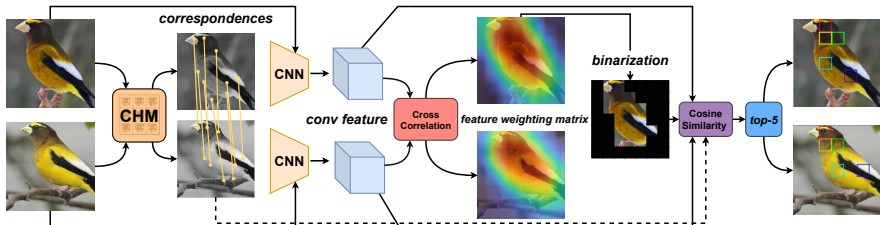

(b) CHM-Corr: First find correspondences via CHM [51], and then compute patch-wise similarity.

Figure 3: EMD-Corr and CHM-Corr both re-rank kNN's top-50 candidates using the patch-wise similarity between the query and each candidate over the top-5 pairs of patches that are the most important and the most similar (i.e. highest EMD flows in EMD-Corr and highest cosine similarity in CHM-Corr).

**EMD-Corr** As kNN compares images using only image-level features, it lacks the capability of paying attention to fine details in images. Therefore, we propose EMD-Corr, a visual correspondence-based classifier that (1) re-ranks the top-$N$ (here, $N = 50$) candidates of kNN using their Earth Mover's Distance (EMD) with the query in a patch embedding space (see Fig. 3a); and (2), similarly to kNN, takes the dominant class among the re-ranked top-20 as the predicted label.

That is, our PyTorch implementation is the same as that in [79, 29] except for three key differences. First, using layer4 features ($7 \times 7 \times 2048$), we divide an image into 49 patches, whose embeddings are $\in \mathbb{R}^{2048}$. Second, for interpretability, in re-ranking, we only use patch-wise EMD instead of a

linear combination of image-level cosine distance and patch-level EMD as in [79, 29], which makes it more opaque how a specific image patch contributes to re-ranking. Third, while the original deep patch-wise EMD [79, 29] between two images (see Eq. 3) is defined as the sum over the *weighted* cosine distances of all $49 \times 49$ patch pairs, we only use $L = 5$ pairs as explanations and therefore, only sum over the corresponding 5 flow weights returned by Sinkhorn optimization [21].

We find $N = 50$ to perform the best among $N \in \{50, 100, 200\}$. We choose $L = 5$, which is also the most common in nearest-neighbor visualizations [47, 45]. In preliminary experiments, we find $L = \{9, 16, 25\}$ to yield so dense correspondence visualizations that hurt user interpretation and $L = 3$ to under-inform users. See Appendix A.3 for more description of EMD-Corr.

**CHM-Corr** EMD-Corr first measures the pair-wise cosine distances for all $49 \times 49$ pairs of patches, and then computes the EMD flow weights for these pairs (Fig. 3a). To leverage the recent impressive end-to-end correspondence methods [51, 60, 48], we also propose CHM-Corr (Fig. 3b), a visual correspondence-based classifier that operates in the opposite manner to EMD-Corr. That is, first, we divide the query image $Q$ into $7 \times 7$ non-overlapping patches (i.e. as in EMD-Corr) and find one corresponding patch in $G_i$ for each of the 49 patches of $Q$ using a state-of-the-art correspondence method (here, CHM [51]). Second, for the query $Q$, we generate a cross-correlation (CC) map (Fig. 3) [29], i.e. a heatmap of cosine similarity scores between the layer4 embeddings of the patches of the query $Q$ and the image-embedding (avgpool after layer4) of each training-set image $G_i$. Third, we binarize the heatmap (using the optimal threshold $T = 0.55$ found on a held-out training subset) to identify a set of the most important patches in $Q$ and compute the cosine similarity between each such patch and the corresponding patch in $G_i$ (i.e., following the CHM correspondence mappings). Finally, the similarity score $D(Q, G_i)$ in CHM-Corr is the sum over the $L = 5$ patch pairs of the highest cosine similarity across $Q$ and $G_i$.

After testing NC-Net [60], ANC-Net [48], and CHM [51] in our classifier, we choose CHM as it has the fastest runtime and the best accuracy. Unlike ResNet-50 [31], which operates at the $224 \times 224$ resolution, CHM uses a ResNet-101 backbone that expects a pair of $240 \times 240$ images. Therefore, in pre-processing, we resize and center-crop each original ImageNet sample differently according to the input sizes of ResNet-50 and CHM. See Appendix A.4 for more description of CHM-Corr.

**CHM-Corr+ classifier based on five groundtruth keypoints of birds** In EMD-Corr and CHM-Corr, we use CC to infer the importance weights of patches. To understand the effectiveness of CC in weighting patches, on CUB, we compare our EMD-Corr and CHM-Corr to CHM-Corr+, a CHM-Corr variant where we use a set of five human-defined important patches instead of those inferred by CC. That is, for each CUB image, instead of taking the five CC-derived important patches (Fig. 3b), we use at most five patches that correspond to a set of five pre-defined keypoints (beak, neck, right wing, right feet, tail), each representing a common body part according to bird identification guides [25] for ornithologists. From the five patches in the query image, we then use CHM to find five corresponding patches in a training-set image, and take the sum of five cosine similarities as the total patch-wise similarity between two images in re-ranking.

A query image may have $< 5$ important patches if some keypoint is occluded. That is, evaluating CHM-Corr+ alone provides an estimate of how hard bird identification on CUB is if the model harnesses five well-known bird features.

## 2.3 User-study design

The interpretable classifiers (Sec. 2.2) are not only capable of classifying images but also *generating explanations*, which may inform and improve users' decision-making [55]. Here, we design a large-scale study to understand the effectiveness of explanations in **two human-AI interaction models** in classification (see Fig. 4): **Model 1:** Users make all the decisions after observing the input, AI decisions, and explanations (Fig. 4a). **Model 2:** AIs make decisions on only inputs that they are the most confident, leaving the rest for users to label. That is, model 2 (Fig. 4b–c) is a practical scenario where we offload most inputs to AIs while users only handle the harder cases.

Like [55], we show each user: (1) a query image; (2) AI top-1 label and confidence score; and (3) an explanation (here, available in kNN, EMD-Corr, CHM-Corr, and CHM-Corr+, but not in ResNet-50). We ask users to decide Y/N whether the top-1 label is correct (example screen in Fig. A7).

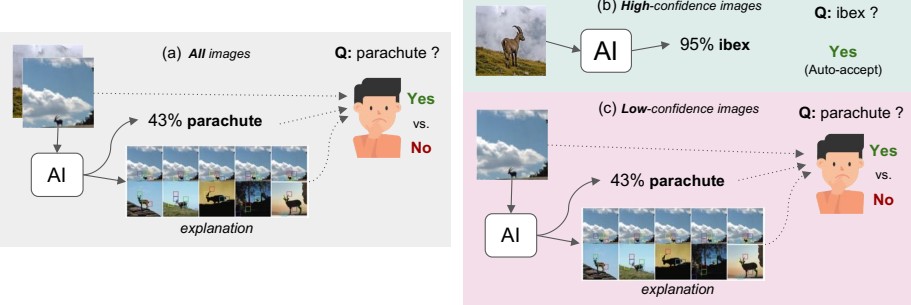

Model 1: Human-only           Model 2: Human-AI team decision makers

Figure 4: Two human-AI interaction models. In model 1 (a) , for all images, users decide (Yes/No) whether the AI's predicted label is correct given the input image, AI top-1 label and confidence score, and explanations. In model 2, AI decisions are *automatically* accepted if AI is highly confident (b) . Otherwise, humans will make decisions (c) in the same fashion as the model 1.

### 2.3.1 Explanation methods

We test the explanations of four main classifiers: ResNet-50, kNN, EMD-Corr, and CHM-Corr. Additionally, we test two *ablated* versions (i.e., EMD-NN and CHM-NN) of the explanations of EMD-Corr and CHM-Corr. In total, we test 6 explanation methods (see examples in Appendix D).

**ResNet-50** is a representative black-box classifier, which only outputs a top-1 label and a confidence score (i.e., *no explanations*).

**kNN explanations** From the top-20 nearest neighbors (as $k = 20$ in our kNN), we show the first five images that are from the predicted class (example in Fig. 1a). In some cases where the predicted class has only $M < 5$ exemplars in the top-20, we still show only those $M$ images (see Fig. A46). That is, we only show at most five neighbors following prior works [55, 67, 49, 65] that reported utility of such few-image explanations. We find explanations consisting of $\geq 10$ images such as those of ProtoPNet [17] are hard to interpret for users [42]. Note that our kNN explanations consist of five support images for each decision of kNN (described in Sec. 2.2) as opposed to the *post-hoc* nearest examples in [55], which do *not* reflect a classifier's decisions.

**EMD-Corr and CHM-Corr explanations** As EMD-Corr and CHM-Corr re-rank the top-50 candidates (shortlisted by kNN) and take the dominant class among the resultant top-20 as the predicted label, we show the five nearest neighbors from the predicted class as in kNN explanations. Instead of showing only five post-reranking neighbors, we also annotate, in each image, *all* five patches (example in Fig. 1b and Fig. 2b) that contribute to the patch-wise re-ranking (Sec. 2.2).

**EMD-NN and CHM-NN** To understand the effects of showing correspondences in the explanations to EMD-Corr and CHM-Corr users, we also test an ablated version where we show exactly the same explanations but without the patch annotations (bottom panels in Fig. 1a and Fig. 2a).

**Confidence scores** While ResNet-50's confidence score is the top-1 output softmax probability, the confidence of kNN, EMD-Corr, and CHM-Corr is the count of the predicted-class examples among the top $k = 20$. In human studies, we display this confidence in percentage (e.g. 10% instead of 2/20; Fig. A46) to be consistent with the confidence score of ResNet-50.

### 2.3.2 ImageNet and CUB datasets

For XAI evaluation, we run two human studies, one on ImageNet and one on CUB. For ImageNet, we use ImageNet-ReaL [14] labels in attempt to minimize the confounders of the human evaluation as ImageNet labels are sometimes misleading and inaccurate to users [55].

**Nearest-neighbor images** To generate the nearest-neighbor explanations for kNN, EMD-Corr, and CHM-Corr, we search for neighbors in the entire training set of ImageNet or CUB (no filtering).

**Query images** In attempt to ensure the quality of the query images that we ask users to label, from 50K-image ImageNet validation set, we discard images that: (a) do not have an ImageNet-ReaL [14]

label; (b) are grayscale or low-resolution (i.e., either width or height $< 224$ px) as in [55]; (c) have duplicates in the ImageNet training set (see Appendix L), resulting in 44,424 images available for sampling for the study. In CUB, we sample from the entire 5,794-image test set and apply no filters.

### 2.3.3 Training, Validation, and Test phases

From the set of query images (Sec. 2.3.2), we sample images for three phases in a user study: Training, validation, and test. Following [55], we first introduce participants to the task and provide them 5 training examples. Then, each user is given a validation job (10 trials for ImageNet and 5 for CUB), where they must score 100% in order to be invited to our 30-trial test phase. Otherwise, they will be rejected and unpaid. Among the 10 validation trials for ImageNet, 5 are correctly-labeled and 5 are misclassified by AIs. This ratio is 3/2 for CUB validation (examples in Appendix E.2).

Right before each trial, we describe the AI's top-1 label to users by showing them 3 training-set images and a 1-sentence WordNet description for each ImageNet class. For CUB classes, we show 6 representative images (instead of 3) for users to better recognize the characteristics of each bird (see Fig. A6).

**Sampling** For every classifier, we randomly sample 300 correctly- and 300 incorrectly-predicted images together with their corresponding explanations for the test trials. Over all 6 explanation methods, we have 2 datasets $\times$ 600 images $\times$ 6 methods = 7,200 test images in total.

### 2.3.4 Participants

We host human studies on Gorilla [11] and recruit lay participants who are native English speakers worldwide via Prolific [56] at a pay rate of USD 13.5 / hr. We have 360 and 355 users who successfully passed our validation test for ImageNet and CUB datasets, respectively. We remove low-quality, bottom-outlier submissions, i.e., who score $\leq 0.55$ (near-random accuracy), resulting in 354 and 355 submissions for ImageNet and CUB, respectively. In each dataset, every explanation method is tested on $\sim$60 users and each pair of (query, explanation) is seen by almost 3 users (details in Table 2).

## 3 Experimental Results

### 3.1 ImageNet kNN classifiers improve upon ResNet-50 on out-of-distribution datasets

Despite impressive test-set performance, ImageNet-trained convolutional neural networks (CNNs) may fail to generalize to natural OOD data [72] or inputs specifically crafted to fool them [54, 18, 15]. It is unknown whether prototype-based classifiers can leverage the known exemplars (i.e. support images) to generalize better to unseen, rare inputs. To test this question, here, we compare kNN with the baseline ResNet-50 classifier (both described in Sec. 2) on ImageNet and related OOD datasets.

On ImageNet and ImageNet-ReaL, kNN performs slightly worse than ResNet-50 by -1.36 and -0.99 points, respectively (Table 1). Yet, interestingly, **on all four OOD datasets, kNN consistently outperforms** ResNet-50. Notably, kNN improves upon ResNet-50 by +1.66 and +4.26 points on DAmageNet and Adversarial Patch. That is, while ResNet-50 and kNN share the exact same backbone, the kNN's process of comparing the input image against the training-set examples prove to be beneficial for generalizing to OOD inputs. Intuitively, our results suggest that it is useful to "look back" at the training-set exemplars to decide a label for hard, long-tail or OOD images.

Consistently, using the same CUB-finetuned backbone, kNN is only marginally worse than ResNet-50 on CUB (85.46% vs. 85.83%; Table 1).

### 3.2 Visual correspondence-based explanations improve kNN robustness further

Recent work found that re-ranking kNN's shortlisted candidates using the patch-wise similarity between the query and training set examples can further improve classification accuracy on OOD data for some image matching tasks [29, 79, 75] such as face identification [29]. Furthermore, patch-level comparison is also useful in prototype-based bird classifiers [17, 22]. Inspired by these prior successes and the fact that EMD-Corr and CHM-Corr base the patch-wise similarity of two images on only 5 patch pairs instead of all $49 \times 49 = 2,401$ pairs as in [29, 79, 75], here we test whether our two proposed re-rankers are able to improve the test-set accuracy and robustness over kNN.

Table 1: Top-1 accuracy (%). ResNet-50 models' classification layer is fine-tuned on a specified training set in (b). All other classifiers are non-parametric, nearest-neighbor models based on pretrained ResNet-50 features (a) and retrieve neighbors from the training set (b) during testing. EMD-Corr & CHM-Corr outperform ResNet-50 models on all OOD datasets (e.g. +4.39 on Adversarial Patch) and slightly underperform on in-distribution sets (e.g. -0.72 on ImageNet-ReaL).

| Test set | Features (a) | Training set (b) | ResNet-50 | kNN | EMD-Corr | CHM-Corr | CHM-Corr+ |
|---|---|---|---|---|---|---|---|
| ImageNet [63] | ImageNet | ImageNet | **76.13** | 74.77 | 74.93 (-1.20) | 74.40 (-1.73) | n/a |
| ImageNet-ReaL [14] | ImageNet | ImageNet | **83.04** | 82.05 | 82.32 (-0.72) | 81.97 (-1.07) | n/a |
| ImageNet-R [35] | ImageNet | ImageNet | 36.17 | 36.18 | **37.75** (+1.58) | 37.62 (+1.45) | n/a |
| ImageNet Sketch [72] | ImageNet | ImageNet | 24.09 | 24.72 | 25.36 (+1.27) | **25.61** (+1.52) | n/a |
| DAmageNet [18] | ImageNet | ImageNet | 5.93 | 7.59 | **8.16** (+2.23) | 8.10 (+2.17) | n/a |
| Adversarial Patch [15] | ImageNet | ImageNet | 55.04 | 59.30 | 59.43 (+4.39) | **59.86** (+4.82) | n/a |
| CUB [71] | ImageNet | CUB | n/a | 54.72 | **60.29** | 53.65 | 49.63 |
| CUB [71] | iNaturalist [70] | CUB | **85.83** | 85.46 | 84.98 (-0.85) | 83.27 (-2.56) | 81.54 |

**Experiment** We run EMD-Corr and CHM-Corr on all datasets and compare their results with that of kNN (Table 1). Both methods (described in Sec. 2) re-rank the top $N = 50$ shortlisted candidates returned by kNN and then take the dominant class in the top-$k$ (where $k = 20$) as the predicted label.

**ImageNet results** Interestingly, despite using only 5 pairs of patches to compute image similarity for re-ranking, both classifiers consistently improve upon kNN further, especially on all OOD datasets. Overall, EMD-Corr and CHM-Corr outperform kNN and ResNet-50 baselines from +1.27 to +4.82 points (Table 1). Intuitively, in some hard cases where the main object is small, the two Corr classifiers ignore irrelevant patches (e.g. the sky in `ibex` images; Fig. 1) and only use the five most relevant patches to make decisions. Similarly, on Adversarial Patch, relying on a few key patches while ignoring adversarial patches enables our classifiers to outperform baselines (Fig. 2). See Appendix I for many qualitative examples comparing Corr and kNN predictions.

**CUB results** Interestingly, using the same ImageNet-pretrained backbones, EMD-Corr outperforms kNN by an absolute +5.57 points when tested on CUB (60.29% vs. 54.72%; Table 1). However, this difference vanishes when using CUB-pretrained backbones (Table 1; 84.98% vs. 85.46%).

Our CUB and ImageNet results are consistent and together reveal a trend: On i.i.d test sets, Corr models perform on par with kNN; however, on OOD images, they consistently outperform kNN, highlighting the benefits of patch-wise comparison.

### 3.3 Corr classifiers leverage five patches that are more important than five bird keypoints

EMD-Corr and CHM-Corr harness five patches per image for computing a patch-wise similarity score for a pair of images (Sec. 2.2). As these five patches are automatically inferred by cross-correlation (Fig. 3), it is interesting to understand further whether replacing these five patches by five user-defined patches in [71] would improve classification accuracy.

**Experiment** Since there are no keypoints provided for ImageNet, we test the importance of the five key patches chosen by Corr methods on CUB because CUB provides ornithologist-defined annotations for each bird image. That is, we create a baseline CHM-Corr+, which is the same as CHM-Corr, except that we use five important patches that correspond to five keypoints in a bird image—beak, belly, tail, right wing, and right foot—as described in Sec. 2.2. We also test CHM-Corr+ sweeping across the number of keypoints $\in \{5, 10, 15\}$.

**Results** On CUB, CHM-Corr outperforms CHM-Corr+ despite the fact that the baseline method leverages **five** human-defined bird keypoints (Table 1; 83.27% vs. 81.51%) while CHM-Corr may also use background patches. Interestingly, when increasing the number of keypoints to 10 and 15, the accuracy of CHM-Corr+ is still lower than that of CHM-Corr (i.e., from 81.51% to 82.34% and 82.27%, respectively). That is, 15 keypoints may correspond to $\leq 15$ different patches per image (15 if each keypoint lies in a unique, non-overlapping patch among all the 49 patches per image).

Our results show strong evidence that the five key patches inferred by CC used in EMD- and CHM-Corr do not necessarily cover the birds but are more important than expert-defined bird keypoints. Qualitative comparisons between CHM-Corr and CHM-Corr+ predictions are in Appendix J.

### 3.4 On ImageNet-ReaL, correspondence-based explanations are more useful to users than kNN explanations

Given that EMD-Corr and CHM-Corr classifiers outperform kNN classifiers on OOD datasets (Sec. 3.2), it is interesting to test how their respective explanations help humans perform classification on ImageNet. Furthermore, in image classification, kNN explanations were found to be more useful to humans than saliency maps [55].

**Experiment** We perform a human study to assess the ImageNet-ReaL classification accuracy of *users* of each classifier (described in Sec. 2.2) when they are provided with a classifier's predictions and explanations. That is, we measure the AI-assisted classification accuracy of users following human-AI interaction model 1 (Fig. 4a). We compare the accuracy between user groups of four classifiers ResNet-50, kNN, EMD-Corr, and CHM-Corr (described in Sec. 2.3.1).

Additionally, to thoroughly assess the impact of showing the correspondence boxes compared to showing only nearest neighbor images (e.g., CHM-Corr vs. CHM-NN in Fig. 1), we test two more user groups of EMD-NN and CHM-NN, i.e. the same explanations as those of the Corr classifiers but with the correspondence boxes *hidden*.

**Results** First, the mean accuracy of kNN users is consistently lower than that of the other models' users (e.g., 75.76% vs. 78.87% of EMD-Corr; Table 2). The EMD-Corr improvement over kNN is statistically significant ($p < 0.01$ via Mann-Whitney U test; Fig. 5a)

Second, interestingly, we find the differences between EMD-, CHM-Corr and their respective baselines are small and not statistically significant (Fig. 5a). That is, on ImageNet-ReaL, quantitatively, showing the correspondence boxes on top of nearest neighbors is not more useful to users. Third, surprisingly, the users of ResNet-50 (mean accuracy of 81.56%; Table 2) outperform all other methods' users, suggesting that on ImageNet, a task of many familiar classes to users, *ante-hoc* explanations hurt user accuracy rather than help. Note that in Nguyen et al. [55], post-hoc kNN explanations were found useful to humans compared to not showing any explanations. Yet, here, each classifier's

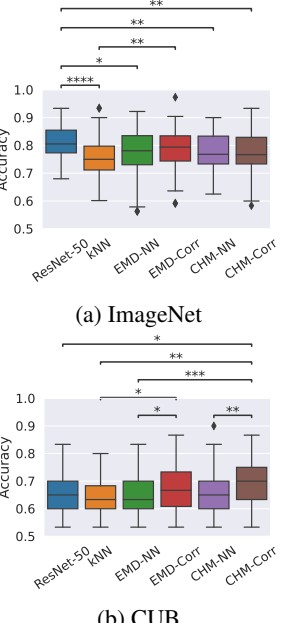

(a) ImageNet

(b) CUB

Figure 5: Mann-Whitney U test of the user accuracy scores of 6 methods.* = $p$-value < 0.05. ** = $p$-value < 0.01. *** = $p$-value < 0.001.

users are provided with a different set of images and AI decisions, which can also influence the user accuracy. When ResNet-50 is wrong, their users are substantially better in detecting such misclassifications compared to other models' users (Fig. A11a).

### 3.5 On CUB fine-grained bird classification, correspondence-based explanations are the most useful to users, helping them to more accurately reject AI misclassifications

To assess whether the findings on ImageNet-ReaL in Sec. 3.4 generalize to a fine-grained classification task, we repeat the user study on CUB—which is considered much harder to lay users than ImageNet.

**Results** Interestingly, we find EMD-Corr and CHM-Corr users consistently outperform ResNet-50, kNN, EMD-NN, and CHM-NN users (Table 2). The differences between EMD-Corr (or CHM-Corr) and every other baseline are statistically significant ($p < 0.05$ via Mann-Whitney U test; Fig. 5b). That is, on CUB, the visual correspondence boxes help users make more accurate decisions compared to (a) having no explanations at all (ResNet-50); (b) showing nearest neighbors sorted by image similarity only, not patch correspondences (Fig. 1a; kNN); and (c) having patch-wise correspondence neighbors but not displaying the boxes (Fig. A39; CHM-NN and EMD-NN).

**Corr explanations help users reject AI misclassifications while kNN is poorly trust-calibrated**
In an attempt to understand why the two Corr classifiers help users the most, we find that EMD-Corr and CHM-Corr users reject AI predictions at the highest rates (32.70% and 33.73%; Table A6) while kNN users reject the least (18.47%).

This might have led to the substantially higher accuracy of CHM-Corr users, compared to all other models' user groups, when *AI predictions are wrong* (Fig. A11b; e.g., 53.45% of CHM-Corr vs.

Table 2: Human-only accuracy (%)

| Method | ImageNet-ReaL | | CUB | |
|---|---|---|---|---|
| | Users | Accuracy | Users | Accuracy |
| ResNet-50 | 60 | **81.56** ± 5.54 | 60 | 65.50 ± 7.46 |
| kNN | 59 | 75.76 ± 8.55 | 59 | 64.75 ± 7.14 |
| EMD-Corr | 59 | **78.87** ± 6.57 | 58 | **67.64** ± 7.44 |
| CHM-Corr | 59 | 77.23 ± 7.56 | 59 | **69.72** ± 9.08 |
| EMD-NN | 57 | 77.72 ± 8.27 | 59 | 64.12 ± 7.07 |
| CHM-NN | 60 | 77.56 ± 6.91 | 60 | 65.72 ± 8.14 |

Table 3: AI-only and Human-AI team accuracy (%)

| Method | ImageNet-ReaL | | CUB | |
|---|---|---|---|---|
| | AI-only | Human-AI | AI-only | Human-AI |
| ResNet-50 | 86.11 | 88.63 (+2.52) | 87.38 | 87.45 (+0.07) |
| kNN | 85.95 | 87.24 (+1.29) | 87.40 | 86.66 (-0.74) |
| EMD-Corr | 85.91 | 88.02 (+2.11) | 86.88 | 86.86 (-0.02) |
| CHM-Corr | 85.36 | 87.89 (+2.53) | 85.48 | 86.25 (+0.77) |
| *mean* | 85.83 | 87.94 (+2.11) | 86.78 | 86.80 (+0.02) |

41.22% of ResNet-50). That is, CHM-Corr users correctly reject 53.45% of the images that the CHM-Corr classifier mislabels. In contrast, kNN users reject the least, only 33.22% of incorrect predictions (Fig. A11b). kNN explanations tend to *fool* users into trusting the kNN's wrong decisions (Fig. 6)—the accuracy of kNN users is 33.22%, much lower than the 41.22% of ResNet-50 users who observe no explanations. On ImageNet (Fig. A11), kNN is also poorly "trust-calibrated" [69, 74].

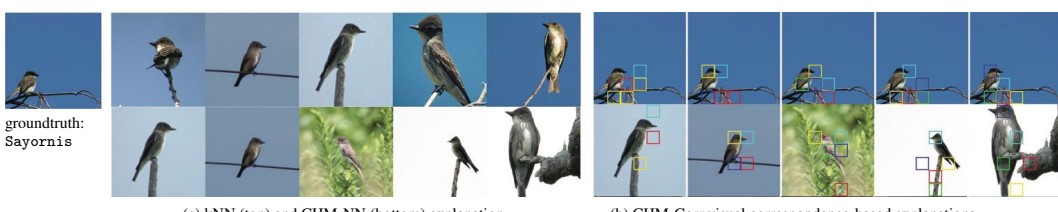

groundtruth:
Sayornis

(a) kNN (top) and CHM-NN (bottom) explanation      (b) CHM-Corr visual correspondence-based explanations

Figure 6: A `Sayornis` bird image is **misclassified** into `Olive Sided Flycatcher` by both kNN and CHM-Corr models. Yet, all 3/3 CHM-Corr users correctly rejected the AI prediction while 4/4 kNN users wrongly accepted. CHM-Corr explanations (b) show users more diverse samples and more evidence against the AI's decision. More similar examples are in Appendix H.1.

We hypothesize that kNN explanations tend to fool users more as their nearest neighbors, by design, show images that are *image-wise* similar to the query (regardless of whether the kNN prediction is correct or not) while EMD-Corr and CHM-Corr re-rank the images based on *patch-wise* similarity. Furthermore, we find the **images in kNN explanations are also less diverse** than those in Corr explanations in both LPIPS [77] and MS-SSIM [73] (Appendix H.2). Corr explanations tend to include more diverse images (Fig. 6a; top vs. bottom) and provide users with more contrastive evidence in order to reject AI's incorrect predictions.

Additionally, we also hypothesize that users are less confident about AI's decisions (and thus reject more) when Corr explanations show some background and uninformative patches used in the matching process (e.g., the 1st and 4th image in Fig. 6b). Yet, such boxes are not available in kNN explanations.

**When Corr explanations allow for more disagreement between AI and users, humans also tend to incorrectly reject AI's correct predictions more often** (Fig. A11b; Corr users are the least accurate among 6 methods when the AI is correct). EMD-Corr and CHM-Corr users score ∼4 points below ResNet-50 users (84.94% and 85.99% vs. 89.87% of ResNet-50). When most users reject AI's correct predictions, we observe that some discriminative features (e.g., the belly stripes of the Field Sparrow; Fig. A35c) are often occluded in the query, leading to human-AI disagreement.

# 4 Related Work

**Patch-wise similarity**    Calculating patch-wise similarity, either intra-image [23] or inter-image [29, 79, 75], has been useful in many tasks as the comparison enables machines to attend to fine-grained details and compute more accurate decisions. Our EMD-Corr harnesses a similar approach to that in [29, 79]; which, however, was not tested on ImageNet classification as in our work. Furthermore, we only compute the total patch-wise similarity over the top-5 patch pairs between the query and each exemplar instead of all pairs as in [29, 79]. Compared to recent patch-wise similarity works that use either cross-attention in ViTs [23] or EMD [79, 75, 29, 43], our work is the first to perform human evaluation of the correspondence-based explanations.

**Prototype-based XAI methods**   Our work is motivated by the recent finding that exemplar-based explanations are more effective than heatmap-based explanations in improving human classification accuracy [55, 38, 42, 24]. However, showing an entire image as an exemplar without any further localization may be confusing as it is unknown which parts of the image the AI is paying attention to [55, 38]. Our EMD-Corr and CHM-Corr present a novel combination of heatmap-based and prototype-based XAI approaches. None of the prior prototype-based XAI methods that operate at the patch level [17, 53, 79, 22] (see Table 3 in [22]) were tested on humans yet. Also, in preliminary tests, we find their explanation formats too dense (i.e., showing over 10 prototypes [17], 9 correspondence pairs per image [22], or an entire prototype tree to humans [52, 53]) to be useful for lay users.

Another major difference is that our Corr classifiers are nonparametric, allowing the training set to be adjusted or swapped with any external knowledgebase for debugging purposes. In contrast, recent prototype-based classifiers [17, 53, 79, 22] are parametric, using a set of learned prototypes and thus may not perform well on OOD datasets as EMD-Corr and CHM-Corr.

**Post-hoc prototype-based explanations**   Some prototype-based methods are post-hoc [41, 55, 20], i.e., generating explanations to explain a decision after-the-fact, which could be highly unfaithful [61, 62]. Instead, our approach is inherently interpretable [62], i.e., retrieving the patches first, and then using them to make classification decisions. While our binary classification task is adopted from [55], our study compares 4 different classifiers while Nguyen et al. [55] instead tested a single classifier with multiple post-hoc explanations.

**Human studies**   Our study has 709 users in total, i.e. ∼60 users per method per dataset, which is substantially larger than that in most prior works. That is, ∼30 and 40 users per method participated in [55] and in [49], respectively while Adebayo et al. [9] had 54 persons in total for the entire study of multiple methods.

**Human-AI teaming**   Human-AI collaboration is becoming more essential in the modern AI era [32]. A large body of prior works has investigated such collaboration in other domains (e.g., NLP [12, 76], healthcare [16] and others [33, 16, 78, 19]); however, only few works investigated human-AI collaboration in the image classification setting [55, 42, 24].

Some prior works predict when to defer the decision-making to humans [58, 37, 40]. However, by simply offloading some inputs to humans based on confidence scores, we achieve complementary human-AI team performance in both ImageNet and CUB. Previous works [12, 64] found that algorithmic explanations benefit human decision-making in general, but did not find XAI methods to yield team complementary performance [32], which we report in this work.

## 5   Discussion and Conclusion

**Limitations**   Due to the limited amount of time and expensive cost of computation related to EMD-Corr or CHM-Corr, we did not experiment on a wider range of OOD datasets (e.g., adversarial poses [10]). We tested our methods on ImageNet-A [36], ObjectNet [13], and ImageNet-C [34] as well, but on a small scale of 5K-image sets (see Table A2). As using online crowdworkers for the XAI human evaluation, we share the same limitations with [55, 9, 42]. That is, despite our best efforts to minimize biases, the human data quality can be improved in highly-controlled laboratory conditions like in [27]. Algorithm-wise, EMD-Corr and CHM-Corr are re-ranking methods and therefore run substantially slower than ResNet-50 (see Fig. A10 for a speed comparison of all models).

Our work is the first attempt to: (1) study the effectiveness of patch-wise comparison in improving the robustness of deep image classifiers on ImageNet OOD benchmarks; (2) show the utility of visual correspondence-based explanations in helping users make more accurate image-classification decisions; (3) achieve human-AI complementary team performance in the image domain.

### Acknowledgement

The authors would like to thank Ken Stanley for the great idea of using correspondences as an explanation for image classification, leading to this work. We also thank Thang Pham, Peijie Chen, and Hai Phan for feedback and discussions of the earlier results. AN was supported by the NSF Grant No. 1850117 & 2145767, and donations from NaphCare Foundation & Adobe Research.

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
