# Appendix for:
# Visual correspondence-based explanations improve AI robustness and human-AI team accuracy

## A Implementation details

### A.1 Fine-tuning iNaturalist-pretrained ResNet-50 for CUB

To make a 200-way classifier using the ResNet-50 model from iNaturalist [4], we remove the 5089-way classification head and add an average pooling layer followed by a linear feed-forward layer with 200 units. We keep all the initialization parameters unchanged and use the Adam optimizer [44] without any hyperparameter tuning. We train the new layer using the CUB training set for 200 epochs. We do not train the intermediate layers since this backbone is shared among all methods (i.e., we freeze all the convolutional layers in the ResNet-50 model). The iNaturalist-pretrained ResNet-50 model has a slight difference compared to the PyTorch reference implementation [8]. This network has 18 extra layers in the last convolutional blocks, but the spatial dimension matches the original ResNet-50 (i.e., $2048 \times 7 \times 7$).

### A.2 Implementation details for kNN

We implement a vanilla kNN classifier that operates at the deep feature space of ResNet-50. That is, given a query image $Q$, we sort all training-set images $\{G_i\}$ based on their distance $D(Q, G_i)$, which is the cosine distance between the two corresponding image features $f(Q)$ and $f(G_i) \in \mathbb{R}^{2048}$ at layer4 of ResNet-50, after avgpool (see code):

$$\mathrm{D}\left(Q, G_i\right) = 1 - \frac{\langle f(Q), f(G_i) \rangle}{\|f(Q)\| \, \|f(G_i)\|} \tag{1}$$

where $\langle \cdot \rangle$ is the dot product, and $\|\cdot\|$ is the $L_2$ norm operator.

### A.3 Implementation details for EMD-Corr

We incorporate the Earth Mover's Distance (EMD) into a 2-stage hierarchical image retrieval, similar to [79, 29]. In the first stage, the kNN classifier selects the $N$ images with the lowest cosine distance – $G_i$ – to the query $Q$. Then, we sort these $N$ images (a.k.a. re-ranking [29]) using patch-wise similarity derived from EMD. The predicted label is finally determined by a majority vote of the labels of the top-k images, as in the kNN classifier, where $k \leq N$. In our classifier, we set $k = 20$ and $N = 50$.

Our patch-wise comparison algorithm in stage 2 (shown in Fig. 3a) is different from [29, 79, 75] as the similarity of an image pair is not determined by all possible patches. While the first stage retrieved images using global features, comparing only a few most similar patches by EMD offers benefits: (1) helping classifiers capture the distinctive image regions only (e.g., head-to-head comparison for birds); and (2) achieving human interpretability as looking at all possible pair-wise comparisons is impossible. We denote each patch-by-patch comparison as "correspondence".

The most similar patches between two images $Q$ and $G$ – both divided into $M$ patches – are found as a set of 2-D coordinates $L$ containing the *highest* values in a *flow* matrix $\boldsymbol{F}$. Let $\mathcal{Q} =$

$\{(q_1, w_{q_1}), (q_2, w_{q_2}), \cdots .(q_M, w_{q_M})\}$ and $\mathcal{G} = \{g_1, w_{g_1}), (g_2, w_{g_2}), \cdots .(g_M, w_{g_M})\}$ denote two sets of non-overlapping image patches, $g_i$ and $g_j$ are the patch embeddings; and $w_{q_i}$ and $w_{g_j}$ are the corresponding importance assigned by a feature weighting algorithm (e.g., Cross Correlation used in [79]). We derive $\boldsymbol{F} = (f_{ij}) \in \mathbb{R}^{M \times M}$ by minimizing the *transport plan cost* in Eq. 2.

$$\text{Cost}(Q, G, \boldsymbol{F}) = \sum_{i=1}^{M} \sum_{j=1}^{M} d_{ij} f_{ij} \qquad (2)$$

where $f_{ij} \geq 0$ and $\sum_{j=1}^{M} \sum_{i=1}^{M} f_{ij} = 1$. We use Eq. 1 to compute the ground distance $d_{ij}$ and run the Sinkhorn algorithm [21] for 100 iterations to seek the *optimal transport plan* $\boldsymbol{F}$. To assign importance weights (i.e., $w_{q_i}$ and $w_{g_j}$), we use cross-correlation (CC) maps from [68].

Finally, using $\boldsymbol{F}$ and D from Eq. 1, the EMD distance between $Q$ and $G$ is computed by Eq. 3. Since we are interested in patch-wise comparison, the features used in stage 2 for $Q$ and $G$ are layer4 from [8]. Our EMD-Corr classifier's stage 2 solely relies on EMD distance for re-ranking instead of mixing EMD and cosine distance like in previous works [79, 29].

$$\text{d}_{\text{EMD}}(D, \boldsymbol{F}) = \sum_{(i,j) \in L} d_{ij} f_{ij} \qquad (3)$$

### A.4  Implementation details for CHM-Corr classifier

Similar to the EMD-Corr classifier, this classifier also consists of 2 stages – selecting $N$ most similar images to the query $Q$ by the kNN classifier, followed by a correspondence-based re-ranking algorithm. For re-ranking, we propose to use a Convolutional Hough Matching network (CHM) [51] to first infer semantic correspondences between $Q$ and $G$, then calculate the similarity score between the two images based on a subset of these correspondences.

The re-ranking algorithm starts with dividing both $Q$ and $G$ into $M$ patches, resulting in two set of $\mathcal{Q} = \{q_1, q_2, \cdots .q_M\}$ and $\mathcal{G} = \{g_1, g_2, \cdots .g_M\}$ image patches. To find the semantic correspondences between two images, we make use of the CHM network to transfer keypoints from the query image $Q$ to image $G$.

The CHM network finds correspondence between two given images in three stages: feature extraction and correlation computation, Hough matching, and keypoint transfer. In the first stage, the CHM network extracts features from multiple layers of a ResNet-101 network to construct a set of multi-scale features $\{(\mathbf{F}_Q, \mathbf{F}_G)\}_{s=1}^{S}$. The feature volume is then used to construct a correlation tensor by comparing all possible pairs in the feature space of two images. In the second stage, the correlation tensor is fed into a Convlutioanl Hough Matching (CHM) layer to perform Hough voting in the space of translation and scale to find candidate matches between two images. In the last stage, a kernel soft-argmax [46] is applied to the output of the CHM layer to create a dense flow field, and then correspondence keypoints are extracted using a soft sampler.

After finding visual correspondence between two images, we assign an importance weight $w_{i,j}$ for the pair $(q_i, g_j)$ using cross-correlation maps from [68]. Finally, the distance between $Q$ and $G$ is the average distance between 5 patch pairs with the lowest cosine distance.

We use the reference implementation of the Convolutional Hough Matching Network pretrained on the PF-Pascal Dataset [30]. There are three variations of CHM networks depending on the parameter sharing strategy, i.e., psi, iso, and full. Our ablation study (Appendix B.3) shows similar performance on a 5K subset of the ImageNet dataset. We select psi with a threshold of $T = 0.55$ for the CHM-Corr classifier.

The CHM network requires a set of initial keypoints on the source image, i.e., a set of keypoint on the query image $Q$. Although some datasets come with this annotation information, generally, this information is not available. To have a comparable classifier with our EMD-Corr classifier, we discretize an image into a $7 \times 7$ grid, resulting in 49 non-overlapping patches. For each patch, we pick a point at its center.

For assigning importance weight $w_{i,j}$ to $(q_i, g_j)$ pair, we first calculate the cross-correlation map between the two images $Q$ and $G$. Calculating a cross-correlation map using the last convolutional layer of the ResNet-50 model will result in two $7 \times 7$ maps for each $Q$ and $G$. For assigning importance weights, we binarize the cross-correlation for $Q$, using a threshold of $T = 0.55$, i.e., we zero out all pairs in the non-salient part according to $Q$, by setting their importance weights to $0$, and for the remaining patches, we set the weights to $1$.

After removing non-salient patch pairs in the last stage, we calculate the cosine similarity between pair $(q_i, g_j)$ using the corresponding feature volume in the last convolutional layer of the ResNet-50 model. The similarity score is the average similarity between top $5$ pairs with the highest cosine similarly.

## A.5  Generating Adversarial Patch dataset

Brown et al. [15] generated a *universal* adversarial patch to fool image classifiers into recognizing everything as `toaster`. This patch misleads the models' attention, by having them look only at the most salient item while ignoring the remaining pixels. We apply this attack on ImageNet validation set, resulting in 50K Adversarial Patch images of $240 \times 240$ px. The patches are circles with a size of $5\%$ the input image, targeting ResNet-50 [31] classifying everything as `toaster` with a target confidence of 90%. The maximum attack iteration for each sample is 500. We only train to optimize the adversarial patch on the ImageNet validation set for one epoch and save the immediate samples for the dataset. We adapt the code from [2] and make minor modifications.

To obtain our Adversarial Patch dataset, from the main repository, you can run the below command to generate the dataset or download the dataset here.

```
cd datasets/adversarial-patch/
python make_patch.py --cuda --epochs 1 --patch_size 0.05 --max_count 500
--netClassifier resnet50 --patch_type circle --train_size 50000
--test_size 0 --image_size 240 --outf output_imgs
```

# B Ablation study and small-scale experiments on ImageNet OODs

## B.1 Different hyperparameters for EMD

Table A1: Accuracy of the EMD-Corr classifier with different EMD hyperparameters (%)

| Datasets | Number of Images | Cross Correlation Corrs-Num$= 5$ $k = 20$ | Cross Correlation Corrs-Num$= 49 \times 49$ $k = 20$ | Uniform Corrs-Num$= 5$ $k = 20$ |
|---|---|---|---|---|
| ImageNet 2012 | 50,000 | 74.93 | 74.59 | 74.47 |
| CUB (iNaturalist ResNet) | 5,794 | 84.98 | 85.42 | 79.72 |
| CUB (ImageNet ResNet) | 5,794 | n/a | 59.44 | 53.47 |

## B.2 Performance of classifiers on a 5K subset of different datasets

Table A2 contains details about the performance of different classifiers on a 5K subset of various OOD datasets.

Table A2: Performance of classifiers on 5K subsets of various OOD datasets – (Accuracy %)

| Datasets | ResNet-50 | kNN | EMD-Corr | CHM-Corr |
|---|---|---|---|---|
| ImageNet [63] | 75.00 | 74.62 | 74.66 | 74.52 |
| ImageNet-R [35] | 35.68 | 34.60 | 35.66 | 36.18 |
| ObjectNet [13] | 36.54 | 34.80 | 36.56 | 35.60 |
| ImageNet Sketch [72] | 23.84 | 23.92 | 24.40 | 25.28 |
| ImageNet-A [36] | 0.00 | 0.32 | 0.50 | 0.46 |
| DAmageNet [18] | 6.38 | 8.92 | 9.72 | 9.06 |
| ImageNet-C Gaussian noise (Level 1) [34] | 59.56 | 59.62 | 59.70 | 59.62 |
| ImageNet-C Gaussian blur (Level 1) [34] | 66.12 | 65.68 | 65.68 | 65.68 |

## B.3 Different weights for CHM

Table A3: Accuracy of the CHM-Corr classifier on a 5K subset of ImageNet [63] with different CHM parameters (%)

| Method | Threshold | | | | | | | |
|---|---|---|---|---|---|---|---|---|
| | 0.2 | 0.3 | 0.4 | 0.5 | 0.55 | 0.6 | 0.7 | 0.8 |
| PSI | 74.26 | 74.36 | 74.36 | 74.26 | **74.52** | 74.38 | 74.44 | 73.78 |
| ISO | 74 | 74.04 | 74 | 74.18 | 74.24 | **74.28** | 74.1 | 73.76 |
| FULL | 74.62 | 74.62 | 74.48 | **74.64** | 74.4 | 74.44 | 74.56 | 74.02 |

## C  Runtime comparison between all methods

In this section, we provide a runtime analysis of all classifiers on a batch of 1000 random queries. For each classifier, we run the classification five times and report the average and standard deviation. We use a single NVIDIA V100 GPU with 16 gigabytes of memory to perform our benchmarks.

Here we also provide a FAISS [39] implementation of the kNN classifier, which is significantly faster than the naive GPU implementation for the nearest neighbor search problem. The FAISS version of kNN requires one-time preprocessing to extract embeddings from the training set. This process takes just a few minutes for the CUB dataset, which has only 5.9K images. For ImageNet, which consists of 1.2 million images, we use a single NVIDIA A100 (40 GB) to extract and cache the embeddings on disk. This process takes less than 90 minutes, and the resulting cache file takes 9.8 GB of disk space. We also use the Linux's `time` tool to calculate the total memory usage of kNN using FAISS during the inference. The peak memory performance (`Maximum resident set size`) for the 1000 images is around 31 GB.

Table A4: Runtime (in seconds) for a set of 1,000 queries averaged over 5 runs – kNN inference is fairly tractable using a FAISS implementation.

| Method | Dataset | |
| --- | --- | --- |
| | ImageNet | CUB |
| ResNet-50 | $9.17 \pm 0.19$ | $8.81 \pm 0.14$ |
| kNN (FAISS - CPU) | $17.35 \pm 1.28$ | $9.7 \pm 0.32$ |
| kNN (Naive - GPU) | $1,112.46 \pm 0.86$ | $23.88 \pm 0.58$ |
| EMD-Corr reranking step | $2,218.92 \pm 99.14$ | $1,927.69 \pm 17.48$ |
| CHM-Corr reranking step | $10,642.85 \pm 1007.87$ | $6,920.76 \pm 67.58$ |

# D Sample explanations

This section contains sample visualizations for kNN, EMD-Corr, and CHM-Corr classifiers.

## D.1 kNN

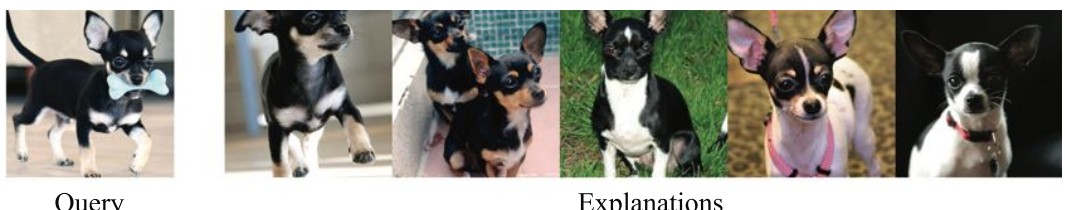

Query                                    Explanations

Figure A1: A sample explanation of the kNN classifier when classifying a `chihuahua` image.

## D.2 EMD-NN

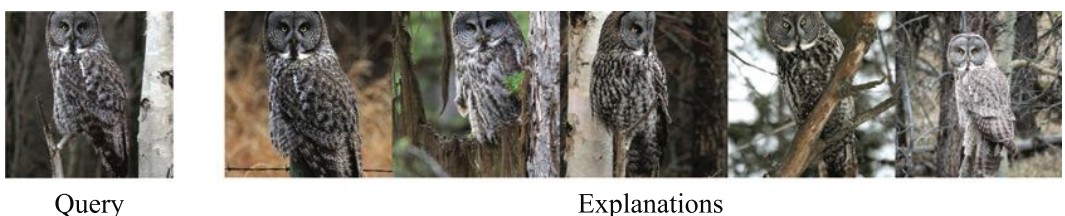

Query                                    Explanations

Figure A2: A sample EMD-NN explanation of the EMD-Corr classifier when classifying a `great grey owl` image. EMD-NN shows only the nearest neighbors after re-ranking.

## D.3 EMD-Corr

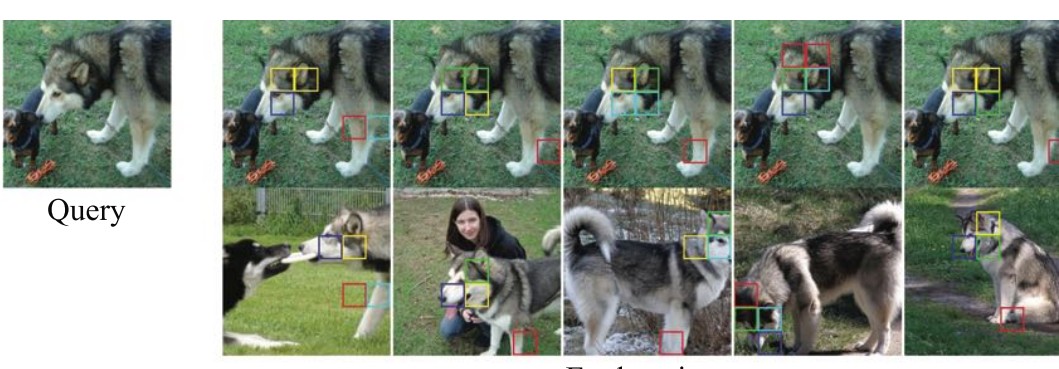

Query

Explanations

Figure A3: A sample explanation of the EMD-Corr classifier when classifying a `malamute` image.

### D.4 CHM-NN

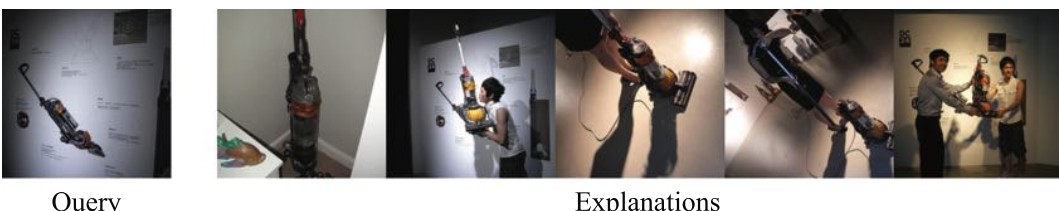

Query                                    Explanations

Figure A4: A sample CHM-NN explanation of the CHM-Corr classifier when classifying a `vacuum` image. CHM-NN only shows the nearest neighbors after re-ranking.

### D.5 CHM-Corr

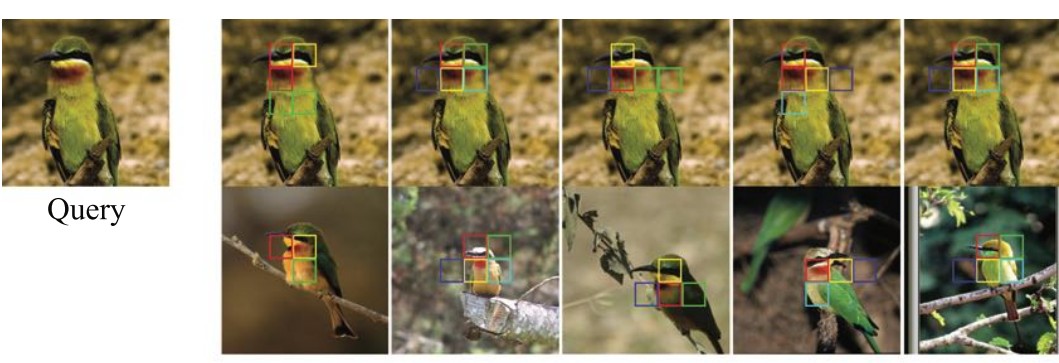

Query

Explanations

Figure A5: A sample explanation of the CHM-Corr classifier when classifying a `bee eater` image.

# E    Sample screens and training examples for human studies

## E.1    Sample screens from human studies

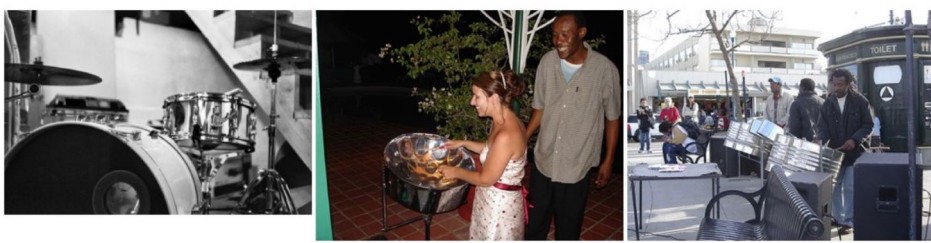

steel drum: a concave percussion instrument made from the metal top of an oil drum

(a) ImageNet studies

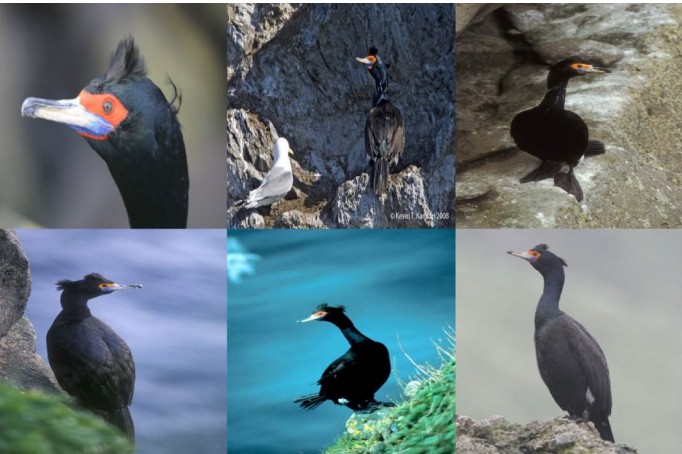

Red-faced cormorant

Continue

(b) CUB studies

Figure A6: In ImageNet-ReaL experiments, before each trial, where users are asked if the query image belongs to the top-1 class $c$ (here, `steel drum`), we show three representative images from $c$ along with a 1-sentence WordNet description (a). Instead of showing 3 images, in CUB experiments, we offer 6 images from the top-1 class (here, `Red-faced Cormorant` to help users better recognize the distinct features of each bird.

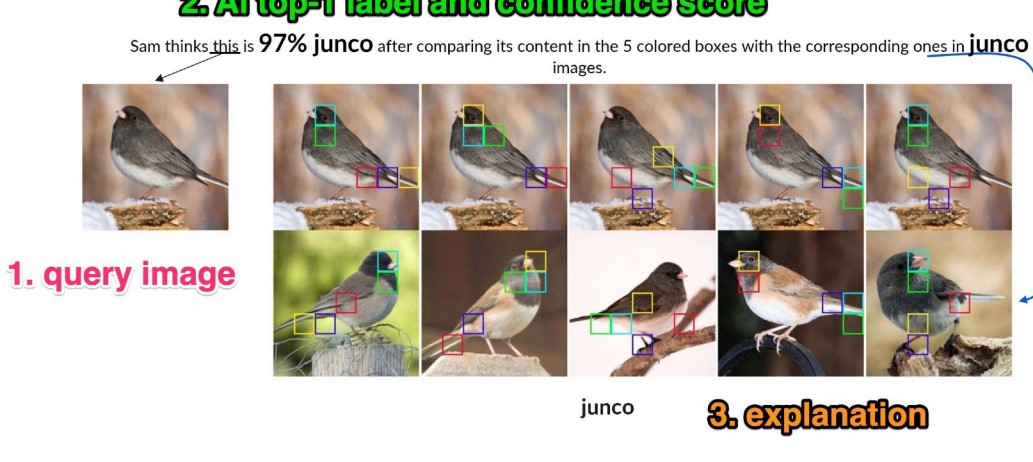

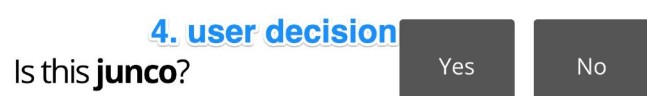

Figure A7: A sample screenshot from a human study of EMD-Corr users. Each user is provided with (1) the query image; (2) the AI top-1 predicted label and confidence score; and (3) an explanation, here the visual correspondence-based explanations of EMD-Corr. They are asked to provide a Yes/No answer to whether the query is an image of junco.

## E.2 Sample groundtruth cases used in the Validation phase of our CUB human studies

Below are example cases that we manually choose to be "groundtruth" in order to control for user quality during the validation phase.

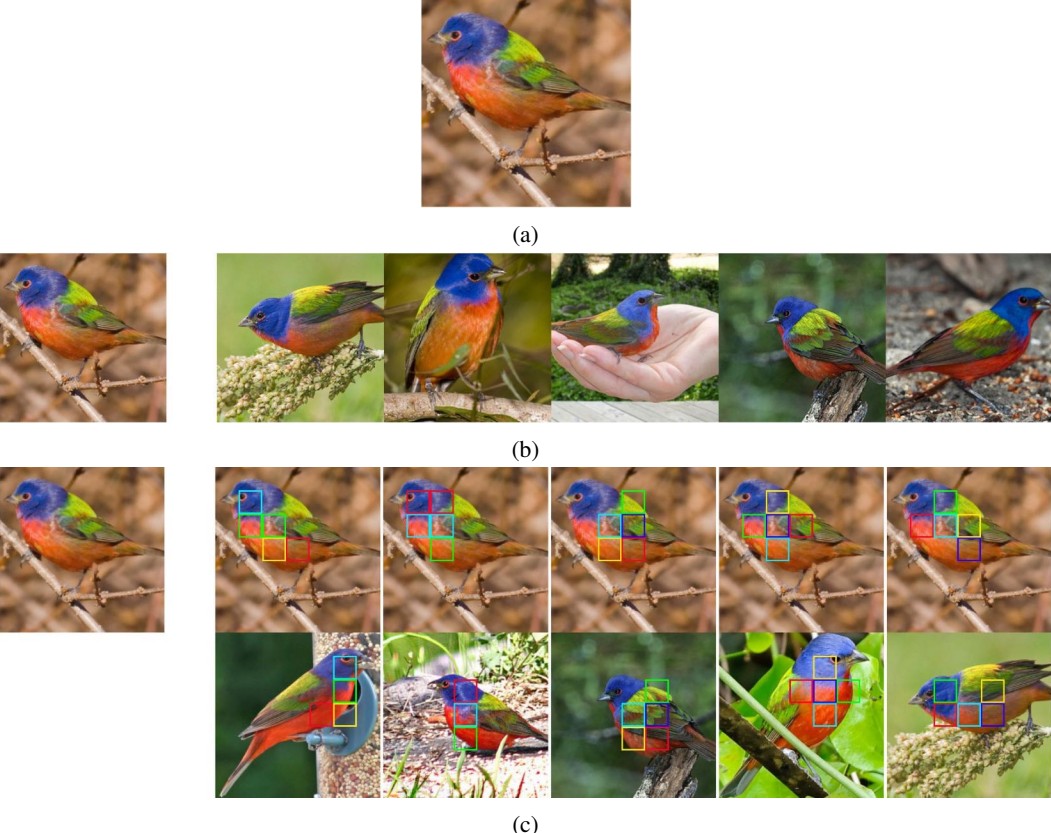

Figure A8: A **groundtruth Yes** validation sample in a CUB study. That is, users are expected to select Yes when being presented with these explanations. The bird is `Painted Bunting`.
(a) ResNet-50—no explanations provided.
(b) kNN nearest-neighbor explanation.
(c) EMD-Corr explanation.

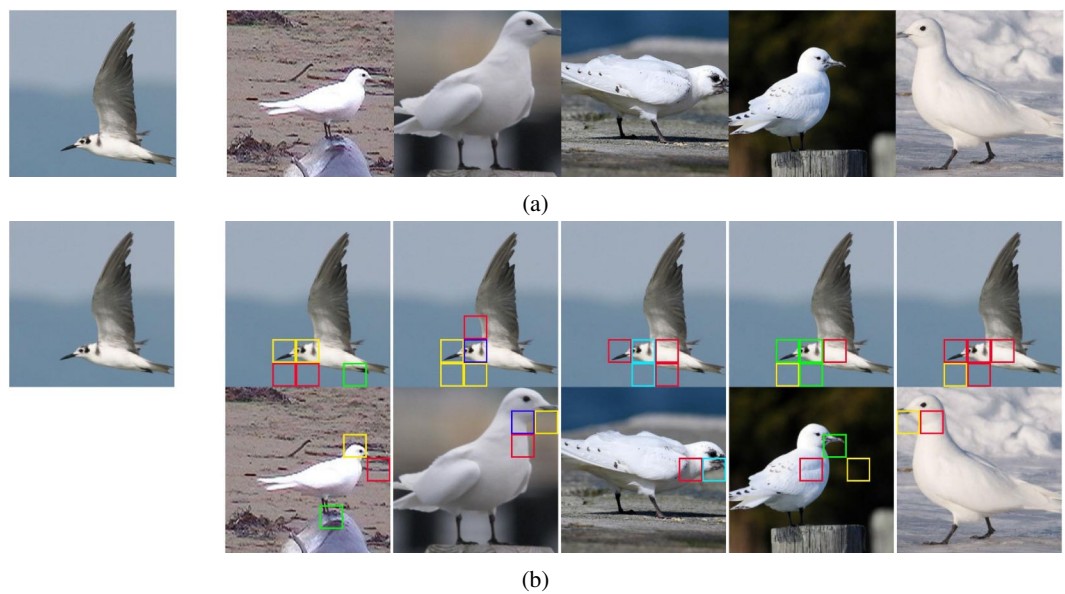

(a)

(b)

Figure A9: A **groundtruth No** validation sample in a CUB study. That is, users are expected to select No when being presented with these explanations. The bird is `Black Tern`.
(a) EMD-NN explanation.
(b) EMD-Corr explanation.

# F Human-AI team performance analysis

This section provides more details about AI and team performance.

## F.1 Defining the difficulty level of queries

To further understand the performance of the classifiers in our study, we categorize each query image into Easy, Medium, and Hard categories based on the model's confidence score and the correctness of the top-1 label (see Table A5). This breakdown allows us to analyze model and user behaviors at a specific level of difficulty.

Table A5: Difficulty levels

|  | **Easy** | **Medium** | **Hard** |
| --- | --- | --- | --- |
| **AI is Correct** | confidence $\in [0.75, 1)$ | confidence $\in [0.35, 0.75)$ | confidence $\in [0, 0.35)$ |
| **AI is Wrong** | confidence $\in [0, 0.35)$ | confidence $\in [0.35, 0.75)$ | confidence $\in [0.75, 1)$ |

## F.2   The acceptance and rejection ratios

In Table A6 we provide details about whether users accepted or rejected AI's decisions for each type of classifier.

Table A6: The frequency of users accepting or rejecting AI's decision per classifier (%).

| Method | ImageNet-ReaL | | CUB | |
|---|---|---|---|---|
| | Accept | Reject | Accept | Reject |
| ResNet-50 | 60.44 | 39.56 | 74.28 | 25.72 |
| kNN | **69.60** | 30.40 | **81.53** | 18.47 |
| EMD-Corr | 64.92 | 35.08 | 67.30 | **32.70** |
| CHM-Corr | 67.51 | 32.49 | 66.27 | **33.73** |
| EMD-NN | 67.49 | 32.51 | 78.76 | 21.24 |
| CHM-NN | 68.94 | 31.06 | 76.94 | 23.06 |

Table A7 shows the ratio of accepts and rejects based on the difficulty level described in Sec. F.1.

Table A7: The ratio of users accepting or rejecting AI's decision per difficulty level (%)

| Difficulty Level | ImageNet-ReaL | | CUB | |
|---|---|---|---|---|
| | Accept | Reject | Accept | Reject |
| Easy | 72.7 | 27.3 | 82.75 | 17.25 |
| Medium | 58.42 | 41.58 | 66.43 | 33.57 |
| Hard | 62.38 | 37.62 | 78.34 | 21.66 |

## F.3   Time performance of users

Fig. A10 shows the average time distribution to finish each trial per method.

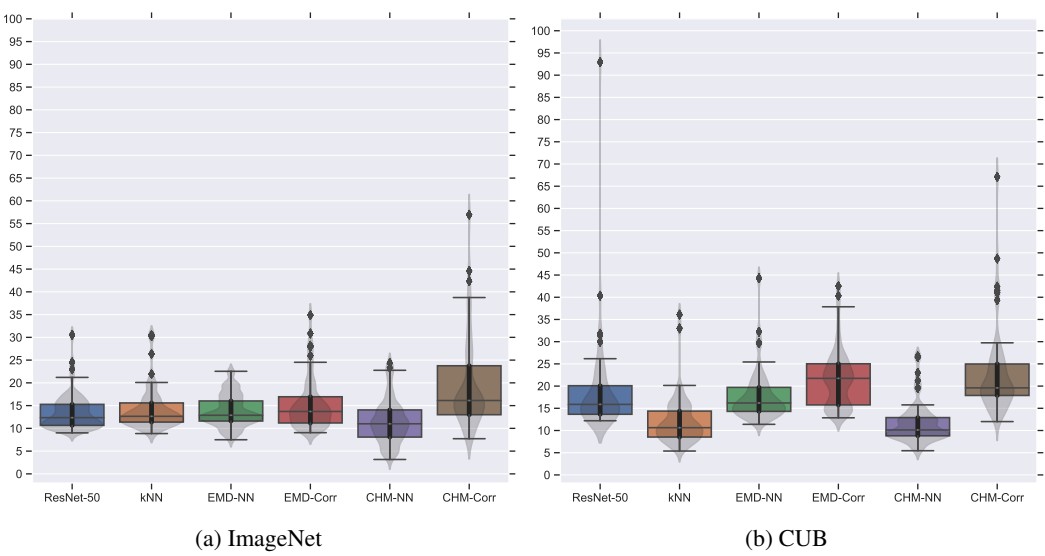

(a) ImageNet                                    (b) CUB

Figure A10: Distribution of the average time taken for each trial (Seconds)

## F.4 Human performance analysis based on AI correctness

Figure A11 shows the breakdown of user accuracy based on the correctness of AI predictions on ImageNet-ReaL and CUB datasets.

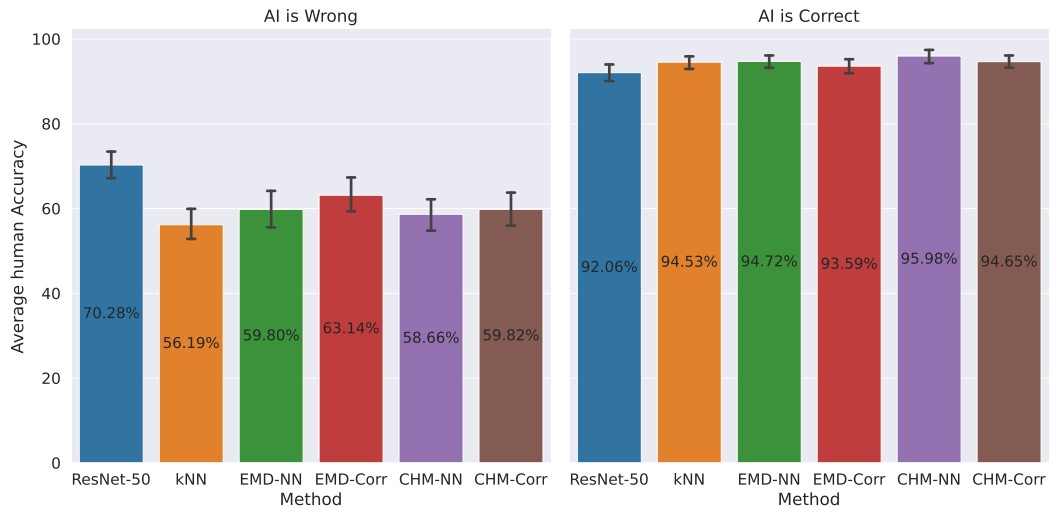

(a) ImageNet-ReaL – Mean user accuracy

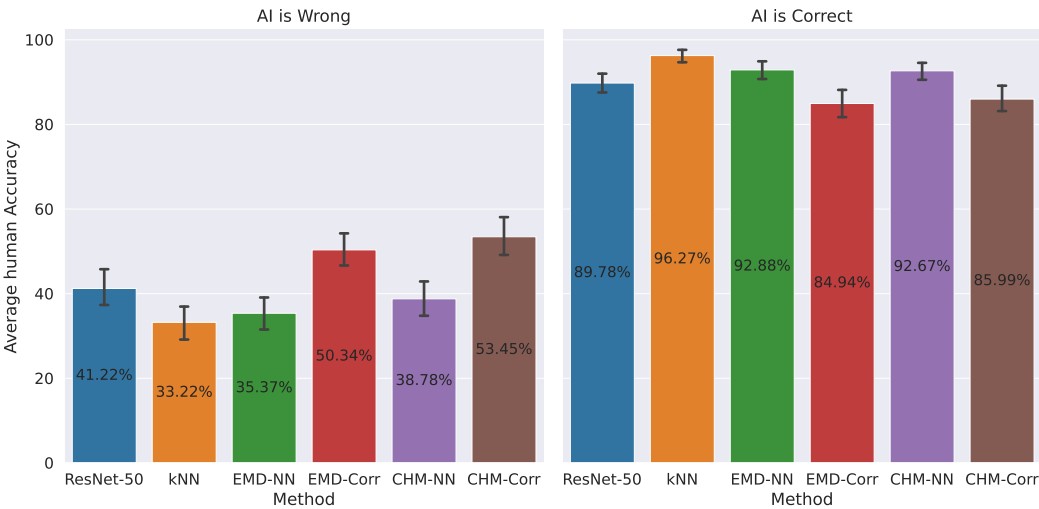

(b) CUB – Mean user accuracy

Figure A11: The breakdown of human performance by AI correctness

## F.5 Human performance analysis based on the difficulty of the query

In this section, we calculate the average user accuracy based on the difficulty level of the query (described in Sec F.1) and the correctness of AI's prediction.

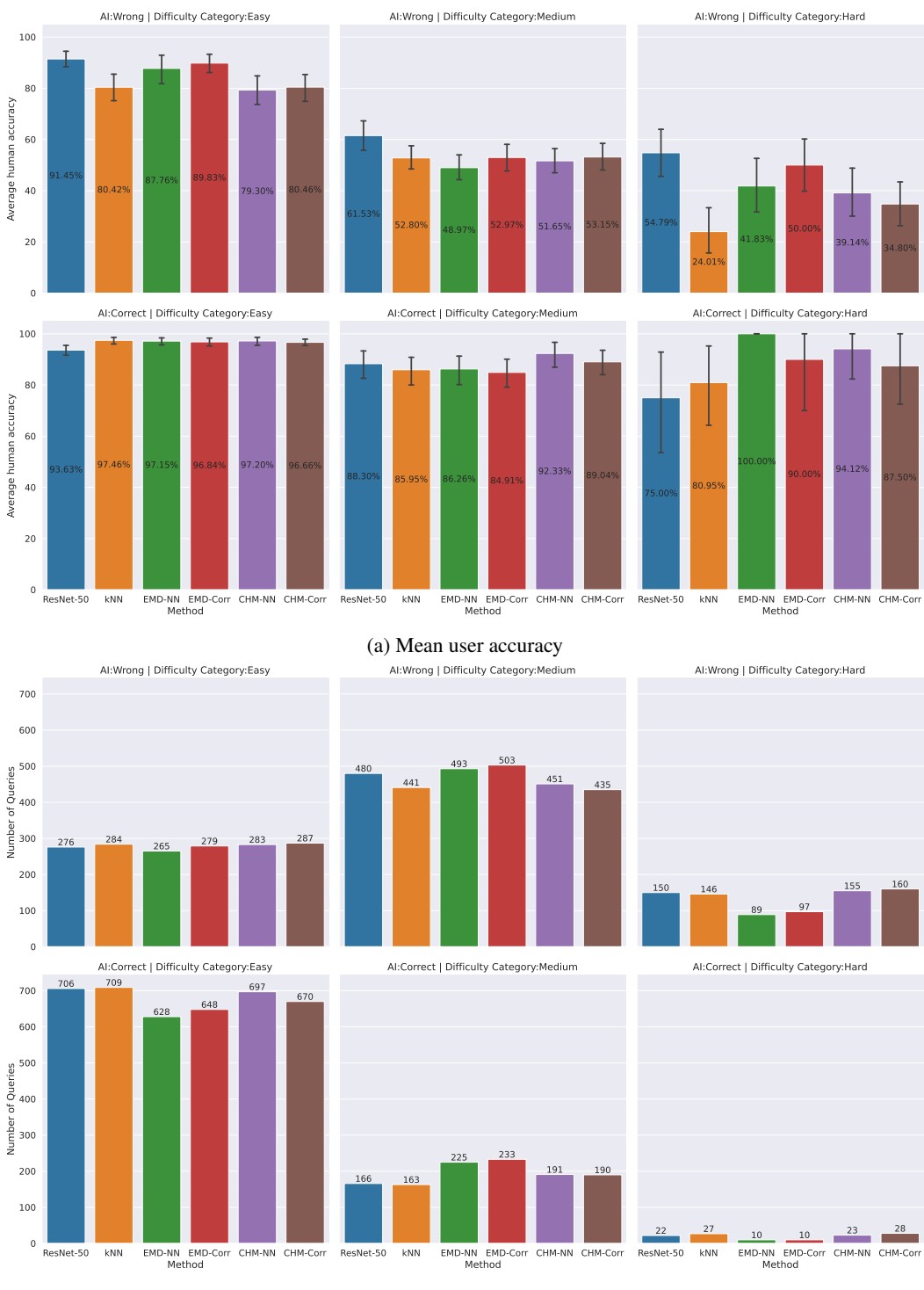

(a) Mean user accuracy

(b) Number of queries

Figure A12: ImageNet – The breakdown of the human performance by 'Difficulty Level' and 'AI Correctness'

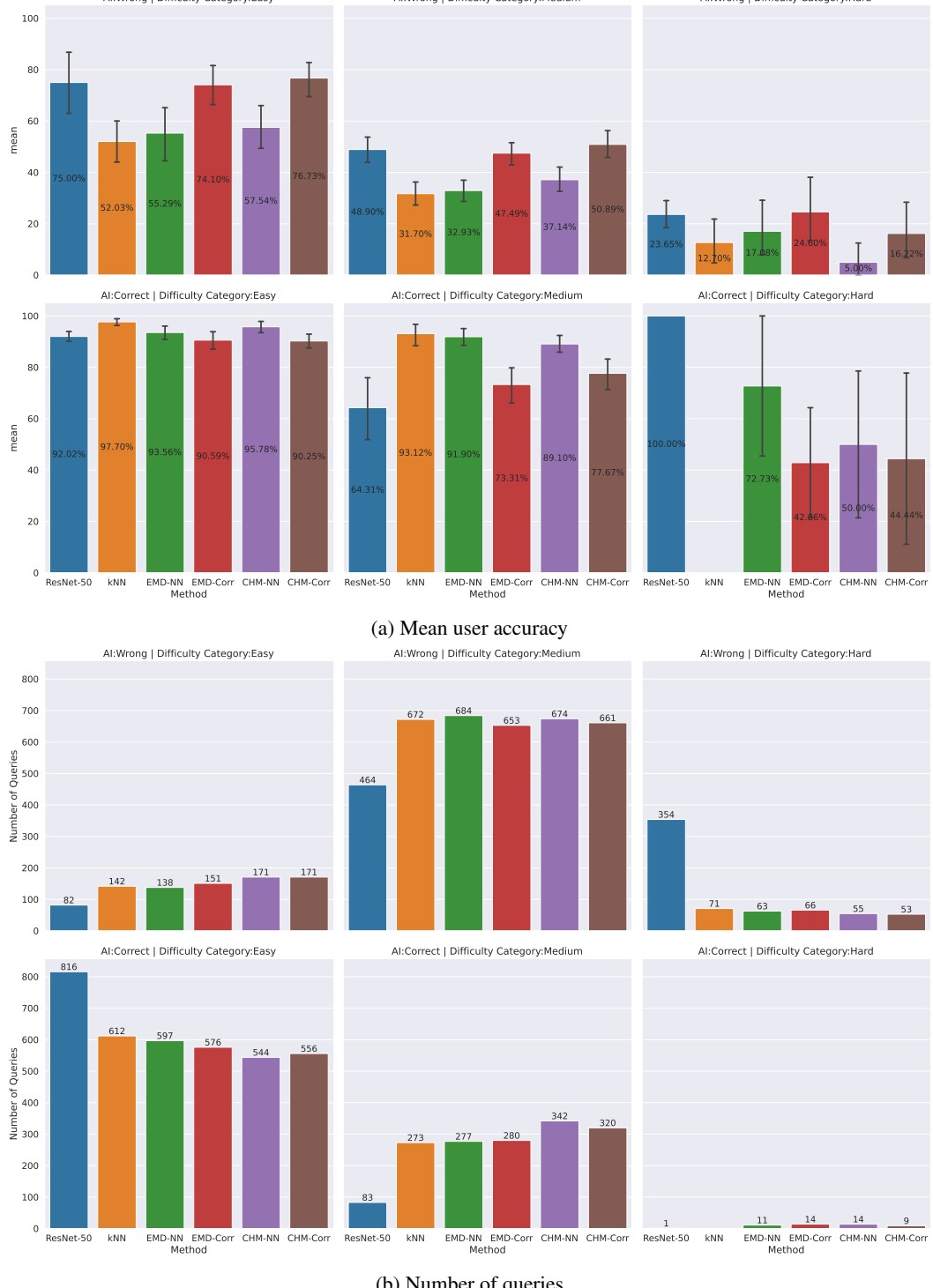

(a) Mean user accuracy

(b) Number of queries

Figure A13: CUB – The breakdown of the human performance by 'Difficulty Level' and 'AI Correctness'

## F.6 Analysis of Hard images for humans in the ImageNet task

This section provides an analysis to understand what kinds of queries are hard for humans, i.e., for what types of images users cannot correctly accept or reject the AI's decision. To this end, we filter the queries with a mean user's accuracy of $0.25$ or below. Figure A14 shows the distribution of Hard images for humans based on the classifier and the classifier's correctness.

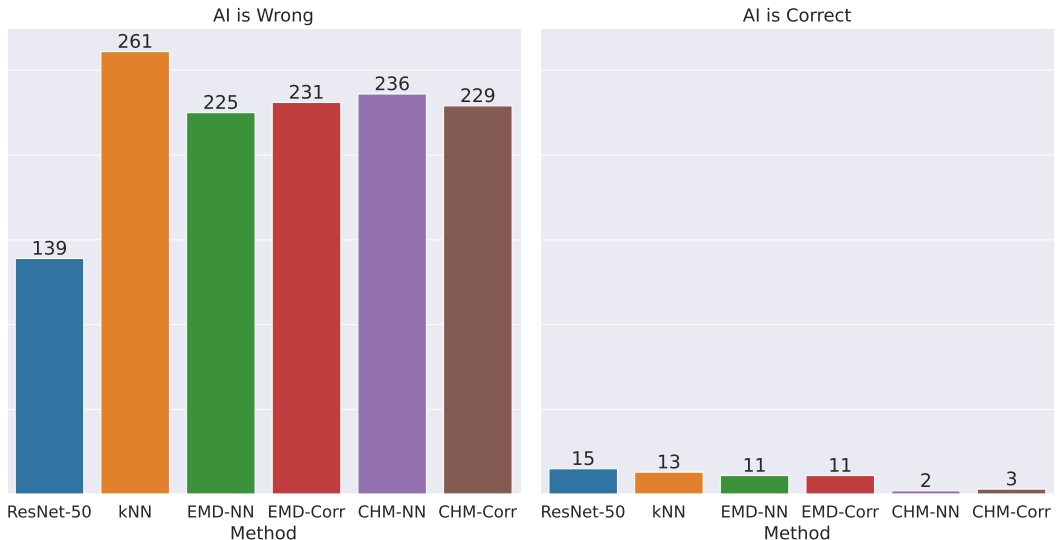

Figure A14: Number of confusing queries per classifier

To better understand what types of images are more challenging for humans, we automatically create supergroups for ImageNet class members. All 1,000 classes of ImageNet are a subgroup of class `entity - n00001740` in the WordNet glossary [50]. To create groups with uniform sizes, we start from the `entity` root node and break up each class into its sub-classes recursively; in each iteration, we pick the supergroup with the largest number of classes. Here we report the parent class of queries after 12 iterations and for queries with only one ImageNet-ReaL label. Using this automated procedure, we can see that the majority of hard images for humans fall into the `carnivore` category, which is a supergroup for cats, lions, dogs, wolves, etc. Details about each parent class and its ImageNet class members can be found in Table A8.

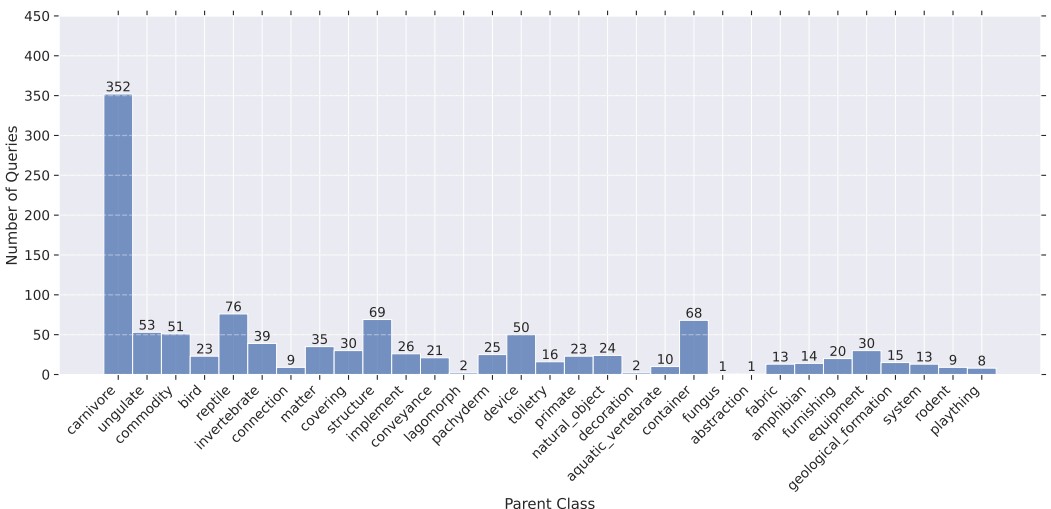

Figure A15: Parent class of confusing queries

Table A8: Parent classes of Hard queries for humans and their ImageNet class members.

| Parent Class | ImageNet Class Members |
| --- | --- |
| Abstraction | Street Sign |
| Amphibian | Axolotl, European Fire Salamander, Tree Frog |
| Aquatic Vertebrate | Goldfish |
| Bird | American Egret, Bulbul, Cock, Oystercatcher, Red-Backed Sandpiper, Ruffed Grouse |
| Carnivore | Beagle, Black-And-Tan Coonhound, Black-Footed Ferret, Bloodhound, Bluetick, Border Terrier, Bouvier Des Flandres, Bull Mastiff, Collie, Coyote, Curly-Coated Retriever, English Foxhound, English Springer, Entlebucher, Eskimo Dog, Flat-Coated Retriever, French Bulldog, German Short-Haired Pointer, Great Dane, Great Pyrenees, Greater Swiss Mountain Dog, Irish Water Spaniel, Kelpie, Lakeland Terrier, Lhasa, Malamute, Miniature Poodle, Norfolk Terrier, Otter, Pembroke, Polecat, Redbone, Rhodesian Ridgeback, Rottweiler, Schipperke, Scotch Terrier, Scottish Deerhound, Sealyham Terrier, Shetland Sheepdog, Siberian Husky, Staffordshire Bullterrier, Standard Poodle, Standard Schnauzer, Tabby, Tibetan Terrier, Tiger Cat, Toy Poodle, Vizsla, Welsh Springer Spaniel, Yorkshire Terrier |
| Commodity | Abaya, Academic Gown, Dishwasher, Dutch Oven, Espresso Maker, Microwave, Military Uniform, Miniskirt, Washer |
| Connection | Chain |
| Container | Ambulance, Cassette, Envelope, Pitcher, Purse, Shopping Basket, Soap Dispenser, Soup Bowl, Tank, Wallet, Washbasin, Whiskey Jug |
| Conveyance | Dogsled, Schooner, Stretcher, Trolleybus, Yawl |
| Covering | Dome, Doormat, Pickelhaube, Prayer Rug, Scabbard, Shower Curtain |
| Decoration | Necklace |
| Device | Analog Clock, Car Wheel, Cello, Combination Lock, Padlock, Projectile, Radiator, Upright, Wall Clock |
| Equipment | Balance Beam, Cd Player, Dumbbell, Horizontal Bar, Monitor, Polaroid Camera |
| Fabric | Wool |
| Fungus | Hen-Of-The-Woods |
| Furnishing | Cradle, Crib, Desk, Entertainment Center |
| Geological Formation | Coral Reef, Lakeside, Seashore |
| Implement | Ballpoint, Plow, Plunger, Quill, Teapot |
| Invertebrate | Barn Spider, Bee, Cricket, Damselfly, Dragonfly, Dungeness Crab, Long-Horned Beetle, Mantis, Sea Slug, Snail, Sulphur Butterfly |
| Lagomorph | Wood Rabbit |
| Matter | Artichoke, Bell Pepper, Cheeseburger, Cucumber, Hay, Plate, Pretzel |
| Natural Object | Banana, Corn, Lemon, Sandbar |
| Pachyderm | African Elephant, Indian Elephant |
| Plaything | Teddy |
| Primate | Gibbon, Gorilla, Langur, Siamang, Titi |
| Reptile | African Crocodile, Alligator Lizard, American Chameleon, Banded Gecko, Boa Constrictor, Frilled Lizard, Green Mamba, Green Snake, Mud Turtle, Night Snake, Rock Python, Terrapin |
| Rodent | Beaver |
| Structure | Bakery, Bannister, Church, Cliff Dwelling, Dam, Dock, Grocery Store, Megalith, Plate Rack, Stupa, Totem Pole |
| System | Radio |
| Toiletry | Hair Spray, Lotion, Sunscreen |
| Ungulate | Bighorn, Bison, Hog, Impala, Llama, Water Buffalo, Wild Boar |

### F.7 Analysis of Hard images for humans in the bird classification task

Similar to the analysis we conducted for ImageNet in Sec.F.6, here we analyze the confusing bird types for humans. We filter queries with a mean user accuracy of less than 0.25. Figure A16 shows the distribution of the most challenging samples for humans based on different classifiers' correctness.

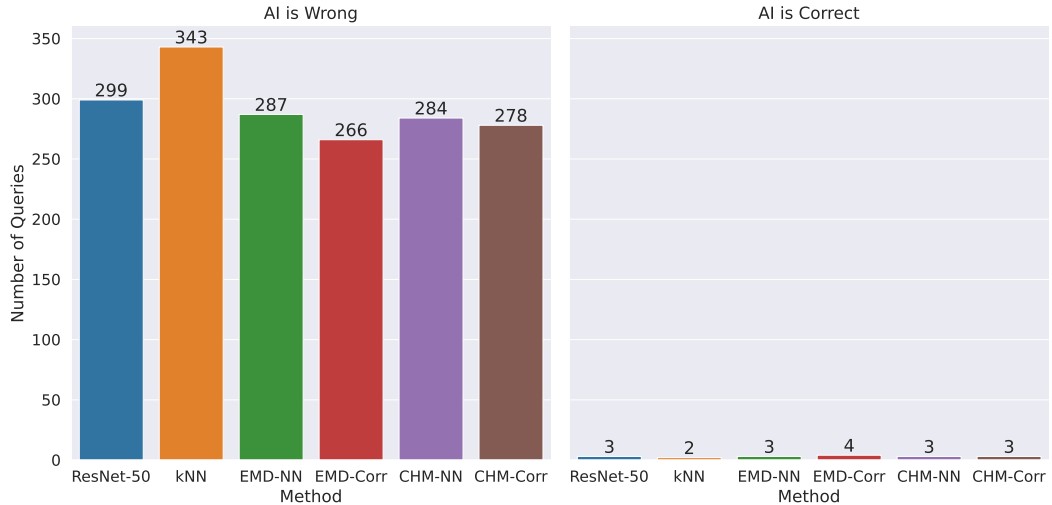

Figure A16: Number of confusing queries per classifier

Table A9 shows the top 5 confusing bird types for human users. Each row in this table shows how many users failed to reject AI's prediction when providing different kinds of explanations.

Table A9: Top-5 confusing bird types for humans per classifier

| Classifier | Ground Truth | Confused with | Users |
|---|---|---|---|
| ResNet-50 | Forsters Tern | Common Tern | 17 |
| | Great Grey Shrike | Loggerhead Shrike | 10 |
| | Nelson Sharp Tailed Sparrow | Le Conte Sparrow | 8 |
| | Acadian Flycatcher | Least Flycatcher | 7 |
| | American Crow | Common Raven | 7 |
| kNN | California Gull | Western Gull | 21 |
| | Elegant Tern | Caspian Tern | 13 |
| | Fish Crow | American Crow | 12 |
| | Rusty Blackbird | Brewer Blackbird | 11 |
| | Acadian Flycatcher | Yellow Bellied Flycatcher | 10 |
| EMD-NN | California Gull | Western Gull | 15 |
| | Common Tern | Artic Tern | 12 |
| | Nelson Sharp Tailed Sparrow | Savannah Sparrow | 8 |
| | Acadian Flycatcher | Yellow Bellied Flycatcher | 8 |
| | Yellow Bellied Flycatcher | Acadian Flycatcher | 8 |
| EMD-Corr | California Gull | Western Gull | 15 |
| | Common Tern | Artic Tern | 12 |
| | Nelson Sharp Tailed Sparrow | Savannah Sparrow | 8 |
| | Acadian Flycatcher | Yellow Bellied Flycatcher | 8 |
| | Yellow Bellied Flycatcher | Acadian Flycatcher | 8 |
| CHM-NN | Great Grey Shrike | Loggerhead Shrike | 19 |
| | Le Conte Sparrow | Nelson Sharp Tailed Sparrow | 15 |
| | California Gull | Western Gull | 15 |
| | Louisiana Waterthrush | Northern Waterthrush | 12 |
| | Horned Grebe | Eared Grebe | 9 |
| CHM-Corr | Great Grey Shrike | Loggerhead Shrike | 20 |
| | Horned Grebe | Eared Grebe | 15 |
| | California Gull | Western Gull | 15 |
| | Louisiana Waterthrush | Northern Waterthrush | 13 |
| | Le Conte Sparrow | Nelson Sharp Tailed Sparrow | 13 |

## F.8 Accepting AI's wrong decision

This section shows samples for which users incorrectly accepted the incorrect AI prediction.

### F.8.1 Accepting the wrong kNN Classifier's prediction

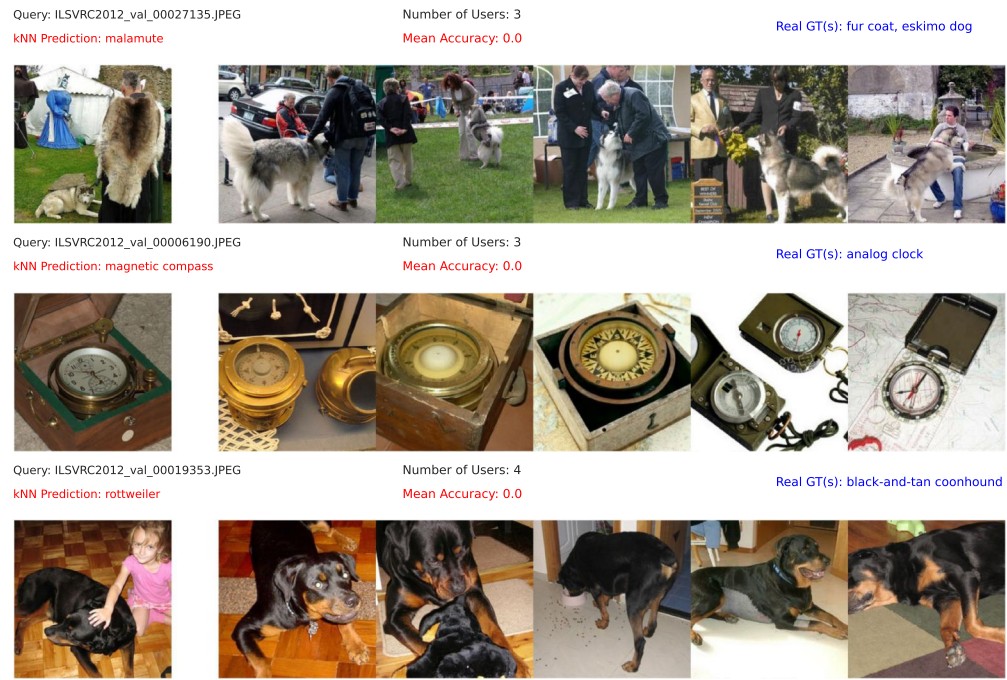

Figure A17: Accepting the wrong kNN prediction due to confusing explanations

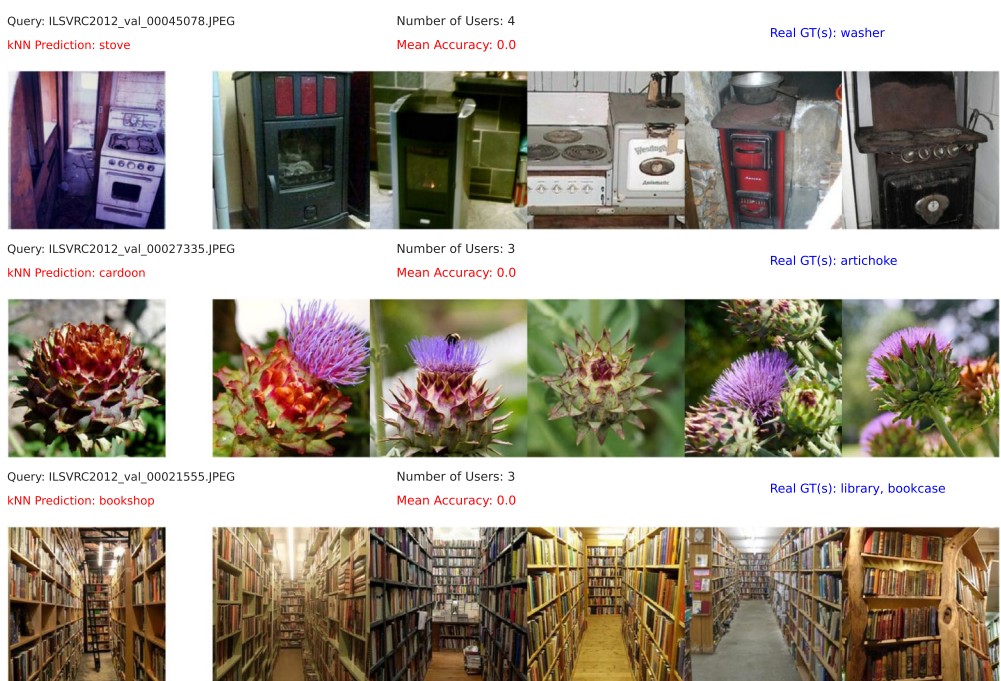

Figure A18: Accepting the wrong kNN prediction due to poor ImageNet-ReaL labeling

## F.8.2 Accepting the wrong EMD-NN Classifier's prediction

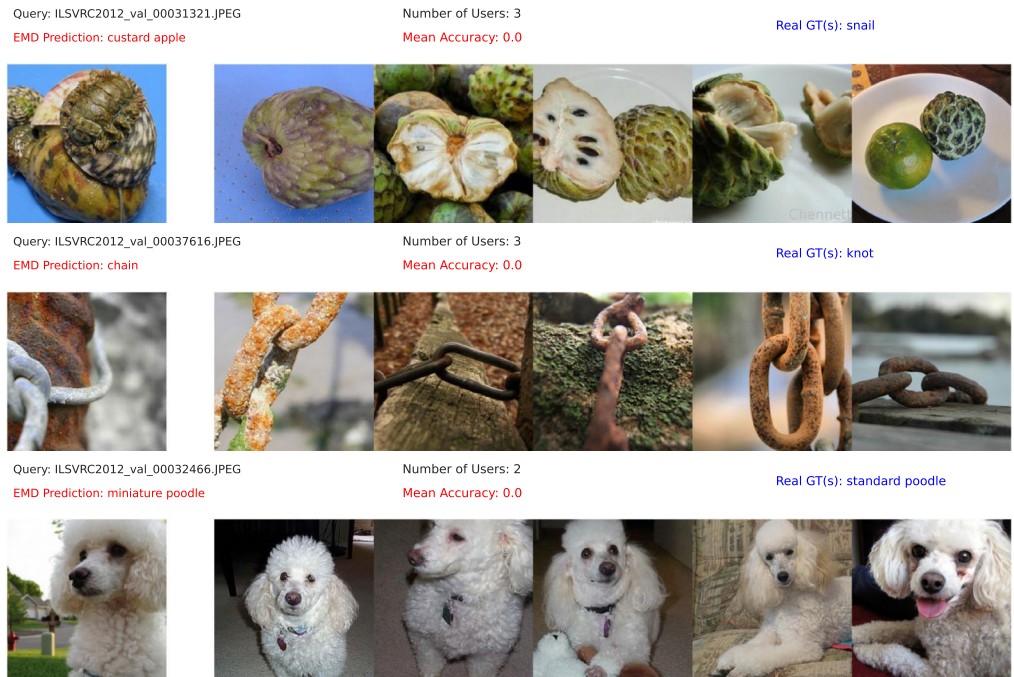

Figure A19: Accepting the wrong EMD-NN prediction due to confusing explanations

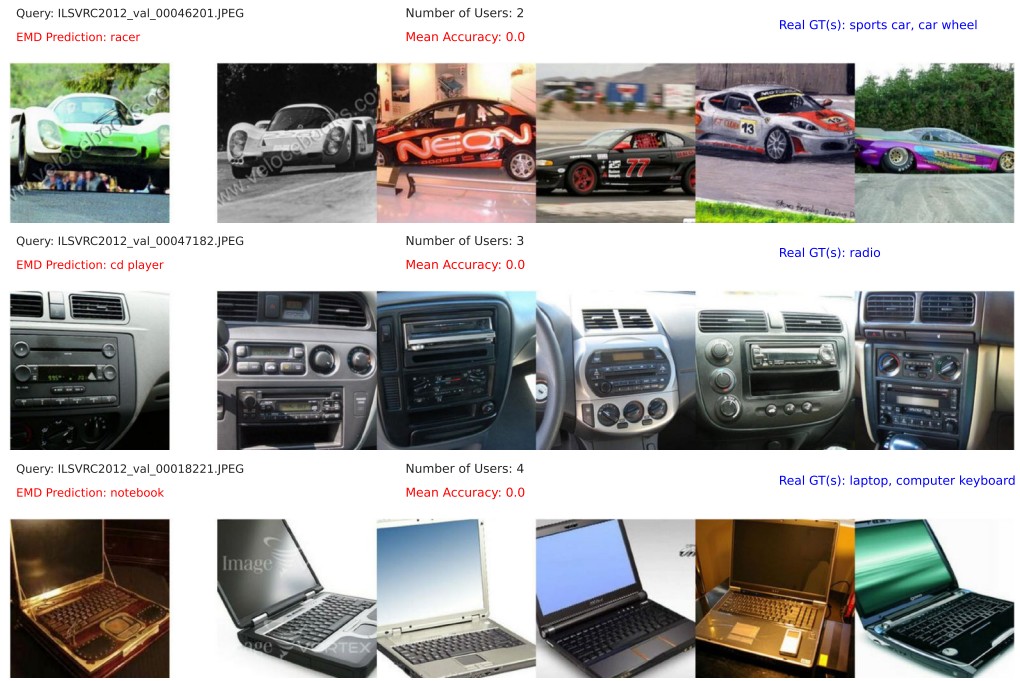

Figure A20: Accepting wrong EMD-NN prediction due to 'Bad Labels'

### F.8.3 Accepting the wrong EMD-Corr Classifier's prediction

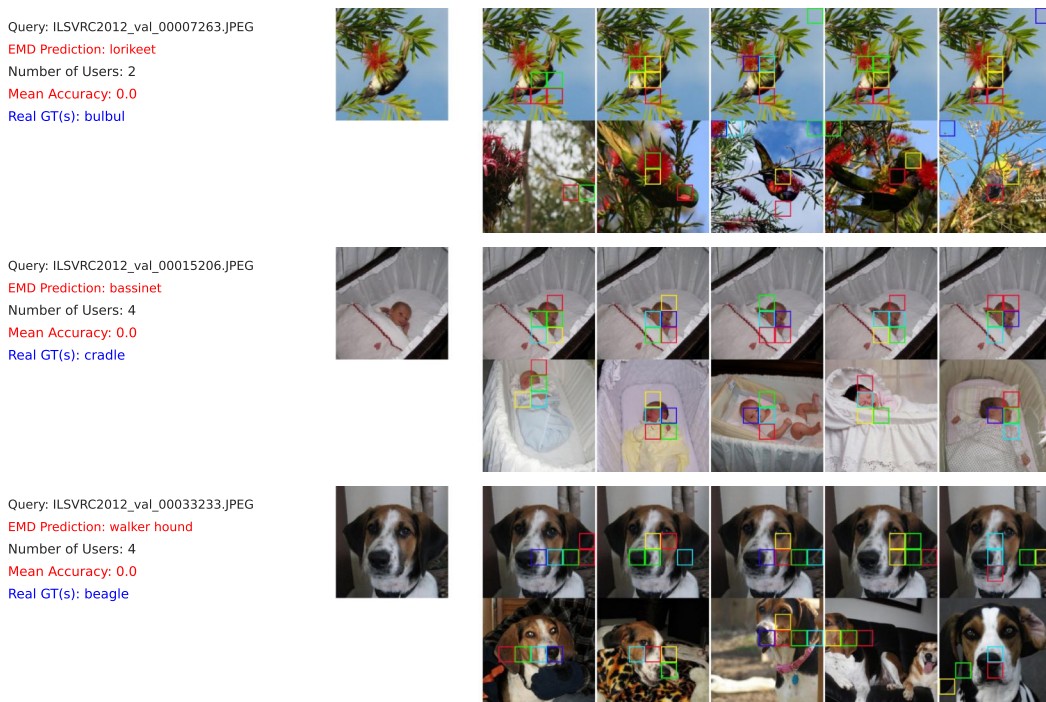

Figure A21: Accepting wrong EMD-Corr prediction due to confusing explanations

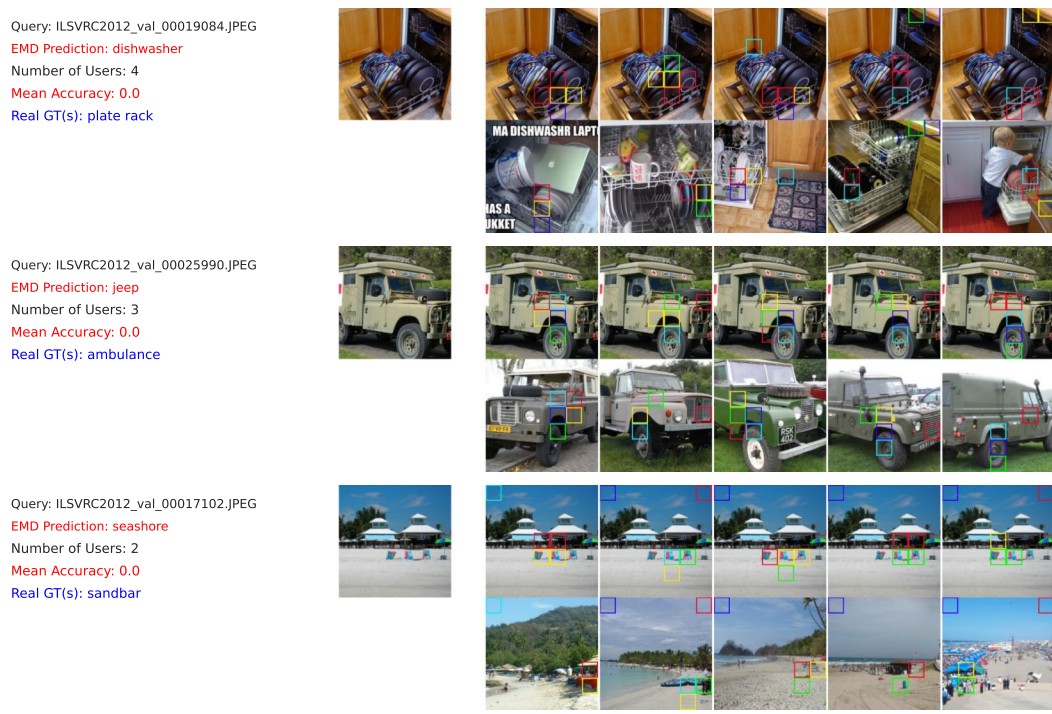

Figure A22: Accepting wrong EMD-Corr prediction due to poor ImageNet-ReaL labeling

## F.8.4 Accepting the wrong CHM-NN Classifier's prediction

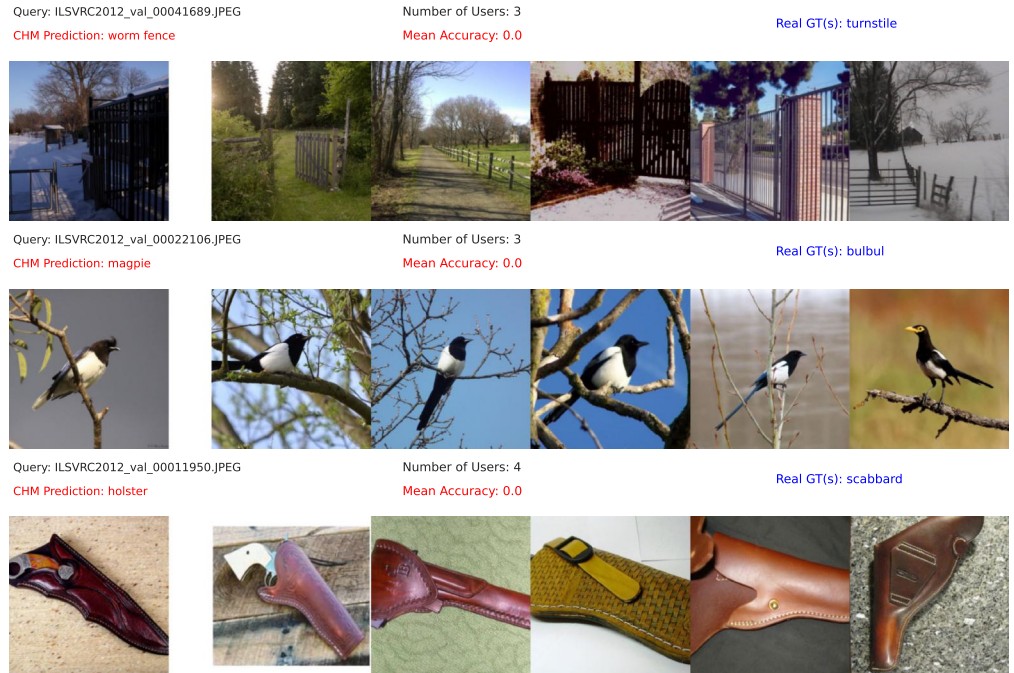

Figure A23: Accepting wrong CHM-NN prediction due to confusing explanations

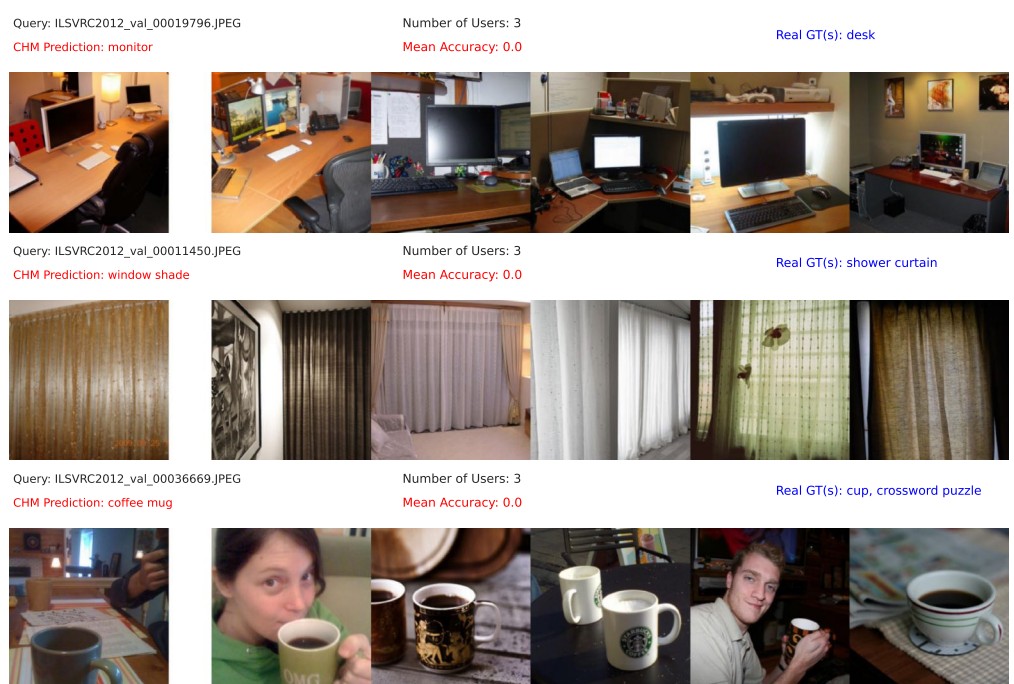

Figure A24: Accepting wrong CHM-NN prediction due to poor ImageNet-ReaL labeling

### F.8.5 Accepting the wrong CHM-Corr Classifier's prediction

Query: ILSVRC2012_val_00002666.JPEG
CHM Prediction: oil filter
Number of Users: 3
Mean Accuracy: 0.0
Real GT(s): saltshaker

Query: ILSVRC2012_val_00026198.JPEG
CHM Prediction: barbell
Number of Users: 3
Mean Accuracy: 0.0
Real GT(s): dumbbell

Query: ILSVRC2012_val_00025908.JPEG
CHM Prediction: thatch
Number of Users: 3
Mean Accuracy: 0.0
Real GT(s): hay

Figure A25: Accepting wrong CHM-Corr prediction due to confusing explanations

Query: ILSVRC2012_val_00021601.JPEG
CHM Prediction: home theater
Number of Users: 3
Mean Accuracy: 0.0
Real GT(s): entertainment center, monitor, desk, television

Query: ILSVRC2012_val_00018863.JPEG
CHM Prediction: combination lock
Number of Users: 3
Mean Accuracy: 0.0
Real GT(s): space bar, computer keyboard

Query: ILSVRC2012_val_00005467.JPEG
CHM Prediction: stage
Number of Users: 3
Mean Accuracy: 0.0
Real GT(s): electric guitar, microphone, acoustic guitar

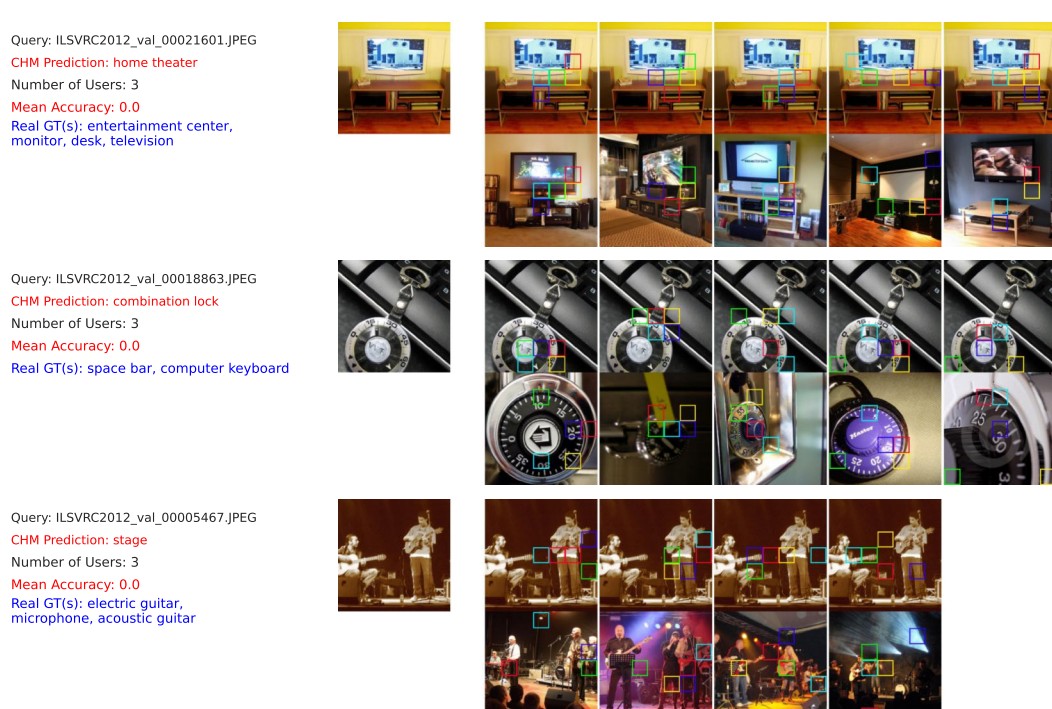

Figure A26: Accepting wrong CHM-Corr prediction due to poor ImageNet-ReaL labeling

## F.9 When explanations fool users

This section provides clear evidence that explanations have the potential to fool human users. Both ResNet-50 and EMD-Corr misclassified an image of `tow truck` into `cab`. When asking a user to accept or reject this particular misclassification, they act differently based on the provided explanation. A total of 6 users who saw the query without any visual explanation were able to correctly reject AI's decision, while 3 out of 6 (50%) users who received either an EMD-NN or EMD-Corr explanation incorrectly accepted the decision.

Figure A27: Samples for human users failing to reject wrong AI decisions—a tow truck misclassified as a `cab` by EMD-Corr classifier.

# G Classification accuracy of Human-AI teams

In this section, we provide a detailed breakdown of human-AI team accuracy at different confidence thresholds.

We divide the set of images into two groups for each confidence threshold $T$: (1) images in which the AI's confidence equals or exceeds $T$, and (2) images in which the AI's confidence is less than $T$. In the first group, we only consider AI's decision, while for the second group, we ask a human user to judge AI's predicted label, i.e., whether users accept or reject AI's classification. The aggregate accuracy of the human-AI team is the weighted average of the accuracy obtained from both groups. To determine the best threshold, we first determine the value of $T$ that results in the best AI-alone accuracy on a small subset of the ImageNet-ReaL (2K images) and CUB (1K images) datasets, and then we evaluate the AI-alone accuracy on the held-out set for each dataset (42K images on ImageNet-ReaL and 4K on CUB).

Table A10: ResNet-50 - Aggregating Human and AI (%) – Bold numbers represent human-AI team performance at the optimal threshold

| | ImageNet | | | | CUB | | | |
|---|---|---|---|---|---|---|---|---|
| $T$ | % of images handled by AI | AI-alone accuracy (confidence $>= T$) | human accuracy (confidence $< T$) | Aggregated human-AI accuracy | % of images handled by AI | AI-alone accuracy (confidence $>= T$) | human accuracy (confidence $< T$) | Aggregated human-AI accuracy |
| 0.00 | 100.00 | 83.14 | n/a | n/a | 100.00 | 85.83 | n/a | n/a |
| 0.05 | 99.98 | 83.16 | 100.00 | 83.16 | 100.00 | 85.83 | n/a | n/a |
| 0.10 | 99.71 | 83.34 | 100.00 | 83.39 | 100.00 | 85.83 | n/a | n/a |
| 0.15 | 98.74 | 83.96 | 89.09 | 84.03 | 99.91 | 85.87 | 100.00 | 85.88 |
| 0.20 | 97.86 | 84.48 | 85.98 | 84.51 | 99.71 | 86.01 | 76.47 | 85.99 |
| 0.25 | 96.43 | 85.29 | 89.82 | 85.45 | 99.40 | 86.18 | 79.49 | 86.14 |
| 0.30 | 94.39 | 86.35 | 92.41 | 86.69 | 98.88 | 86.47 | 83.93 | 86.44 |
| 0.35 | 92.47 | 87.32 | 89.14 | 87.46 | 98.19 | 86.89 | 76.40 | 86.70 |
| 0.40 | 90.84 | 88.13 | 86.73 | 88.00 | 97.45 | 87.32 | 72.17 | 86.93 |
| 0.45 | 88.50 | 89.15 | 84.62 | **88.63** | 96.01 | 87.99 | 69.36 | 87.25 |
| 0.50 | 85.88 | 90.20 | 83.79 | 89.29 | 94.32 | 88.82 | 65.38 | 87.49 |
| 0.55 | 82.65 | 91.35 | 81.52 | 89.64 | 92.16 | 89.85 | 59.27 | **87.45** |
| 0.60 | 78.96 | 92.59 | 80.80 | 90.11 | 89.47 | 90.91 | 60.78 | 87.74 |
| 0.65 | 76.57 | 93.36 | 80.50 | 90.35 | 87.68 | 91.81 | 57.23 | 87.55 |
| 0.70 | 72.85 | 94.50 | 77.83 | 89.98 | 84.69 | 92.81 | 54.56 | 86.95 |
| 0.75 | 70.17 | 95.24 | 76.06 | 89.52 | 82.52 | 93.45 | 54.60 | 86.66 |
| 0.80 | 66.77 | 96.04 | 76.10 | 89.41 | 79.41 | 94.48 | 52.55 | 85.85 |
| 0.85 | 61.89 | 96.99 | 75.65 | 88.86 | 75.51 | 95.52 | 51.72 | 84.79 |
| 0.90 | 57.63 | 97.65 | 75.63 | 88.32 | 71.88 | 96.37 | 51.91 | 83.87 |
| 0.95 | 47.42 | 98.67 | 76.08 | 86.79 | 61.08 | 97.68 | 54.55 | 80.89 |
| 1.00 | 0.47 | 100.00 | 81.52 | 81.61 | 0.00 | n/a | 65.50 | n/a |

Table A11: kNN - Aggregating Human and AI (%) – Bold numbers represent human-AI team performance at the optimal threshold

| | ImageNet | | | | CUB | | | |
|---|---|---|---|---|---|---|---|---|
| $T$ | % of images handled by AI | AI-alone accuracy (confidence >= $T$) | human accuracy (confidence < $T$) | Aggregated human-AI accuracy | % of images handled by AI | AI-alone accuracy (confidence >= $T$) | human accuracy (confidence < $T$) | Aggregated human-AI accuracy |
| 0.00 | 100.00 | 82.14 | n/a | n/a | 100.00 | 85.47 | n/a | n/a |
| 0.05 | 100.00 | 82.14 | n/a | n/a | 100.00 | 85.47 | n/a | n/a |
| 0.10 | 99.99 | 82.16 | 100.00 | 82.16 | 100.00 | 85.47 | n/a | n/a |
| 0.15 | 98.26 | 83.34 | 97.14 | 83.58 | 99.86 | 85.59 | 100.00 | 85.61 |
| 0.20 | 98.26 | 83.34 | 97.14 | 83.58 | 99.86 | 85.59 | 100.00 | 85.61 |
| 0.25 | 96.52 | 84.36 | 90.23 | 84.57 | 99.62 | 85.76 | 68.18 | 85.69 |
| 0.30 | 91.89 | 86.85 | 80.06 | 86.30 | 97.20 | 87.18 | 50.70 | 86.16 |
| 0.35 | 89.25 | 88.10 | 77.86 | 87.00 | 94.55 | 88.68 | 47.83 | 86.45 |
| 0.40 | 89.25 | 88.10 | 77.86 | 87.00 | 94.55 | 88.68 | 47.83 | 86.45 |
| 0.45 | 86.34 | 89.44 | 73.40 | **87.24** | 91.15 | 90.13 | 50.85 | **86.66** |
| 0.50 | 83.25 | 90.67 | 70.78 | 87.34 | 86.73 | 92.02 | 50.70 | 86.54 |
| 0.55 | 79.74 | 91.91 | 67.99 | 87.06 | 81.01 | 93.97 | 47.31 | 85.11 |
| 0.60 | 72.51 | 94.24 | 67.87 | 86.99 | 71.52 | 96.67 | 47.39 | 82.63 |
| 0.65 | 72.51 | 94.24 | 67.87 | 86.99 | 71.52 | 96.67 | 47.39 | 82.63 |
| 0.70 | 65.44 | 96.15 | 67.43 | 86.23 | 62.15 | 97.81 | 49.68 | 79.59 |
| 0.75 | 65.44 | 96.15 | 67.43 | 86.23 | 62.15 | 97.81 | 49.68 | 79.59 |
| 0.80 | 61.82 | 96.91 | 66.50 | 85.30 | 56.87 | 98.12 | 51.14 | 77.86 |
| 0.85 | 53.34 | 98.10 | 66.93 | 83.55 | 45.50 | 99.01 | 52.95 | 73.91 |
| 0.90 | 53.34 | 98.10 | 66.93 | 83.55 | 45.50 | 99.01 | 52.95 | 73.91 |
| 0.95 | 36.77 | 99.19 | 70.42 | 81.00 | 28.58 | 99.28 | 58.60 | 70.23 |
| 1.00 | 36.77 | 99.19 | 70.42 | 81.00 | 28.58 | 99.28 | 58.60 | 70.23 |

Table A12: EMD-NN Aggregating Human and AI (%) – Bold numbers represent human-AI team performance at the optimal threshold

| | ImageNet | | | | CUB | | | |
|---|---|---|---|---|---|---|---|---|
| $T$ | % of images handled by AI | AI-alone accuracy (confidence >= $T$) | human accuracy (confidence < $T$) | Aggregated human-AI accuracy | % of images handled by AI | AI-alone accuracy (confidence >= $T$) | human accuracy (confidence < $T$) | Aggregated human-AI accuracy |
| 0.00 | 100.00 | 82.39 | n/a | n/a | 100.00 | 84.98 | n/a | NaN |
| 0.05 | 100.00 | 82.39 | n/a | n/a | 100.00 | 84.98 | NaN | NaN |
| 0.10 | 99.99 | 82.40 | 100.00 | 82.40 | 100.00 | 84.98 | NaN | NaN |
| 0.15 | 98.19 | 83.63 | 96.10 | 83.86 | 99.81 | 85.15 | 60.00 | 85.10 |
| 0.20 | 98.19 | 83.63 | 96.10 | 83.86 | 99.81 | 85.15 | 60.00 | 85.10 |
| 0.25 | 96.36 | 84.72 | 95.24 | 85.10 | 99.50 | 85.34 | 68.75 | 85.26 |
| 0.30 | 91.86 | 87.12 | 88.36 | 87.22 | 96.50 | 87.03 | 55.70 | 85.94 |
| 0.35 | 89.14 | 88.37 | 83.11 | 87.80 | 93.68 | 88.39 | 49.21 | 85.92 |
| 0.40 | 89.14 | 88.37 | 83.11 | **87.80** | 93.68 | 88.39 | 49.21 | 85.92 |
| 0.45 | 86.25 | 89.59 | 81.49 | 88.47 | 89.40 | 90.19 | 47.67 | **85.69** |
| 0.50 | 83.19 | 90.80 | 77.45 | 88.56 | 83.98 | 92.40 | 48.92 | 85.43 |
| 0.55 | 79.56 | 92.13 | 73.59 | 88.34 | 78.29 | 94.47 | 48.74 | 84.54 |
| 0.60 | 72.11 | 94.53 | 70.41 | 87.80 | 68.47 | 96.45 | 49.35 | 81.60 |
| 0.65 | 72.11 | 94.53 | 70.41 | 87.80 | 68.47 | 96.45 | 49.35 | 81.60 |
| 0.70 | 65.02 | 96.16 | 68.68 | 86.55 | 57.96 | 97.89 | 50.90 | 78.13 |
| 0.75 | 65.02 | 96.16 | 68.68 | 86.55 | 57.96 | 97.89 | 50.90 | 78.13 |
| 0.80 | 61.30 | 96.94 | 68.46 | 85.92 | 51.90 | 98.17 | 52.96 | 76.42 |
| 0.85 | 52.32 | 98.18 | 69.70 | 84.60 | 39.68 | 99.30 | 55.29 | 72.76 |
| 0.90 | 52.32 | 98.18 | 69.70 | 84.60 | 39.68 | 99.30 | 55.29 | 72.76 |
| 0.95 | 35.24 | 99.18 | 73.15 | 82.33 | 19.57 | 99.38 | 60.80 | 68.35 |
| 1.00 | 35.24 | 99.18 | 73.15 | 82.33 | 19.57 | 99.38 | 60.80 | 68.35 |

Table A13: EMD-Corr Aggregating Human and AI (%) – Bold numbers represent human-AI team performance at the optimal threshold

| | ImageNet | | | | CUB | | | |
| $T$ | % of images handled by AI | AI-alone accuracy (confidence $>= T$) | human accuracy (confidence $< T$) | Aggregated human-AI accuracy | % of images handled by AI | AI-alone accuracy (confidence $>= T$) | human accuracy (confidence $< T$) | Aggregated human-AI accuracy |
|---|---|---|---|---|---|---|---|---|
| 0.00 | 100.00 | 82.39 | n/a | n/a | 100.00 | 84.98 | n/a | n/a |
| 0.05 | 100.00 | 82.39 | n/a | n/a | 100.00 | 84.98 | n/a | n/a |
| 0.10 | 99.99 | 82.40 | 100.00 | 82.40 | 100.00 | 84.98 | n/a | n/a |
| 0.15 | 98.19 | 83.63 | 95.29 | 83.84 | 99.81 | 85.15 | 100.00 | 85.17 |
| 0.20 | 98.19 | 83.63 | 95.29 | 83.84 | 99.81 | 85.15 | 100.00 | 85.17 |
| 0.25 | 96.36 | 84.72 | 95.57 | 85.11 | 99.50 | 85.34 | 86.67 | 85.35 |
| 0.30 | 91.86 | 87.12 | 89.27 | 87.29 | 96.50 | 87.03 | 69.70 | 86.43 |
| 0.35 | 89.14 | 88.37 | 85.19 | 88.02 | 93.68 | 88.39 | 60.36 | 86.62 |
| 0.40 | 89.14 | 88.37 | 85.19 | **88.02** | 93.68 | 88.39 | 60.36 | 86.62 |
| 0.45 | 86.25 | 89.59 | 82.59 | 88.62 | 89.40 | 90.19 | 58.70 | **86.86** |
| 0.50 | 83.19 | 90.80 | 79.17 | 88.85 | 83.98 | 92.40 | 57.24 | 86.77 |
| 0.55 | 79.56 | 92.13 | 74.67 | 88.56 | 78.29 | 94.47 | 57.26 | 86.39 |
| 0.60 | 72.11 | 94.53 | 72.40 | 88.36 | 68.47 | 96.45 | 58.34 | 84.43 |
| 0.65 | 72.11 | 94.53 | 72.40 | 88.36 | 68.47 | 96.45 | 58.34 | 84.43 |
| 0.70 | 65.02 | 96.16 | 70.44 | 87.16 | 57.96 | 97.89 | 58.20 | 81.20 |
| 0.75 | 65.02 | 96.16 | 70.44 | 87.16 | 57.96 | 97.89 | 58.20 | 81.20 |
| 0.80 | 61.30 | 96.94 | 70.71 | 86.79 | 51.90 | 98.17 | 59.04 | 79.35 |
| 0.85 | 52.32 | 98.18 | 71.74 | 85.57 | 39.68 | 99.30 | 60.70 | 76.01 |
| 0.90 | 52.32 | 98.18 | 71.74 | 85.57 | 39.68 | 99.30 | 60.70 | 76.01 |
| 0.95 | 35.24 | 99.18 | 74.63 | 83.28 | 19.57 | 99.38 | 64.53 | 71.35 |
| 1.00 | 35.24 | 99.18 | 74.63 | 83.28 | 19.57 | 99.38 | 64.53 | 71.35 |

Table A14: CHM-NN Aggregating Human and AI (%)

| | ImageNet | | | | CUB | | | |
| $T$ | % of images handled by AI | AI-alone accuracy (confidence $>= T$) | human accuracy (confidence $< T$) | Aggregated human-AI accuracy | % of images handled by AI | AI-alone accuracy (confidence $>= T$) | human accuracy (confidence $< T$) | Aggregated human-AI accuracy |
|---|---|---|---|---|---|---|---|---|
| 0.00 | 100.00 | 82.05 | n/a | n/a | 100.00 | 83.28 | n/a | n/a |
| 0.05 | 100.00 | 82.05 | n/a | n/a | 100.00 | 83.28 | n/a | n/a |
| 0.10 | 99.99 | 82.06 | n/a | n/a | 100.00 | 83.28 | n/a | n/a |
| 0.15 | 98.53 | 83.03 | 94.74 | 83.20 | 99.86 | 83.36 | 80.00 | 83.35 |
| 0.20 | 98.53 | 83.03 | 94.74 | 83.20 | 99.86 | 83.36 | 80.00 | 83.35 |
| 0.25 | 96.95 | 83.96 | 88.97 | 84.11 | 99.52 | 83.56 | 61.54 | 83.45 |
| 0.30 | 92.86 | 86.17 | 81.37 | 85.82 | 95.79 | 85.64 | 55.14 | 84.36 |
| 0.35 | 90.35 | 87.40 | 80.58 | 86.75 | 92.18 | 87.25 | 54.92 | 84.72 |
| 0.40 | 90.35 | 87.40 | 80.58 | 86.75 | 92.18 | 87.25 | 54.92 | 84.72 |
| 0.45 | 87.60 | 88.65 | 79.22 | 87.48 | 87.04 | 89.63 | 53.19 | **84.91** |
| 0.50 | 84.55 | 89.90 | 77.88 | **88.05** | 81.12 | 91.81 | 51.94 | 84.28 |
| 0.55 | 80.85 | 91.27 | 73.72 | 87.91 | 74.28 | 94.17 | 50.63 | 82.97 |
| 0.60 | 73.48 | 93.72 | 69.48 | 87.29 | 60.87 | 97.22 | 52.05 | 79.55 |
| 0.65 | 73.48 | 93.72 | 69.48 | 87.29 | 60.87 | 97.22 | 52.05 | 79.55 |
| 0.70 | 66.57 | 95.65 | 68.99 | 86.74 | 48.43 | 98.43 | 54.79 | 75.92 |
| 0.75 | 66.57 | 95.65 | 68.99 | 86.74 | 48.43 | 98.43 | 54.79 | 75.92 |
| 0.80 | 63.00 | 96.34 | 69.23 | 86.31 | 41.37 | 98.71 | 57.00 | 74.25 |
| 0.85 | 54.32 | 97.77 | 68.79 | 84.53 | 25.51 | 99.26 | 60.18 | 70.14 |
| 0.90 | 54.32 | 97.77 | 68.79 | 84.53 | 25.51 | 99.26 | 60.18 | 70.14 |
| 0.95 | 37.96 | 98.97 | 71.49 | 81.92 | 9.48 | 99.64 | 64.32 | 67.66 |
| 1.00 | 37.96 | 98.97 | 71.49 | 81.92 | 9.48 | 99.64 | 64.32 | 67.66 |

Table A15: CHM-Corr Aggregating Human and AI (%) – Bold numbers represent human-AI team performance at the optimal threshold

| | ImageNet | | | | CUB | | | |
|---|---|---|---|---|---|---|---|---|
| $T$ | % of images handled by AI | AI-alone accuracy (confidence >= $T$) | human accuracy (confidence < $T$) | Aggregated human-AI accuracy | % of images handled by AI | AI-alone accuracy (confidence >= $T$) | human accuracy (confidence < $T$) | Aggregated human-AI accuracy |
| 0.00 | 100.00 | 82.05 | n/a | n/a | 100.00 | 83.28 | n/a | n/a |
| 0.05 | 100.00 | 82.05 | n/a | n/a | 100.00 | 83.28 | n/a | n/a |
| 0.10 | 99.99 | 82.06 | n/a | n/a | 100.00 | 83.28 | n/a | n/a |
| 0.15 | 98.53 | 83.03 | 91.89 | 83.16 | 99.86 | 83.36 | 100.00 | 83.38 |
| 0.20 | 98.53 | 83.03 | 91.89 | 83.16 | 99.86 | 83.36 | 100.00 | 83.38 |
| 0.25 | 96.95 | 83.96 | 86.99 | 84.05 | 99.52 | 83.56 | 53.85 | 83.42 |
| 0.30 | 92.86 | 86.17 | 81.90 | 85.86 | 95.79 | 85.64 | 72.22 | 85.07 |
| 0.35 | 90.35 | 87.40 | 78.35 | 86.53 | 92.18 | 87.25 | 71.06 | 85.98 |
| 0.40 | 90.35 | 87.40 | 78.35 | 86.53 | 92.18 | 87.25 | 71.06 | 85.98 |
| 0.45 | 87.60 | 88.65 | 77.39 | 87.26 | 87.04 | 89.63 | 63.58 | **86.25** |
| 0.50 | 84.55 | 89.90 | 76.85 | **87.89** | 81.12 | 91.81 | 62.92 | 86.35 |
| 0.55 | 80.85 | 91.27 | 73.35 | 87.84 | 74.28 | 94.17 | 61.15 | 85.68 |
| 0.60 | 73.48 | 93.72 | 70.30 | 87.51 | 60.87 | 97.22 | 60.52 | 82.86 |
| 0.65 | 73.48 | 93.72 | 70.30 | 87.51 | 60.87 | 97.22 | 60.52 | 82.86 |
| 0.70 | 66.57 | 95.65 | 70.21 | 87.15 | 48.43 | 98.43 | 62.36 | 79.83 |
| 0.75 | 66.57 | 95.65 | 70.21 | 87.15 | 48.43 | 98.43 | 62.36 | 79.83 |
| 0.80 | 63.00 | 96.34 | 70.56 | 86.80 | 41.37 | 98.71 | 63.37 | 77.99 |
| 0.85 | 54.32 | 97.77 | 69.32 | 84.78 | 25.51 | 99.26 | 65.20 | 73.89 |
| 0.90 | 54.32 | 97.77 | 69.32 | 84.78 | 25.51 | 99.26 | 65.20 | 73.89 |
| 0.95 | 37.96 | 98.97 | 71.30 | 81.80 | 9.48 | 99.64 | 68.29 | 71.26 |
| 1.00 | 37.96 | 98.97 | 71.30 | 81.80 | 9.48 | 99.64 | 68.29 | 71.26 |

### G.1 Human-AI team is better than AI-only

Because there is a subset of images for which AIs are not confident, and have very low accuracy (accuracy = $47/408$ at $T = 0.45$) (Table A28). Therefore, humans helped increase the accuracy by looking at this subset and rejecting AI's incorrect predictions. These images are easy for humans to reject (Figure A28).

Table A16: Breakdown of the number of trials at different thresholds – ResNet-50 – ImageNet

| T | Human Performance | # Trials | # Trials Correct AI Prediction | # Trials Wrong AI Prediction |
|---|---|---|---|---|
| 0.00 | n/a | n/a | 0 | 1 |
| 0.05 | 100.00 | 3 | 0 | 3 |
| 0.10 | 100.00 | 16 | 0 | 16 |
| 0.15 | 89.09 | 55 | 3 | 52 |
| 0.20 | 85.98 | 107 | 8 | 99 |
| 0.25 | 89.82 | 167 | 17 | 150 |
| 0.30 | 92.41 | 224 | 20 | 204 |
| 0.35 | 89.14 | 313 | 22 | 291 |
| 0.40 | 86.73 | 392 | 34 | 358 |
| **0.45** | **84.62** | **455** | **47** | **408** |
| 0.50 | 83.79 | 543 | 55 | 488 |
| 0.55 | 81.52 | 633 | 85 | 548 |
| 0.60 | 80.80 | 729 | 124 | 605 |
| 0.65 | 80.50 | 800 | 151 | 649 |
| 0.70 | 77.83 | 875 | 160 | 715 |
| 0.75 | 76.06 | 944 | 188 | 756 |
| 0.80 | 76.10 | 996 | 209 | 787 |
| 0.85 | 75.65 | 1072 | 258 | 814 |
| 0.90 | 75.63 | 1145 | 310 | 835 |
| 0.95 | 76.08 | 1292 | 413 | 879 |
| 1.00 | 81.52 | 1797 | 891 | 906 |

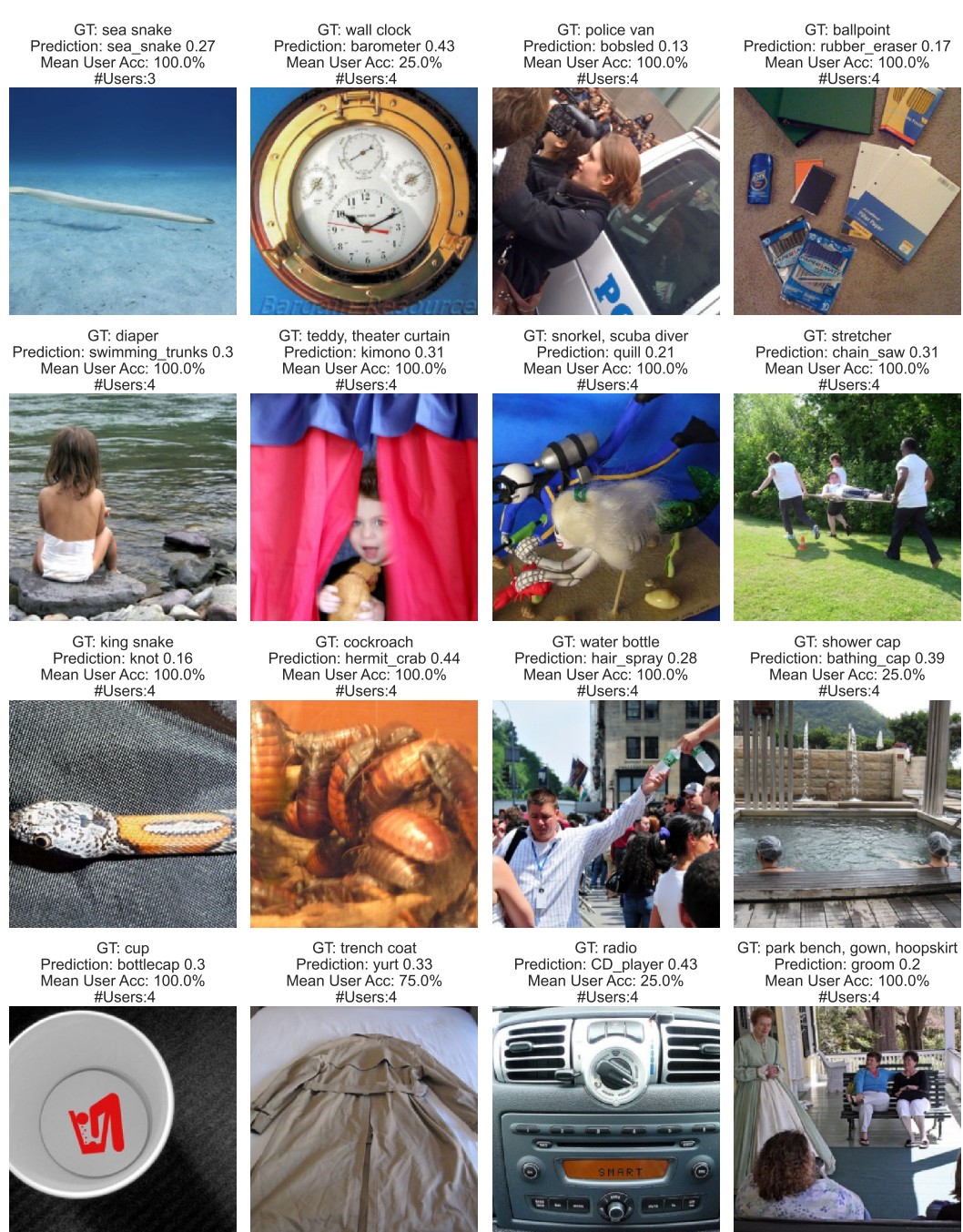

Figure A28: ImageNet samples at $T = 0.45$ – ResNet-50 classifier

## G.2 Human-AI team is better than human-only

Because humans are not trained explicitly to perform image classification on CUB and ImageNet-ReaL, the mean human-only accuracy is 65.50% and 81.56% respectively. When teaming up with AI, human-AI teams perform slightly better on ImageNet (81.56% vs. 86.80%) but substantially better on CUB (65.50% vs. 87.94%). See Table 3.

Table A17: Breakdown of the number of trials at different thresholds – ResNet-50 – CUB

| T | Human Performance | # Trials | # Trials Correct AI Prediction | # Trials Wrong AI Prediction |
|---|---|---|---|---|
| 0.00 | n/a | n/a | 0 | 1 |
| 0.05 | n/a | n/a | 0 | 1 |
| 0.10 | n/a | n/a | 0 | 1 |
| 0.15 | 100 | 5 | 1 | 4 |
| 0.20 | 76.47 | 17 | 1 | 16 |
| 0.25 | 79.49 | 39 | 1 | 38 |
| 0.30 | 83.93 | 56 | 1 | 55 |
| 0.35 | 76.4 | 89 | 1 | 88 |
| 0.40 | 72.17 | 115 | 1 | 114 |
| 0.45 | 69.36 | 173 | 8 | 165 |
| 0.50 | 65.38 | 234 | 19 | 215 |
| **0.55** | 59.27 | 329 | 31 | 298 |
| 0.60 | 60.78 | 408 | 46 | 362 |
| 0.65 | 57.23 | 484 | 52 | 432 |
| 0.70 | 54.56 | 570 | 65 | 505 |
| 0.75 | 54.6 | 630 | 84 | 546 |
| 0.80 | 52.55 | 685 | 96 | 589 |
| 0.85 | 51.72 | 787 | 125 | 662 |
| 0.90 | 51.91 | 865 | 151 | 714 |
| 0.95 | 54.55 | 1056 | 271 | 785 |
| 1.00 | 65.5 | 1800 | 900 | 900 |

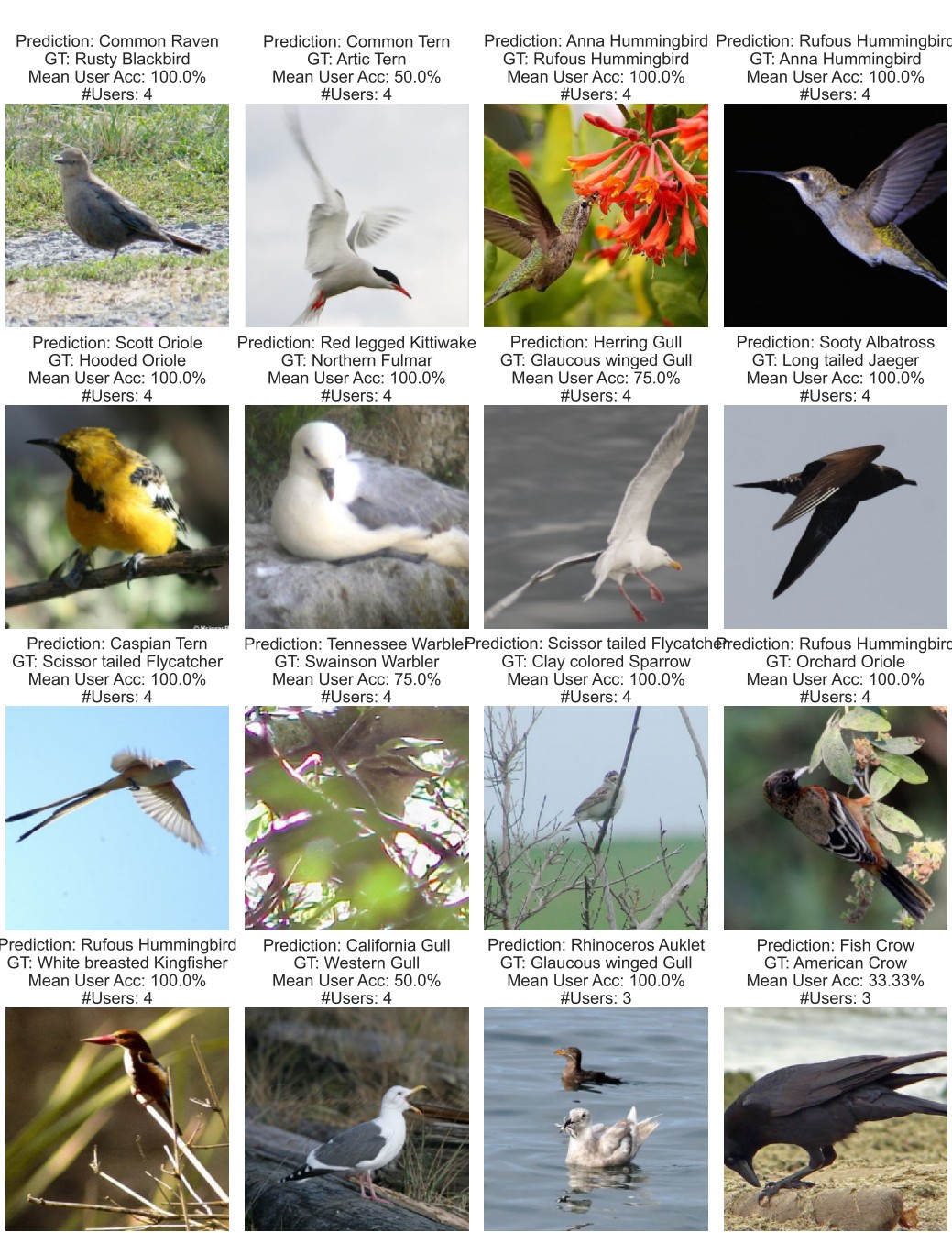

Figure A29: CUB samples at $T = 0.55$ – ResNet-50

# H Analysis for CUB

## H.1 Correspondences help users to reject wrong AI prediction

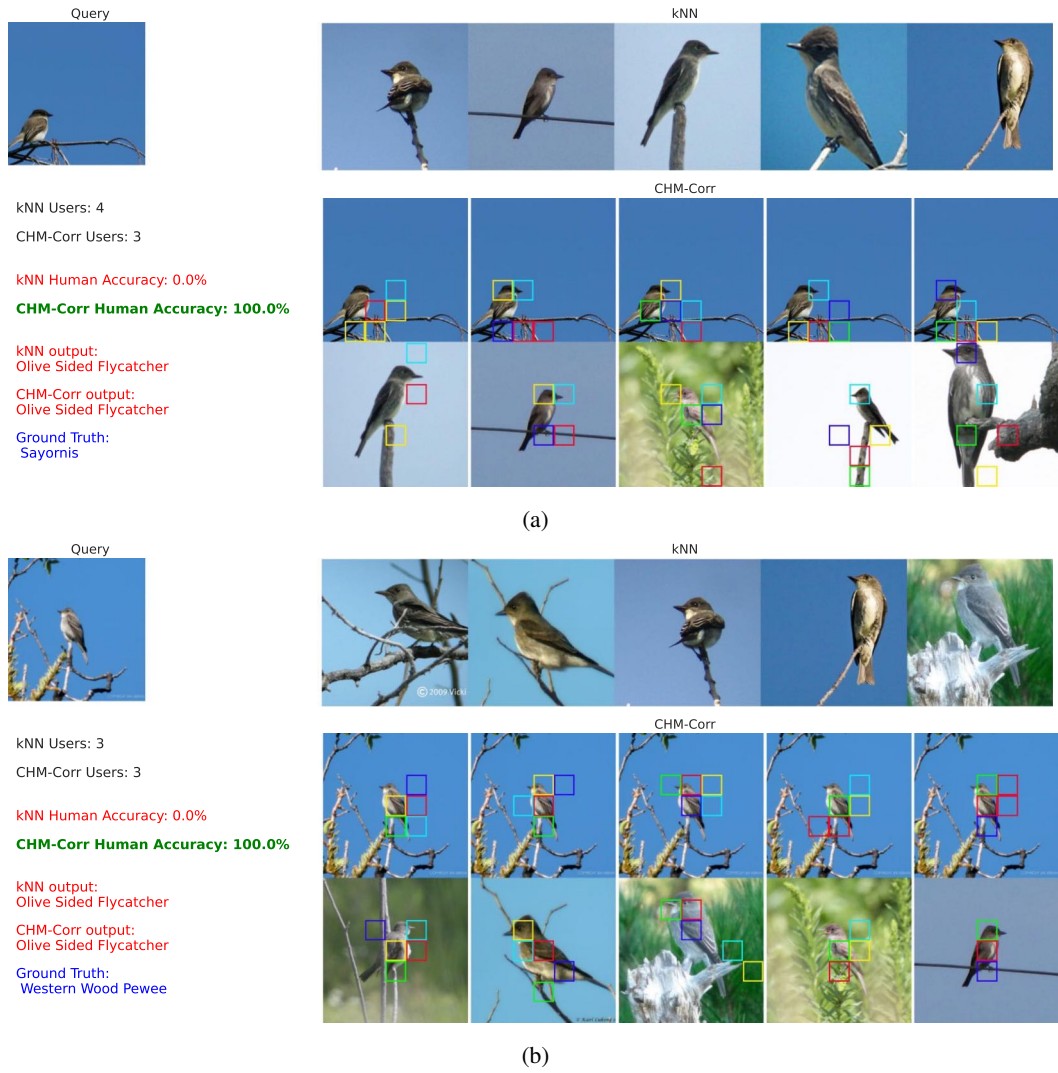

Figure A30: When correspondences help users to reject wrong AI prediction – **(a)** Both kNN and CHM-Corr classifiers misclassified an image of `Sayornis` into `Olive Sided Flycatcher`. Using kNN explanations, 4/4 of users failed to reject this wrong prediction, while using CHM-Corr explanations, 3/3 of users successfully rejected AI decisions. **(b)** Both kNN and CHM-Corr classifiers misclassified an image of `Western Wood Pewee` into `Olive Sided Flycatcher`. Using kNN explanations, 3/3 of users failed to reject this wrong prediction, while using CHM-Corr explanations, 3/3 of users successfully rejected AI decisions.

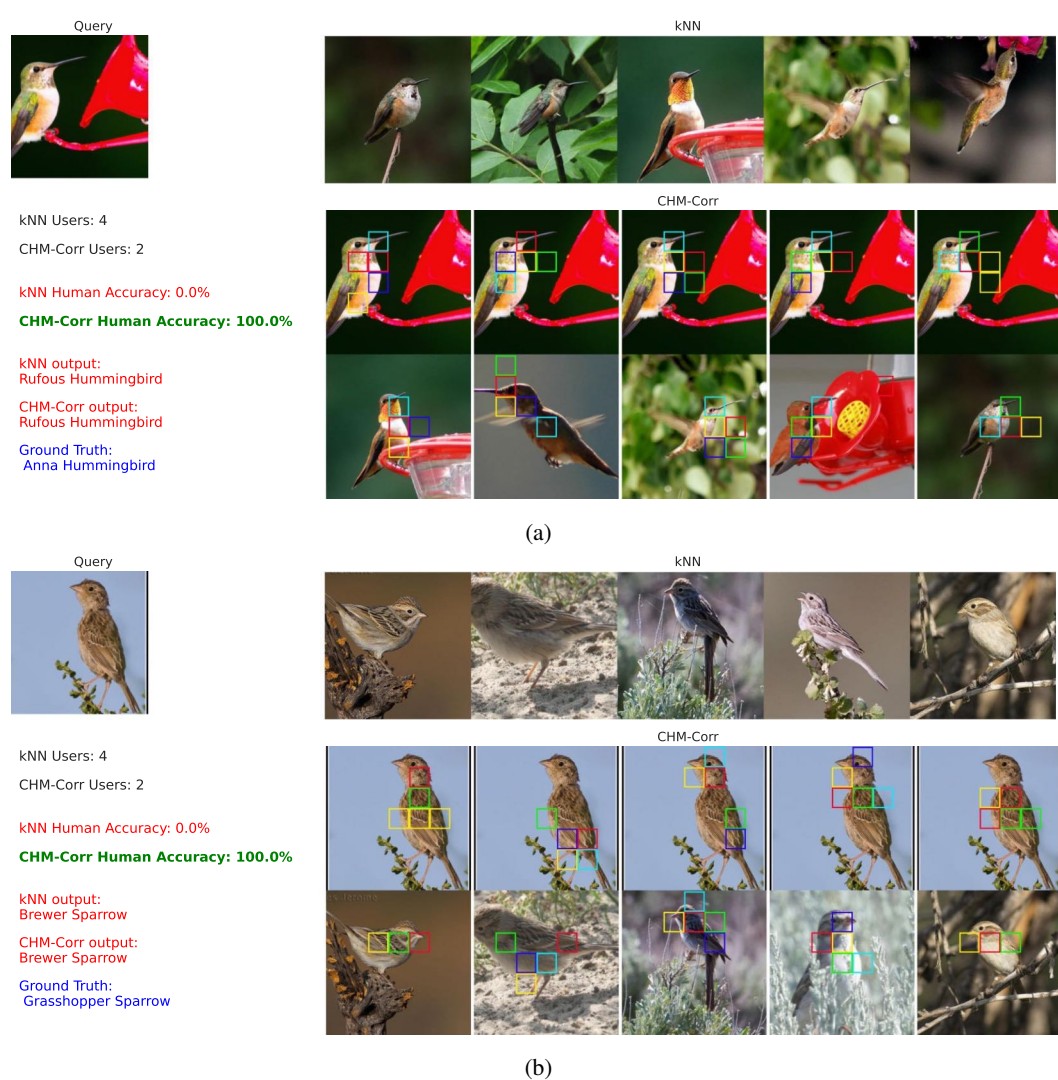

(a)

(b)

Figure A31: When correspondences help users to reject wrong AI prediction – **(a)** Both kNN and CHM-Corr classifiers misclassified an image of `Anna Hummingbird` as a `Rufous Hummingbird`. Using kNN explanations, 4/4 of users failed to reject this wrong prediction, while using CHM-Corr explanations, 2/2 of users successfully rejected AI decisions. **(b)** Both kNN and CHM-Corr classifiers misclassified an image of `Grasshopper Sparrow` as a `Brewer Sparrow`. Using kNN explanations, 4/4 of users failed to reject this wrong prediction, while using CHM-Corr explanations, 2/2 of users successfully rejected AI decisions.

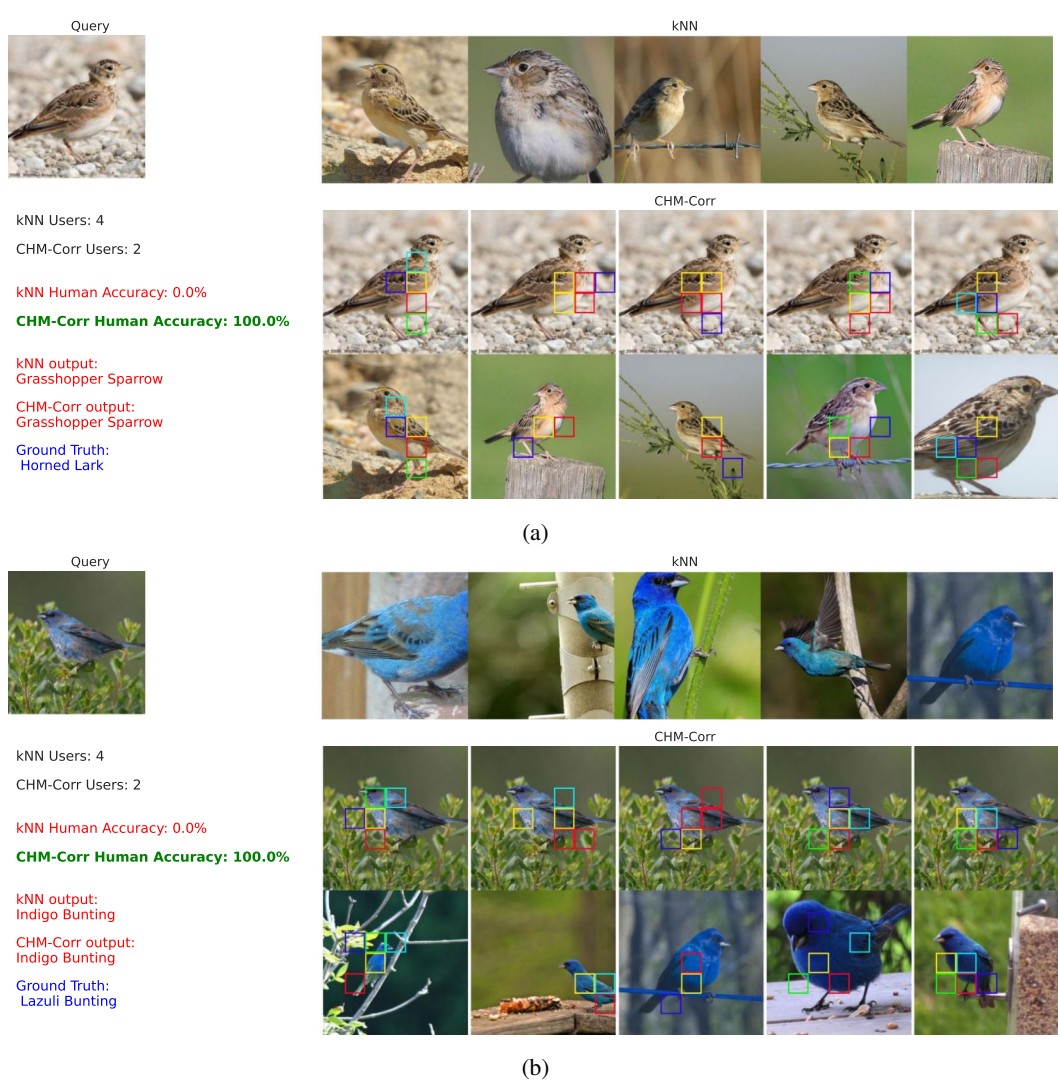

Figure A32: When correspondences help users to reject wrong AI prediction – **(a)** Both kNN and CHM-Corr classifiers misclassified an image of `Horned Lark` as a `Grasshopper Sparrow`. Using kNN explanations, 4/4 of users failed to reject this wrong prediction, while using CHM-Corr explanations, 2/2 of users successfully rejected AI decisions. **(b)** Both kNN and CHM-Corr classifiers misclassified an image of `Lazuli Bunting` as an `Indigo Bunting`. Using kNN explanations, 4/4 of users failed to reject this wrong prediction, while using CHM-Corr explanations, 2/2 of users successfully rejected AI decisions.

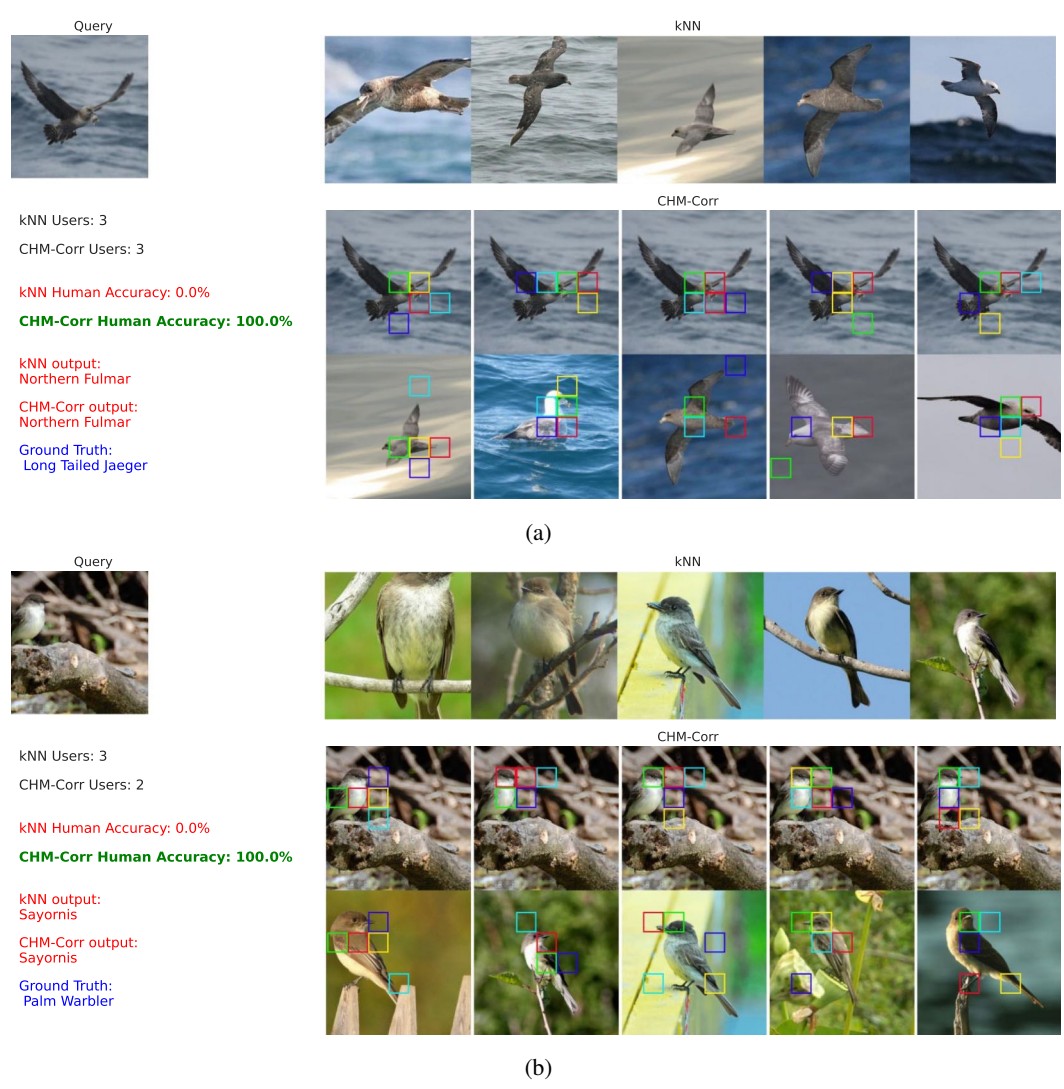

Figure A33: When correspondences help users to reject wrong AI prediction – **(a)** Both kNN and CHM-Corr classifiers misclassified an image of `Long Tailed Jaege` as a `Northern Fulmar`. Using kNN explanations, 3/3 of users failed to reject this wrong prediction, while using CHM-Corr explanations, 3/3 of users successfully rejected AI decisions. **(b)** Both kNN and CHM-Corr classifiers misclassified an image of `Palm Warbler` as a `Sayornis`. Using kNN explanations, 3/3 of users failed to reject this wrong prediction, while using CHM-Corr explanations, 2/2 of users successfully rejected AI decisions.

## H.2 Diversity of images in kNN and, EMD-Corr, and CHM-Corr explanations

We hypothesize that when the AI prediction is wrong, the diversity among the five nearest neighbors of kNN differs from EMD-Corr and CHM-Corr, leading to users rejecting the decision. To this end, we calculated LPIPS and MS-SSIM metrics between all possible pairs of explanations on the relevant queries.

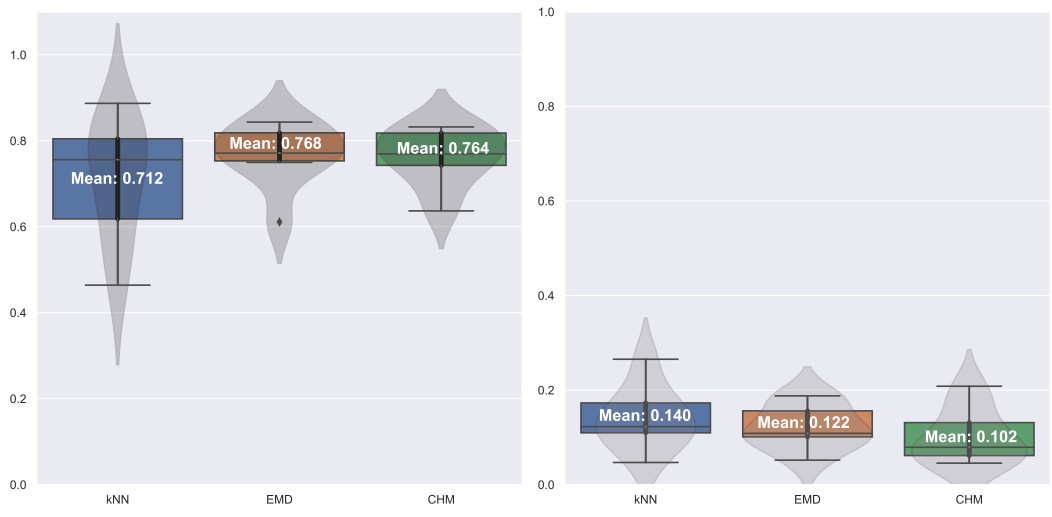

(a) LPIPS score. Higher is more diverse.

(b) MS-SSIM score. Lower is more diverse.

Figure A34: Analysis of the diversity between all 10 possible pairs of five nearest neighbors for queries with the average user's accuracy of 0% when kNN explanation is provided and the average user's accuracy of 100% when CHM-Corr explanation is provided (CUB). **The images in kNN explanations are consistently less diverse under both LPIPS (a) and MS-SSIM (b) than those in EMD-Corr and CHM-Corr explanations.** That is, this is evidence explaining why kNN users tend to be fooled into accepting kNN wrong decisions the most.

## H.3 When the user rejects the correct AI prediction

This section provides a brief qualitative explanation for the cases where users incorrectly rejected a correct AI prediction.

Query: California_Gull_0132_40836
CHM Prediction: California Gull
Users: 3
Accuracy: 0

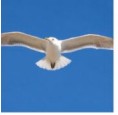 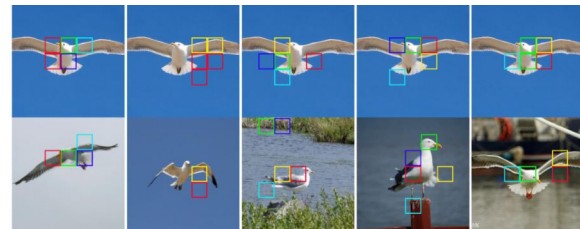

(a) The CHM-Corr classifier missed tips at wings, and the legs' black tips were occluded.

Query: Magnolia_Warbler_0021_165919
CHM Prediction: Magnolia Warbler
Users: 2
Accuracy: 0

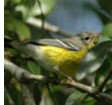 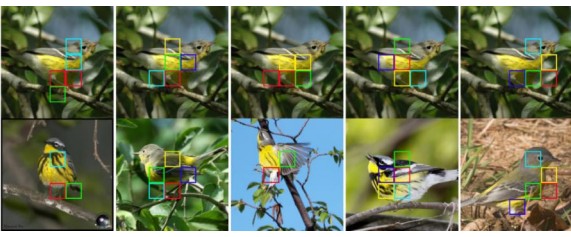

(b) The CHM-Corr classifier missed the stripes at the belly.

Query: Field_Sparrow_0092_113580
CHM Prediction: Field Sparrow
Users: 3
Accuracy: 0

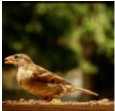 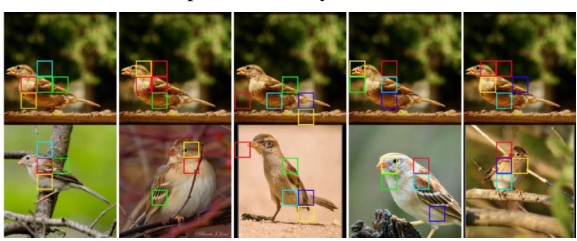

(c) Low-quality query – No distinctive features can be recognized from the input image.

Query: Sayornis_0030_98343
CHM Prediction: Sayornis
Users: 3
Accuracy: 0

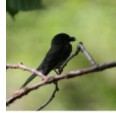 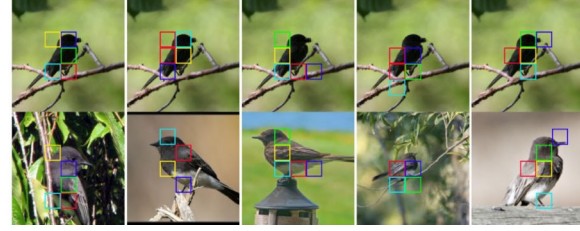

(d) Low-quality query – No distinctive features can be recognized from the input image.

Query: Shiny_Cowbird_0043_796857
CHM Prediction: Shiny Cowbird
Users: 3
Accuracy: 0

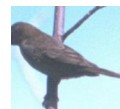 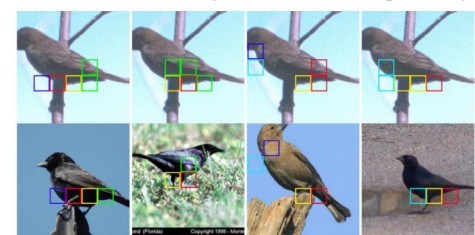

(e) Low-quality query – No distinctive features can be recognized from the input image.

Figure A35: Analysis of queries that user's rejected correct CHM-Corr prediction.

# I Comparing explanation methods

This section compares explanations provided by kNN, EMD-Corr, and CHM-Corr for various sets of queries.

## I.1 ImageNet samples

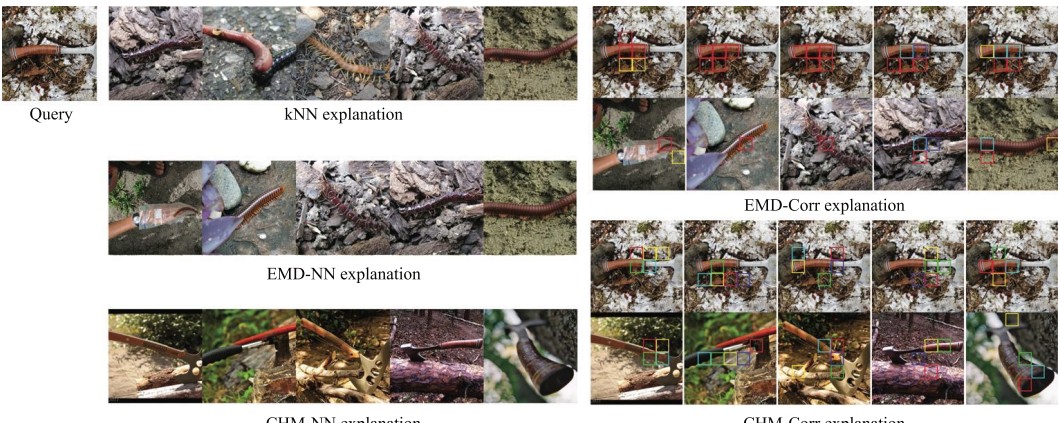

Figure A36: The kNN and EMD-Corr misclassify an image of `hatchet` as a `centipede`. The CHM-Corr correctly classifies this image.

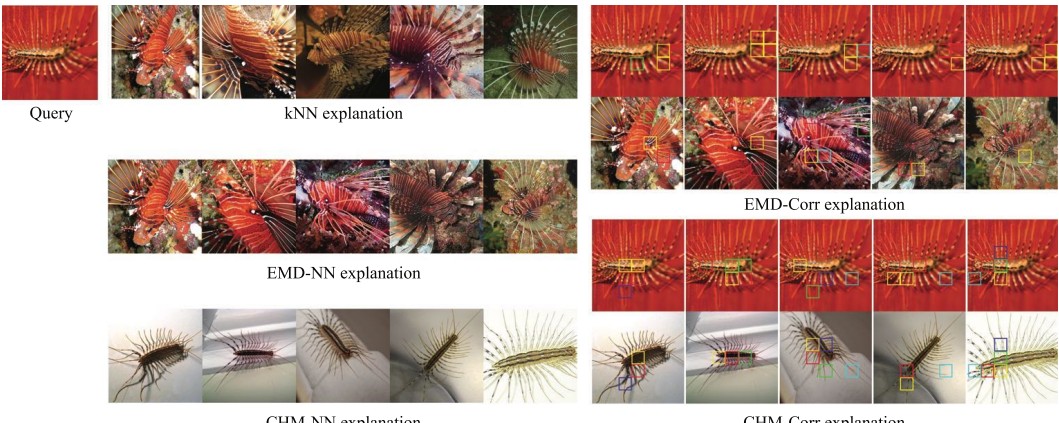

Figure A37: The kNN and EMD-Corr misclassify an image of `centipede` as a `lionfish` due to the dominant red color in the background. The CHM-Corr correctly classifies this image.

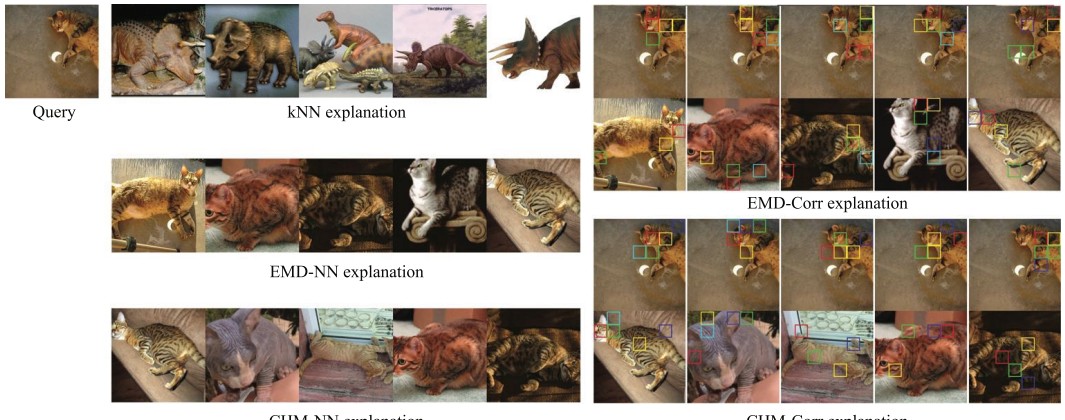

Figure A38: The kNN misclassifies an ImageNet image of `tiger cat` into `triceratops`. The EMD-Corr and CHM-Corr are both correctly classifying this image.

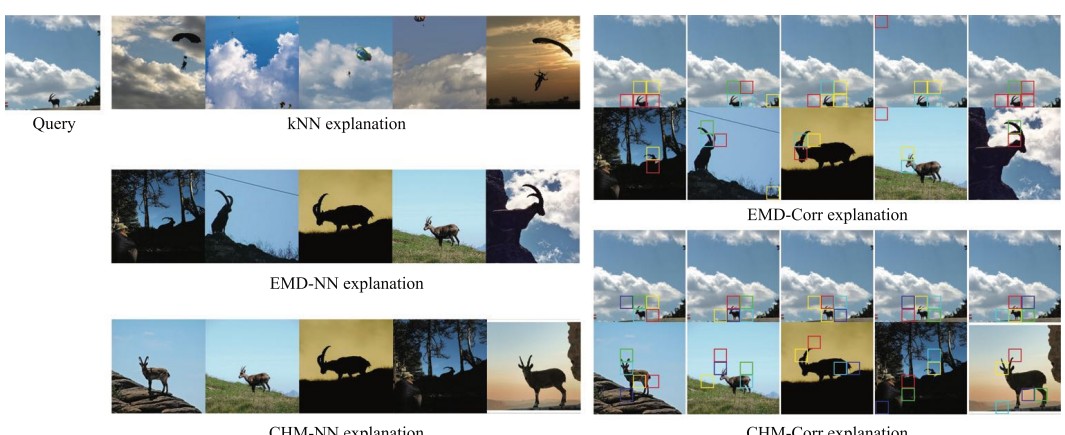

Figure A39: The kNN misclassifies an image of `ibex` as a `parachute` due to the dominant background. The EMD-Corr and CHM-Corr are both correctly classifying this image.

## I.2 Adversarial samples

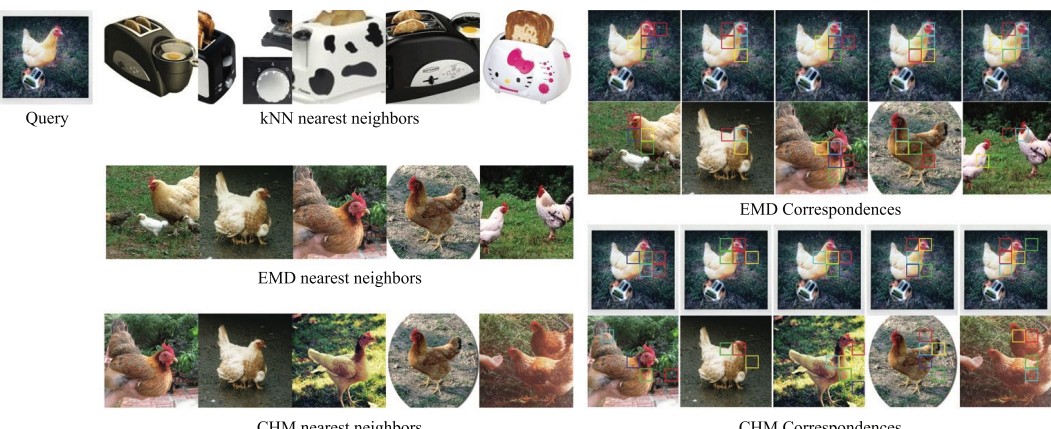

Query  kNN nearest neighbors

EMD nearest neighbors

EMD Correspondences

CHM nearest neighbors  CHM Correspondences

Figure A40: The kNN misclassifies an image of `hen` as a `toaster` due to an adversarial patch. The EMD-Corr and CHM-Corr are both correctly classifying this image.

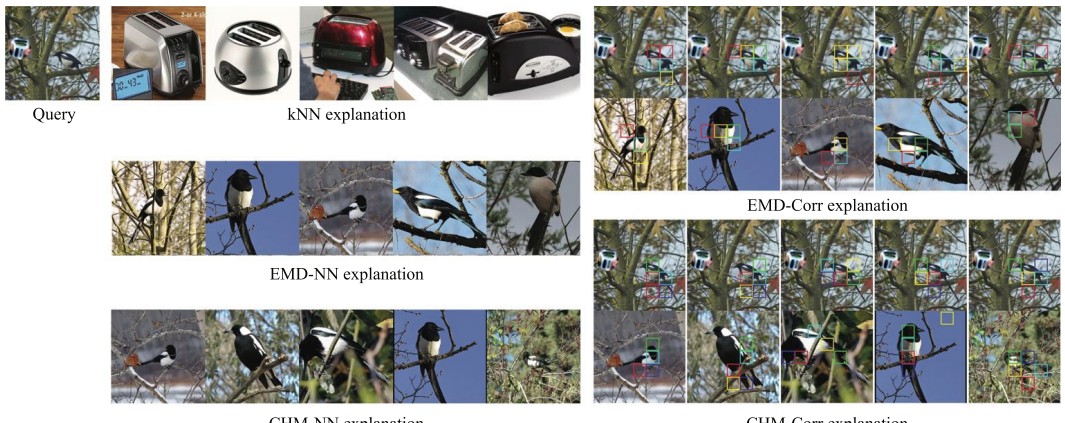

Query  kNN explanation

EMD-NN explanation

EMD-Corr explanation

CHM-NN explanation  CHM-Corr explanation

Figure A41: The kNN misclassifies an image of `magpie` as a `toaster` due to adversarial patch. The EMD-Corr and CHM-Corr are both correctly classifying this image.

## J    Controlling keypoints in CHM-Corr+ for the CUB dataset

Here, we compare CHM-Corr and CHM-Corr+ classifiers to understand the low performance of CHM-Corr+. Using a set of **five** keypoints may not help CHM-Corr+ focus on the right patches. Sometimes, the five provided keypoints are not among the discriminative features to correctly classify a bird.

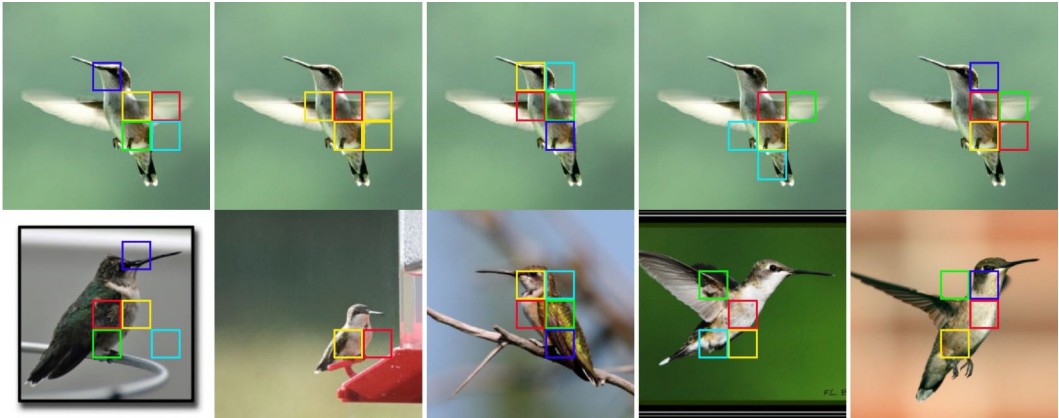

(a) The explanation of a correct classification by CHM-Corr.

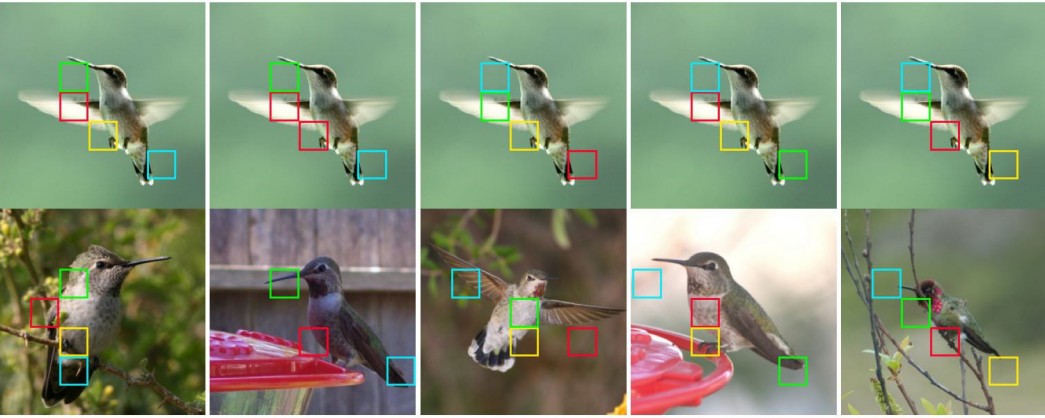

(b) The explanation of a misclassification by CHM-Corr+

Figure A42: A `Ruby-throated Hummingbird` misclassified into `Anna Hummingbird` by CHM-Corr+. An example of low-quality keypoints leading to selecting and comparing mostly background (uninformative) patches.

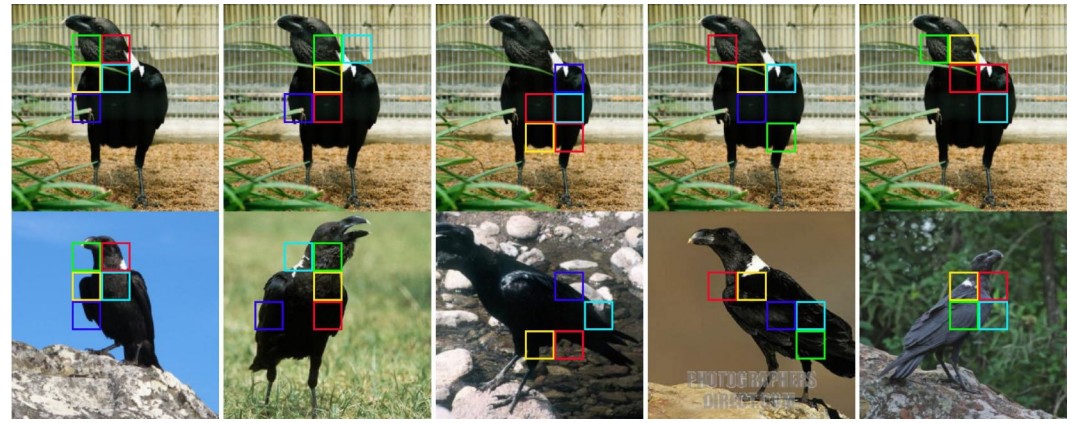

(a) The explanation of a correct classification by CHM-Corr.

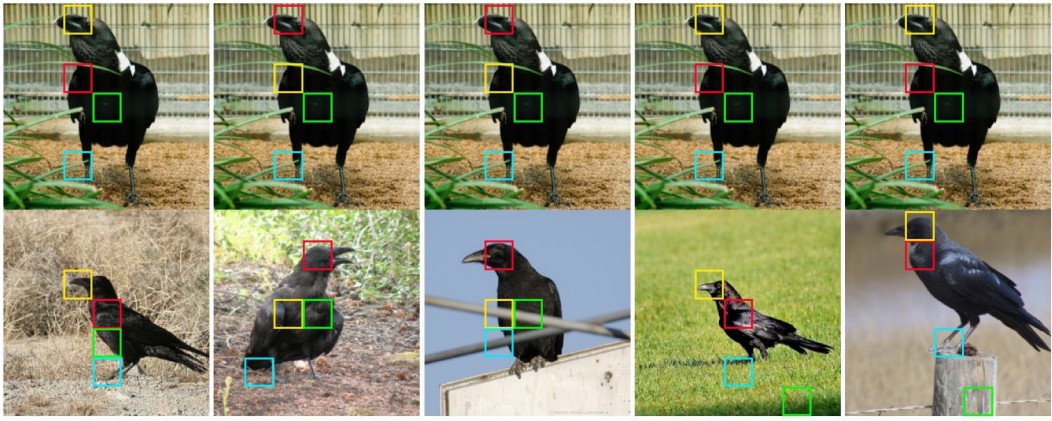

(b) The explanation of a misclassification by CHM-Corr+.

Figure A43: A `White necked Raven` misclassified as a `Common Raven` by CHM-Corr+ – The distinctive part of the bird is 'the white feathers on the neck', which is missed in the keypoints selection step in the CHM-Corr+. The CHM-Corr classifier correctly classifies this image.

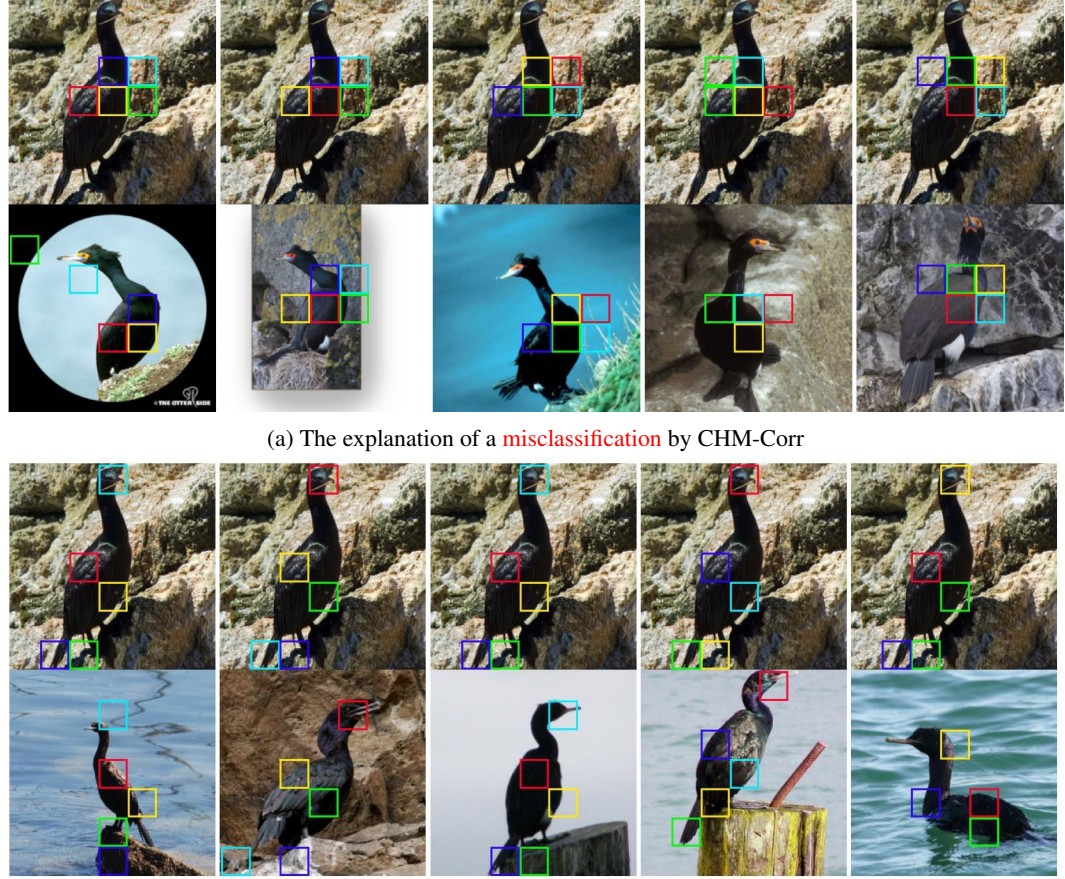

(a) The explanation of a misclassification by CHM-Corr

(b) The explanation of a correct prediction by CHM-Corr+.

Figure A44: A `Pelagic Cormorant` misclassified as a `Red Faced Cormorant` by CHM-Corr. The face of the bird was not among the top-5 correspondences picked by CHM-Corr, which led to misclassification. The CHM-Corr+ classifier correctly classifies this image.

# K    Samples for ImageNet-Sketch dataset

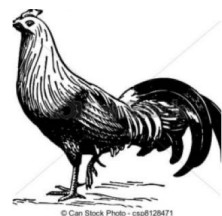

(a) Query – Cock

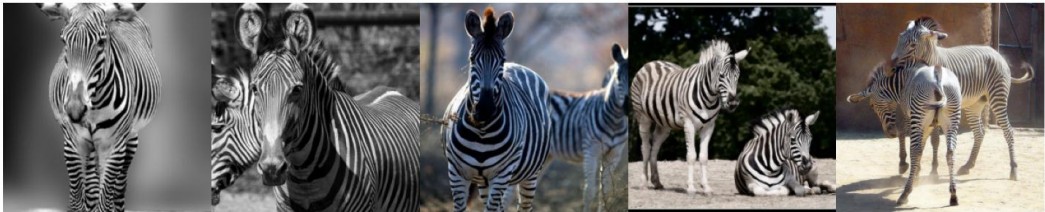

(b) Nearest neighbors using kNN

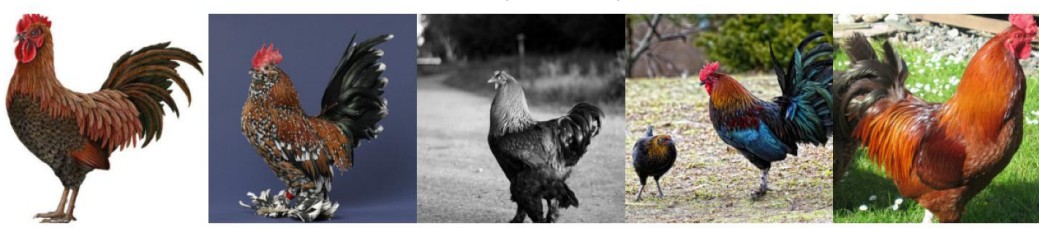

(c) Nearest neighbors after re-ranking using CHM-Corr

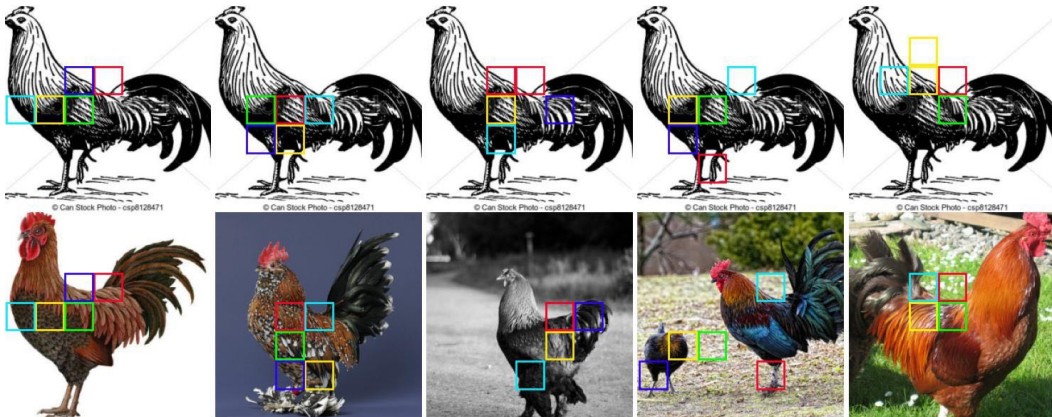

(d) CHM-Corr explanation

Figure A45: A misclassification by the kNN classifier. The black-and-white stripe patterns in cock confuse the kNN classifier, while the CHM-Corr classifier correctly labels the query.

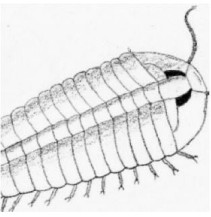

(a) Query – `Trilobite`

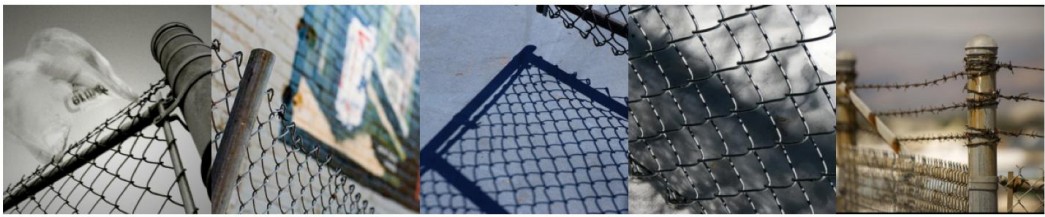

(b) Nearest neighbors using kNN

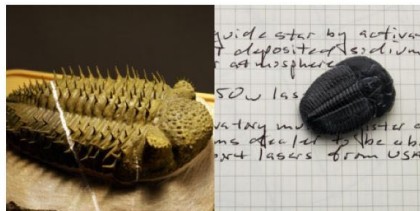

(c) Nearest neighbors after re-ranking using CHM-Corr

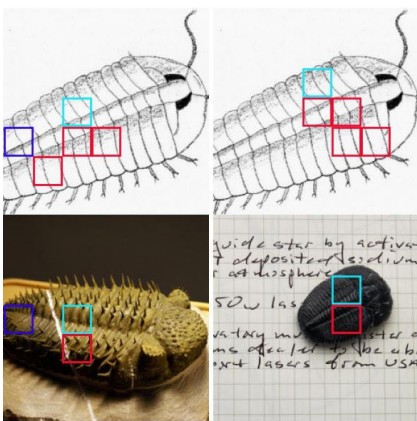

(d) CHM-Corr explanation

Figure A46: A misclassification by the kNN classifier. An image of `trilobite` misclassified as a `chainlink fence` by the kNN classifier, while the CHM-Corr classifier correctly classifies the query. The confidence score of CHM-Corr is only 2/20, i.e., 10%. That is, only two `trilobite` images are among the top 20 candidates.

## L Removing duplicated images from the ImageNet validation set

Some of the images from the ImageNet validation set are also present in the training set. For the human study, we excluded such images from our study. Figure A47, shows some of these samples along with their five nearest neighbor images from the training set.

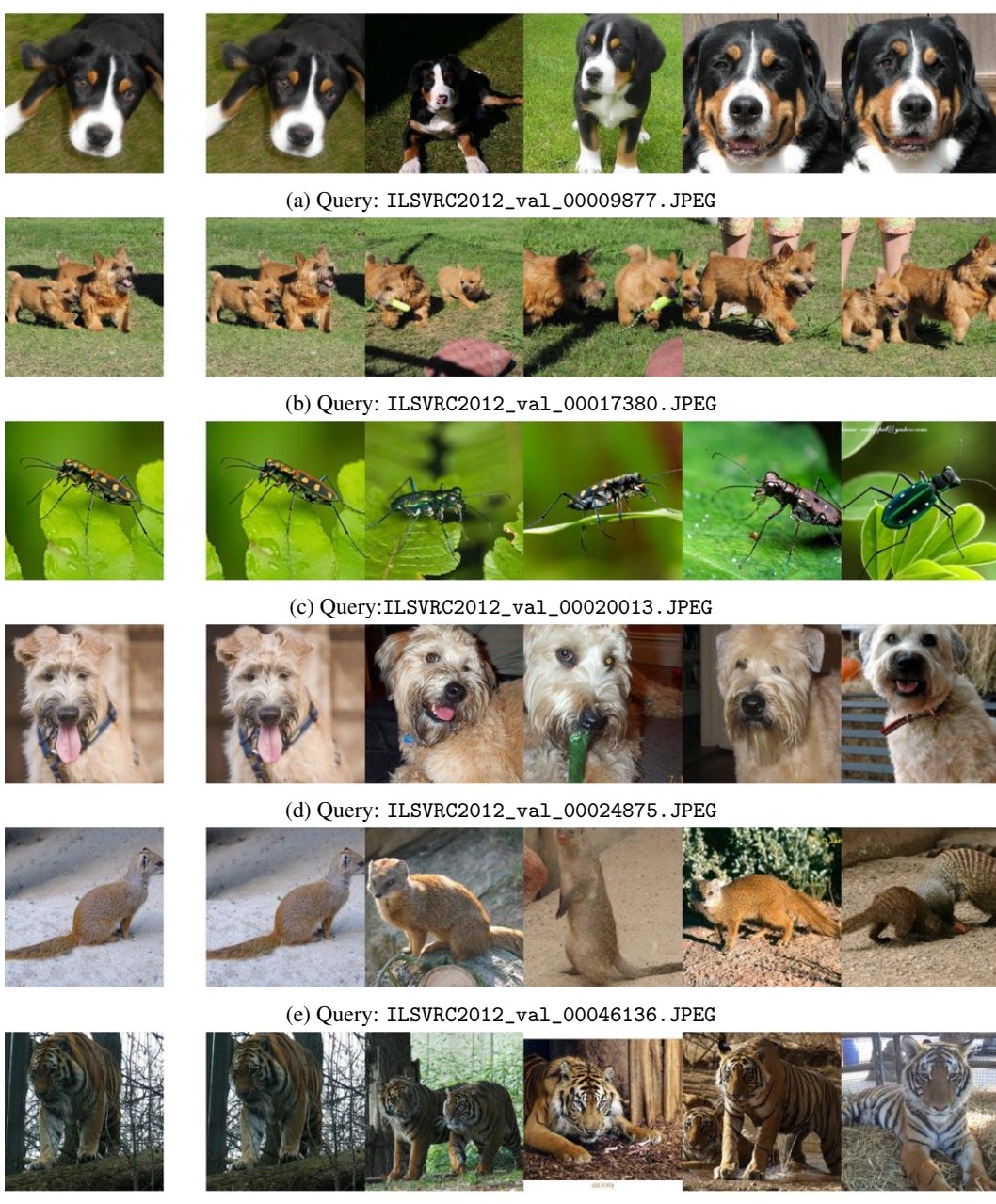

(a) Query: ILSVRC2012_val_00009877.JPEG

(b) Query: ILSVRC2012_val_00017380.JPEG

(c) Query:ILSVRC2012_val_00020013.JPEG

(d) Query: ILSVRC2012_val_00024875.JPEG

(e) Query: ILSVRC2012_val_00046136.JPEG

(f) Query: ILSVRC2012_val_00014815.JPEG

Figure A47: In each panel, the leftmost image is in the validation set, and the top-5 nearest images on the right are from the training set. We find images that exist both in the training set and validation set and remove them from our validation set (in order to not unfairly favor kNN in the study).

# M  Human-AI teams outperform both AIs alone and humans alone

In Sec. 3.5, we find that user classification accuracy can be improved when humans are provided with AI predictions and explanations. Here, we leverage the same data collected from the previous ImageNet-ReaL and CUB human studies (Secs. 3.4 and 3.5) to estimate the accuracy of a human-AI team that allows both humans and AIs to make final decisions (Fig. 4; Model 2).

That is, AIs make binary Yes/No predictions on all the $X\%$ of query images that they assign a high confidence score $\geq T$ where $T \in [0, 1]$ (Fig. 4b). Given these images and AI predictions, we compute an accuracy score, $\mathrm{acc_{AI}}$. For the set of remaining images (i.e. whose AI confidence is $< T$), we take their user predictions and also compute an accuracy score $\mathrm{acc_{human}}$. We define the human-AI team accuracy $\mathrm{acc_{team}}$ as:

$$\mathrm{acc_{team}} = X\% \times \mathrm{acc_{AI}} + (100 - X)\% \times \mathrm{acc_{human}}$$

As the interaction model 2 is more practical and scalable, it is interesting to test how the $\mathrm{acc_{team}}$ compares with the accuracy when users or AIs alone classify all images (i.e. when $X = 0$ or $100$).

**Experiment**    For each classifier (ResNet-50, kNN, EMD-Corr, and CHM-Corr), we use a 2K-image held-out subset of the ImageNet-ReaL validation[3] set to find an optimal threshold $T$ that maximizes the classifier's binary classification accuracy. Then, we use the remaining ∼42K ImageNet-ReaL validation images for testing. For CUB, we tune $T$ using 1K test images and test on the remaining ∼4.7K test-set images. For both ImageNet and CUB, we did not use the training sets to tune $T$ because the top-1 neighbors retrieved by kNN would be identical to the query all the time, biasing the AIs as well as humans when they perform classification.

After obtaining an AI accuracy score for each value of $T \in \{0.05, 0.10, 0.15, ..., 0.95\}$, we find the best $\mathrm{acc_{team}}$ (at an optimal $T^*$) and repeat the same process to find the best AI-only accuracy. More details of the sweeps are in Appendix G.

**Results**    First, across all four classifiers and two datasets, AI-only accuracy is consistently higher than human-only accuracy (Table 3 vs. Table 2). That is, letting users make all the AI-assisted decisions is both more labor-intensive and less accurate compared to allowing AIs to classify all the data themselves. This result is consistent with the prior studies that find AIs to outperform humans [28, 78, 59, 26] (see [12] for a summary).

Second, interestingly, human-AI teams consistently outperform the AIs alone (Table 3) and humans alone (Table 2). That is, lay-users may be considered "expert" on ImageNet's everyday objects and therefore, when teaming up with humans to form human-AI teams, the accuracy substantially increases on average by +2.11 (Table 3). On CUB, which is more challenging to lay-users, this benefit of teaming up with users is negligible (Table 3; +0.02)

Third, among four classifiers, ResNet-50 yields the highest human-AI team accuracy on both ImageNet-ReaL and CUB (Table 3). However, the variance in team accuracy across the classifiers is small. Our results interestingly imply that while there is significant evidence that Corr explanations are useful to the AI-assisted decision-making of humans in the interaction model 1 (Table 2; CUB), such benefits of XAI models average out in the interaction model 2.

---

[3]Because the training set is used by non-parametric classifiers during testing, for ImageNet, we tune $T$ using 2K-image validation images with ImageNet-ReaL labels. We use 1K test images for CUB.