# OpenReview forum: "Visual correspondence-based explanations improve AI robustness and human-AI team accuracy"
_NeurIPS.cc/2022/Conference — NeurIPS 2022 Accept_

### Official Review · Reviewer_riHj · 2022-07-07

**Rating:** 3
**Confidence:** 4
**Soundness:** 2 fair
**Presentation:** 1 poor
**Contribution:** 2 fair

**Summary:**

This paper proposes a classifier based on different refining strategies of the closest nearest neighbors using intermediate features from a ResNet50 network. These refining is based on a spatial patch-based correspondence, providing explanations in the form of example images showing the top similar sub-areas to a query.

Experiments show that these kind of classification pipeline is robust when classifying adversarial samples. Moreover, human-studies indicate that the intermediate patch-matching method convey information about the computer’s decision such that humans can improve upon their own classification baseline (and that of the model alone).

**Questions:**

1. What are the computational costs associated with this kind of classifiers, compared to a regular Resnet50?

2. Why are the results of an AI-Only evaluation on ResNet-50 (Table 2 reports an accuracy of 86.05%) much higher than the performance of the one reported in the baseline evaluation (Table 1 reports an accuracy of 76.13%)?

3. How does classification accuracy compare to the usual "tricks" that can be applied to pre-trained models (e.g., fine-tuning some of the deeper layers) or adversarial training?

**Limitations:**

1. The high computational cost of the proposed methods (as mentioned towards the end of the paper) has not been addressed quantitatively. This is especially important given the incremental improvements that have been reported with respect to their baselines.

**Strengths And Weaknesses:**

- *Originality*
    - The overall methods to find correspondences, sort matching pairs and issue a classification prediction are mostly based on previous work. The method is mostly similar to those of Hai or Zhao ([25] and [67] in the paper). The proposed modifications to focus on the top patches and the use of a re-ranking optimization serve as a well-founded refinement of this method, making the computations more traceable and therefore, more interpretable.

- *Quality*
    - The selection of experiments is thorough and extensive.
    - From all variants of Imagenet, one that is particularly interesting when evaluating XAI methods is the subset of “natural adversarial samples” (a.k.a. Imagenet-A) [2]. Unfortunately, this dataset was not included as part of the evaluation.

- *Clarity*:
    - The paper lacks a clear storyline and often jumps between results, experimental setup and motivation. Even though the contributions appear to be sound and significant, the lack of clarity presents problems for reproducibility.
    - Important parts of the experimental setup (including data and methods) are been cited without elaborating further on the specific parts. While not necessarily a mistake, the paper ends is not self-contained at all, which heavily impacts readability and reproducibility.
    - Quite a few passages are poorly structured. Even though it is possible to make sense of what is being said, the clarity of the paper suffers considerably.
    - The discourse of the paper jumps between showing classification improvements and OOD robustness. These are two different problem setups that are not necessarily interchangeable.

- *Significance:*
    - Evaluating on adversarial datasets sets an ambitious and valuable goal.
    - The inclusion of large human studies with careful policies for selecting (and discarding) subjects potentiates the credence of these results.
    - Experiments show that in 7/8 of setups, the kNN baseline is within 1.6pp. of the proposed non-parametric classifiers (the second to last one is under 6pp.). Despite its statistical significance, there is not enough evidence that can justify the additional computational resources for running kNN at this stage, let alone the proposed re-ranking metrics. More concretely, how do these results compare to the usual "tricks" that can be applied to pre-trained models (e.g., fine-tuning some of the deeper layers)?
    - Regarding experiments on Damagenet and adversarial patches: the reported improvements, even though they can be statistically significant, are still low relative to the performance on clean samples. Moreover, it is known that this kind of improvements get proven useless by generating adversarial attacks that are tailored to the classifier (kNN classifiers in this case). For instance, by using non-differentiable approximations [1].

### References:

1. Athalye, Anish, Nicholas Carlini, and David Wagner. "Obfuscated gradients give a false sense of security: Circumventing defenses to adversarial examples." *International Conference on machine learning*. PMLR, 2018.
2. Hendrycks, Dan, et al. "Natural adversarial examples." *Proceedings of the IEEE/CVF Conference on Computer Vision and Pattern Recognition*. 2021.

---

> ### Author Response · Authors · 2022-08-02
> **we clarified our novel contributions and explained the AI-only accurate**
>
> Thank you so much for your detailed and thoughtful comments!
> On OpenReview, our paper has been revised in light of your valuable feedback. We have substantially improved the clarity of the writing and presentation of the results.
>
> We believe our revised manuscript has addressed all of your concerns. Thus, we wish you consider accepting our work.
> Please see our inline responses to your comments below.
>
> ### **> Improve manuscript clarity**
>
> We agree our first submitted version has presentation issues that affected readability. In light of your constructive comments, we revised entirely the manuscript to improve the clarity signficantly and provide details to make the experiments much easier to understand and reproduce. The list below contains the main changes in our new revision:
>
> - For reproducibility, we've released [an anonymized github repo](https://github.com/visualcorr/anonymous-repo) with detailed instructions for how to run our classifiers and to download our generated Adversarial Patch dataset. In the repo, we also provide a [walkthrough video](https://drive.google.com/file/d/14yBI5TIaD6s-6YkWwa4yEzSBGmI-nw8d/view) showcasing our Gorilla setup for the community's reproduction of our human studies.
>
> - We added diagrams and clarified description (Fig. 4 and Sec. 2.3) of our two human-AI interaction models assessed in this paper.
>
> - We added a runtime analysis of all four main classifiers (Appendix C). We note that using FAISS library, running kNN is fairly tractable. That is, on CUB, kNN takes 9.7s for 1,000 queries on one Nvidia V100 while that takes ResNet-50 8.81s. These numbers are 17.35s and 9.17s on ImageNet.
>
> - We revised the entire paper, fixing many typos, improving the clarity of the writing, and and re-organizing the results into a single storyline: We first propose building and testing kNN and visual correspondence-based XAI classifiers on i.i.d. and OOD data (Secs. 3.1 and 3.2) and then we test them on assisting users in two human-AI interaction models (Secs. 3.3, 3.4, and Appendix M). Our work extensively assessed the classifiers on **three binary-decision-maker settings: AI-only, human-only, and human-AI teams.**
>
>
> ---
>
> ### **>Why are the results of an AI-Only evaluation on ResNet-50 (Table 2 reports an accuracy of 86.05%) much higher than the performance of the one reported in the baseline evaluation (Table 1 reports an accuracy of 76.13%)?**
>
> In Table 1, we report the top-1 accuracy for 1000-way ImageNet classification and 200-way CUB classification.
>
> Because we further evaluate our classifiers on two different human-AI interaction models (see new Fig. 4 in our revision) where humans make a binary Yes/No decision following the setup in the XAI literature [a][b] (accepting or rejecting AI's top-1 label), we needed to also compare with an AI-only baseline where AIs make Yes/No decisions in the same setup without humans (i.e. AI-only in Table 3).
> That is, we use a simple thresholding mechanism to allow AIs to accept its own decisions (Yes) or refrain from making decisions (No) if its confidence is below a threshold T (detailed description in Appendix M).
>
> As seen in Table 3, adding humans in the loop substantially improve AI-only accuracy on this binary classification task.
>
> References
>
> [a] The effectiveness of feature attribution methods and its correlation with automatic evaluation scores. NeurIPS 2021
>
> [b] HIVE: Evaluating the Human Interpretability of Visual Explanations. ECCV 2022

---

> ### Author Response · Authors · 2022-08-02
> **we ran your requested results on ImageNet-A**
>
> ### **> From all variants of Imagenet, one that is particularly interesting when evaluating XAI methods is the subset of “natural adversarial samples” (a.k.a. Imagenet-A).**
>
> Thank you for this interesting suggestion!
>
> We were not sure whether you suggest testing our XAI methods on ImageNet-A, **with or without** humans in the loop.
> Therefore, we did both and report the results below.
>
>
> **AI-only results.**
> Consistent with other OOD results, on ImageNet-A, kNN outperforms ResNet-50 and the Corr classifiers further improve the performance.
> The improvement is small but consistent.
> Importantly, our results show that clever use of the exact same ResNet-50 features can further improve OOD accuracy.
> Like other OOD results (gray rows in Table 1), ImageNet-A is an out-of-distribution test (we never tuned any hyperparameters for ImageNet-A or any OOD datasets in the paper).
>
>
> |  Dataset | # of images | ResNet-50 |  kNN  | EMD-Corr | CHM-Corr |
> |:---------:|:-------:|:------------------:|:-----:|:--------:|:--------:|
> | ImageNet-A |   7,500  |        0.04%       | 0.32% |   0.59%  |   0.42%  |
>
>
> **Human-only results.**
> Most human studies in the XAI research for computer vision test classifiers on the ImageNet validation set and CUB [a][b][c].
> We find only a 4-page preprint [d] that conducts an XAI human study on ImageNet-A. However, Folke et al. 2021 [d] asks users to select the AI's top-1 predicted label from two given labels, which measures how well users _understand AI_ as opposed to how well users perform image classification (the objective of our study).
>
> We conducted a small-scale, 5-user study on ImageNet-A with the same human-study setup as we did with ImageNet-validation and CUB-200 (Sec. 2.3).
> Users are not informed of which classifiers they are working with and the classifiers' accuracy.
> Below are the results.
>
> |   Method   |   User  1    |   User 2    |    User 3    |    User 4    |   User 5    | Mean |      Std      |
> |:---------:|:-------:|:-------:|:-------:|:-------:|:-------:|:-------:|:-------------:|
> |   No XAI  | 100.00% | 100.00% |  95.00% | 100.00% | 100.00% |  99.00% | 2% |
> |    KNN    |  85.00% |  70.00% | 100.00% | 100.00% |  90.00% |  89.00% |  12%  |
> | EMD-Corr  |  70.00% |  75.00% |  90.00% |  80.00% |  90.00% |  81.00% |  8% |
> |  CHM-Corr |  90.00% | 100.00% |  95.00% |  90.00% |  90.00% |  93.00% | 4% |
>
>
> Our observation: ResNet-50 misclassifications are easy to detect by humans because the incorrect top-1 label is often irrelevant to the groundtruth label (e.g., scorpion vs. black bear).
> Thus, ImageNet-A poses a trivial task for humans to detect whether ResNet-50 is correct or not **without any explanations**.
>
> A reason why ResNet-50 user accuracy on ImageNet-A is higher than that of kNN users is the following: Since AI accuracy is mostly near-zero, showing nearest neighbors will hurt user accuracy because explanations tend to increase the rates that humans accept AI's decisions (regardless of whether the decision is correct or not).
>
> - 62.33% to 63.56% in Table. A9 [a].
> - 60.44% to 69.60% in Table. A6 of our revised paper.
>
> This tendency of users to mistrust AI decisions when explanations are presented was commonly reported in many XAI studies, including [a][b].
>
> We will discuss the ImageNet-A result and Folke et al. [d] in the camera-ready paper.
>
>
> References
>
> [a] The effectiveness of feature attribution methods and its correlation with automatic evaluation scores. Nguyen et al. NeurIPS 2021.
>
> [b] HIVE: Evaluating the human interpretability of visual explanations. Kim et al. ECCV 2022.
>
> [c] What I Cannot Predict, I Do Not Understand. Fel et al. 2021
>
> [d] Explainable AI for Natural Adversarial Images. Folke et al. ICLR workshop 2021.

---

> ### Author Response · Authors · 2022-08-02
> **we provide an runtime comparison between all classifiers; kNN runs fairly fast**
>
> ### **> Experiments show that in 7/8 of setups, the kNN baseline is within 1.6pp. of the proposed non-parametric classifiers (the second to last one is under 6pp.). Despite its statistical significance, there is not enough evidence that can justify the additional computational resources for running kNN at this stage, let alone the proposed re-ranking metrics.**
>
> We wish to clarify the rationale of running kNN as follows:
>
> 1. In machine learning, previous works find kNN to improve OOD robustness on MNIST and CIFAR-10 [a,c], but no experiments were run on ImageNet-scale tests. In XAI, many prior studies find nearest-neighbor-based explanations useful in various settings [b,d,e,f,g,h]. Motivated by these works, we test kNN on OOD ImageNet datasets and find that the results, on both fronts, can be further improved by patch-based re-ranking.
>
> 2. After a one-time pre-processing using FAISS (a few mins on CUB or 90 mins for 1.3M ImageNet training set), running inference on kNN is fairly fast compared to ResNet-50.
> Our kNN classifier implemented using the FAISS library [x], enables us to perform an efficient similarity search on massive datasets in milliseconds compared to a naive kNN implementation.
> On top of kNN runtime, EMD-Corr re-ranking step takes additionally ~2 seconds per query, which is a reasonable wait time for high-stakes decision tasks.
> CHM-Corr is our slowest classifier, taking an extra ~10 seconds for re-ranking kNN results (the slow CHM-Corr PyTorch implementation can be significantly improved with an efficient JAX implementation).
>
> Please see the detailed runtime comparison below (and more in Appendix C).
> We ran each classifier 5 times on a batch of 1000 random queries.
>
> **Table**: Average runtime (in seconds) on 1,000 queries over 5 runs on one Nvidia V100.
>
> |        **Method**       |     ImageNet     |       CUB       |
> |:-----------------------|----------------:|---------------:|
> | ResNet-50               |    9.17 ± 0.19   |   8.81 ± 0.14   |
> | kNN (*FAISS)            |    17.35 ± 1.28    |    9.70 ± 0.32   |
> | kNN (Naive - GPU)       |   1,112.46 ± 0.86   |   23.88 ± 0.58  |
> | EMD-Corr reranking step |   2,218.92 ± 99.14  | 1,927.69 ± 17.48 |
> | CHM-Corr reranking step | 10,642.85 ± 1007.87 | 6,920.76 ± 67.58 |
>
>
> References
>
> [x] Billion-scale similarity search with GPUs - IEEE Transactions on Big Data 2019
>
> [a] Deep k-Nearest Neighbors: Towards Confident, Interpretable and Robust Deep Learning
>
> [b] The effectiveness of feature attribution methods and its correlation with automatic evaluation scores
>
> [c] How Can I Explain This to You? An Empirical Study of Deep Neural Network Explanation Methods
>
> [d] Interpreting Neural Networks with Nearest Neighbors
>
> [e] Advancing Nearest Neighbor Explanation-By-Example With Critical Classification Regions
>
> [f] Representer Point Selection for Explaining Deep Neural Networks
>
> [g] DeepFace-EMD: Re-ranking Using Patch-wise Earth Mover’s Distance Improves Out-Of-Distribution Face Identification
>
> [h] p-DkNN: Out-of-Distribution Detection Through Statistical Testing of Deep Representations

---

> ### Author Response · Authors · 2022-08-02
> **We provide results of adversarially trained ResNet-50 backbone**
>
> ### **> How do these results compare to the usual "tricks" that can be applied to pre-trained models (e.g., fine-tuning)?**
>
> This is an interesting question!
>
> We think that it is _not_ a direct comparison to compare finetuning to kNN. Here, we test kNN on exactly out-of-distribution tests (e.g. ImageNet-A) where there are no extra samples to adapt the model or to add to the training/support set of kNN.
>
> Instead, we believe one can improve the backbone models using finetuning, adversarial training, or other "usual tricks", and then, apply kNN on top of the improved backbone for a further performance boost.
> Per your request, we test an ImageNet-trained ResNet-50 trained via AdvProp PGD-5 [a] (i.e., trained on both real ImageNet data and adversarial images generated via PGD-5).
> We compare this model and its own kNN results using the same AdvProp ResNet-50 backbone.
>
>
> Due to the time constraint of the rebuttal period, we only test on a 5K random subset of each dataset. Interestingly, **we observe the same trend as reported in the paper: kNN further improves AdvProp ResNet-50 on OOD datasets** and is worse on i.i.d. ImageNet.
>
> |    Datasets    | ResNet-50 | k-NN |
> |:--------------|------------------:|-------------------------------------------------------:|
> |  ImageNet |        **77.18**       |                           73.20                          |
> |   ImageNet-R   |        40.32       |                           **42.50**                          |
> |   ObjectNet    |        36.92       |                          **37.02**                          |
> | ImageNet-Sketch |        27.30        |                           **28.30**                          |
> |   ImageNet-A   |        4.26        |                           **6.48**                          |
> |    DAmageNet   |        22.84       |                          **27.46**                          |
>
> Compared to our results when using the vanilla ResNet-50 backbone (Table 1 in our paper), these results for AdvProp ResNet-50 backbone are substantially higher.
>
> For completeness, we will add this interesting result to our camera-ready version. Thank you for the great suggestion!
>
> References
>
> [a] Adversarial Examples Improve Image Recognition. Xie et al. CVPR 2020.

---

> ### Author Response · Authors · 2022-08-02
> **clarifying the DAmageNet, Adversarial Patch experiments**
>
> ### **> Regarding experiments on DamageNet and adversarial patches: the reported improvements, even though they can be statistically significant, are still low relative to the performance on clean samples.**
>
> We agree. However, we would note that we did not tune our algorithms based on the OOD datasets (all the gray rows in Table 1 of the revised manuscript are OOD results) while ResNet-50 was trained directly on ImageNet.
>
> ---
> ### **> Moreover, it is known that this kind of improvements get proven useless by generating adversarial attacks that are tailored to the classifier (kNN classifiers in this case). For instance, by using non-differentiable approximations.**
>
> **First**, we would note that our comparison follows the standard in the adversarial machine learning field, specifically, Table II in [a] where the authors show the deep classifiers completely fail to defend against adversarial attacks while kNN can to some extent.
>
> **Second**, we agree methods like BPDA (Anish, Carlini, Wagner 2018) would work on kNN. However, here, DAmageNet is a dataset of _universally robust_ adversarial examples generated against a set of classifiers, **not specifically targeting ResNet-50 or kNN**. Yet, we observe kNN to outperform ResNet-50 classifiers.
> Here, DAmageNet is effective against kNN and Corr methods since their accuracy scores are far from their own scores on ImageNet.
>
> **Third**, we believe the difference interestingly lies in the parametric vs. nonparametric contrast between ResNet-50 and (kNN, EMD-Corr, CHM-Corr).
> As ResNet-50 was trained directly on ImageNet, this parametric classifier's weights are already specialized for ImageNet, admitting less flexibility for generalizing to OOD data.
> In contrast, despite using the exact same feature extractor as in ResNet-50, kNN and Corr methods leverage a large set of fixed support images to "double-check" every unusual query image, enabling better robustness.
>
> **Fourth**, to provide a more complete answer to your comment, we also generate a new Adversarial Patch dataset (using the same setup described in Sec. 2.1) by attacking ResNet-18.
> And then, we compare the same ResNet-50, kNN, and Corr classifiers on this new Adversarial Patch dataset.
>
> Our results show the same trend as in the paper: kNN outperforms ResNet-50 and Corr classifiers outperforms kNN.
>
> |      Datasets     | # of Images | ResNet-18 | ResNet-50 |   kNN  | EMD-Corr | CHM-Corr |
> |:-----------------:|:-------:|:---------:|:---------:|:------:|:--------:|:--------:|
> | Adversarial Patch ResNet-18 |   5,000  |   46.18   |   59.12  | 60.04 |   60.80  |  **61.42**  |
>
> References
>
> [a] Deep k-Nearest Neighbors: Towards Confident, Interpretable and Robust Deep Learning. Papernot and McDaniel. 2018.

---

### Official Review · Reviewer_dcDH · 2022-07-11

**Rating:** 7
**Confidence:** 3
**Soundness:** 3 good
**Presentation:** 3 good
**Contribution:** 3 good

**Summary:**

This work proposes a novel framework and two interpretable classifiers based on retrieval and re-ranking strategy. In this framework, a visual correspondence between the query image and exemplars can be computed as explanations. This strategy is shown to be more robust and achieve comparable performance to black-box classifiers, and improve the performance when working with humans.



**Questions:**

"our approach is inherently interpretable". The features extracted by deep models are still black-box. Consider rephrasing this sentence.

**Limitations:**

addressed.

**Strengths And Weaknesses:**

- Comprehensive experiments.
- Interesting ideas

---

> ### Author Response · Authors · 2022-08-02
> **thank you for your comments! we've substantially improved the paper writing and presentation**
>
> Thank you for your positive feedback and constructive comments that strengthen our paper!
> On OpenReview, our paper has been revised in light of your valuable comments. We substantially improved the clarity of the writing and presentation of the results.
>
> Please find our replies and clarification below.
>
>
> ### **> "our approach is inherently interpretable". The features extracted by deep models are still black-box. Consider rephrasing this sentence.**
>
> Thank you for your suggestion on improving the clarity!
> We took this "inherently interpretable" jargon from many landmark papers, e.g., [a]. Yet, we agree this term can be confusing since the deep features are still black-box. "Inherently interpretable" here means models that first output an explanation, and then, use it to make a prediction (as opposed to post-hoc explanations, which aim to reverse-engineer a thought process and explain a decision after it is made).
>
> We now add "first-explain-then-decide" in L27 to clarify the meaning of "interpretable".
>
> References
>
> [a] Stop explaining black box machine learning models for high stakes decisions and use interpretable models instead. Cynthia Rudin. Natural Machine Learning. 2019.
>
> ---
> ### **> Manuscript improvements**
>
> Thanks to the reviewers' feedback, we have substantially improved the manuscript with the below major changes:
>
> - For reproducibility, we've released [an anonymized github repo](https://github.com/visualcorr/anonymous-repo) with detailed instructions for how to run our classifiers and to download our generated Adversarial Patch dataset. In the repo, we also provide a [walkthrough video](https://drive.google.com/file/d/14yBI5TIaD6s-6YkWwa4yEzSBGmI-nw8d/view) showcasing our Gorilla setup for the community's reproduction of our human studies.
>
> - We added diagrams and clarified description (Fig. 4 and Sec. 2.3) of our two human-AI interaction models assessed in this paper.
>
> - We added a runtime analysis of all four main classifiers (Appendix C). We note that using FAISS library, running kNN is fairly tractable. That is, on CUB, kNN takes 9.7s for 1,000 queries on one Nvidia V100 while that takes ResNet-50 8.81s. These numbers are 17.35s and 9.17s on ImageNet.
>
> - We revised the entire paper, fixing many typos, improving the clarity of the writing, and and re-organizing the results into a single storyline: We first propose building and testing kNN and visual correspondence-based XAI classifiers on i.i.d. and OOD data (Secs. 3.1 and 3.2) and then we test them on assisting users in two human-AI interaction models (Secs. 3.3, 3.4, and Appendix M). Our work extensively assessed the classifiers on **three binary-decision-maker settings: AI-only, human-only, and human-AI teams.**

---

### Official Review · Reviewer_Wgym · 2022-07-11

**Rating:** 6
**Confidence:** 3
**Soundness:** 3 good
**Presentation:** 2 fair
**Contribution:** 3 good

**Summary:**

This work proposed a method to explain classifier prediction by feature correspondences. Two main methods are presented -- 1. patch correspondence by earth moving distance (EMD) in cross-correlation maps from image features, 2. patch correspondence by deep hough voting. Both methods uses the matching score from the top matched patches to re-rank how "explainable" the training data is for the query data. The method can be used in two ways -- 1. the re-ranking mechanism can be applied as a kNN algorithm for robust classification, 2. the top matched patch pair can be used to visualize the network prediction, which is claimed to help improve human-AI team accuracy. Several evaluations for different applications is done, including out-of-distribution robustness, human-alone image classification, human-AI team classification, and correcting AI's wrong decisions.

**Questions:**

In figure 4a, there are 4 stars but without explanation in the caption. Although it is explained in the appendix, it is better to also mention it in the main text.

**Limitations:**

Please address the limitations mentioned in the weakness section.

**Strengths And Weaknesses:**

[Update: I have raised my rating to a weak accept, since the authors have addressed most of my concerns.]

Strengths:
1. It is interesting to use feature correspondences to explain model predictions.

Weaknesses:
1. It's hard to understand what is the main contribution of this paper. There are several application for the proposed method. However, the method only shows consistent but marginal improvement in classification robustness.
2. In most of the case, just using the prediction from plain resnet-50 achieves best performance in human-AI team performance. It also performs best in human-alone performance in ImageNet.
3. It might be good to compare against several XAI baselines, at least for the human-alone performance. For example, grad-CAM [1].
4. It might be great to include more mathematical descriptions for the method, especially for CHM-Corr.

Summary:
In my opinion, the paper needs to provide stronger results to show the proposed method is convincing. However, I am happy to adjust my rating according to the authors' responses.

[1] Selvaraju et al. Grad-CAM: Visual Explanations From Deep Networks via Gradient-Based Localization.

---

> ### Author Response · Authors · 2022-08-02
> **our novel contributions**
>
> Thank you very much for your positive feedback and constructive comments that help improve our paper! On OpenReview, our paper has been revised in light of your valuable comments. We have substantially improved our presentation and writing of the manuscript.
>
> We believe our manuscript and replies have addressed all of your concerns. We hope you would consider accepting our work.
>
> Please find our inline replies to your comments below.
>
>
> ### **> It's hard to understand what is the main contribution of this paper. There are several application for the proposed method. However, the method only shows consistent but marginal improvement in classification robustness.**
>
>
> Our main contributions are (1) adoption of the existing established correspondence methods into XAI; (2) the first evaluation of patch-based XAI classifiers on ImageNet and OOD datasets; (3) the first human study that shows that patch-based XAI classifiers help users improve human's image-classification accuracy and the human-AI team's classification accuracy.
>
> 2. That is, despite the popularity of deep kNN [a], and recent EMD-based retrieval methods [b][c], it is still **unknown** how they perform on the standard, large-scale **ImageNet and out-of-distribution datasets at the ImageNet scale** (e.g., ImageNet-R, ImageNet-Sketch, ...etc).
> We are the first to perform this test and find that interestingly kNN and patch-based classifiers outperform standard ResNet-50 classifiers given the same ResNet-50 backbones.
>
> 3. In the explainable AI research, many patch-based XAI methods have been proposed (see Table 3 in [d] for a list of methods) but **none of them have been tested** on humans. We are among the first to evaluate patch-based XAI classifiers on humans (there is a concurrent work recently accepted into ECCV 2022 [e]). Compared to all prior work and concurrent work [e], our paper is the **first to report the effectiveness of patch-based XAI classifiers in improving human image-classification accuracy**.
> In contrast, all prior works did not test on humans [d] or tested but did not find patch-based XAI classifiers to be useful to humans [e].
>
> References
>
> [a] Deep k-Nearest Neighbors: Towards Confident, Interpretable and Robust Deep Learning. Papernot and McDaniel 2018.
>
> [b] Towards Interpretable Deep Metric Learning with Structural Matching. Zhao et al. ICCV 2021
>
> [c] DeepEMD: Few-Shot Image Classification with Differentiable Earth Mover's Distance and Structured Classifiers. Zhang et al. CVPR 2020
>
> [d] Deformable ProtoPNet: An Interpretable Image Classifier Using Deformable Prototypes. CVPR 2022
>
> [e] HIVE: Evaluating the Human Interpretability of Visual Explanations. ECCV 2022
>
> [f] The effectiveness of feature attribution methods and its correlation with automatic evaluation scores. Nguyen et al, NeurIPS 2021
>
> [g] Grad-CAM: Visual Explanations From Deep Networks via Gradient-Based Localization. ICCV 2017
>
> [h] Understanding deep networks via extremal perturbations and smooth masks. Fong et al. ICCV 2019
>
> ---
> ### **> Manuscript improvements**
>
> Thanks to the reviewers' feedback, we have substantially improved the manuscript with the below major changes:
>
> - For reproducibility, we've released [an anonymized github repo](https://github.com/visualcorr/anonymous-repo) with detailed instructions for how to run our classifiers and to download our generated Adversarial Patch dataset. In the repo, we also provide a [walkthrough video](https://drive.google.com/file/d/14yBI5TIaD6s-6YkWwa4yEzSBGmI-nw8d/view) showcasing our Gorilla setup for the community's reproduction of our human studies.
>
> - We added diagrams and clarified description (Fig. 4 and Sec. 2.3) of our two human-AI interaction models assessed in this paper.
>
> - We added a runtime analysis of all four main classifiers (Appendix C). We note that using FAISS library, running kNN is fairly tractable. That is, on CUB, kNN takes 9.7s for 1,000 queries on one Nvidia V100 while that takes ResNet-50 8.81s. These numbers are 17.35s and 9.17s on ImageNet.
>
> - We revised the entire paper, fixing many typos, improving the clarity of the writing, and and re-organizing the results into a single storyline: We first propose building and testing kNN and visual correspondence-based XAI classifiers on i.i.d. and OOD data (Secs. 3.1 and 3.2) and then we test them on assisting users in two human-AI interaction models (Secs. 3.3, 3.4, and Appendix M). Our work extensively assessed the classifiers on **three binary-decision-maker settings: AI-only, human-only, and human-AI teams.**

---

> > ### Comment · Reviewer_Wgym · 2022-08-07
> > **Thanks for the clarification**
> >
> > Thanks for the clarification! The authors' response cleared a lot of my doubt. Before I change my rating to a weak accept, I have one more confusion. It seems like for both ImageNet and CUB, AI-Human team performance performs strongly on resnet. I guess the way to view this is even if plain resnet50 has a strong performance, having visual explanations is still a desired property?

---

> > > ### Author Response · Authors · 2022-08-07
> > > **thank you for a great question! (short answer: Yes)**
> > >
> > > That's an interesting, spot-on question!
> > >
> > > Our short answer is **Yes**. In light of your suggestion/question, we will add a corresponding note to the paper.
> > > Please see our long answer below.
> > >
> > > ### Long Answer:
> > > In terms of _accuracy_, we agree that ResNet-50 performs strongly in the Human-AI interaction model 2 (where both Humans and AIs make decisions on complementary subsets of inputs) compared to XAI models.
> > > The finding is aligned with slide 6 [b] by DARPA XAI program presentation [a]. That is, our explainable classifiers obtain a comparable accuracy to no-explanation ResNet-50 but _more explainability_.
> > >
> > > Having visual explanations can be a desired property for several objectives:
> > >  1. In high-stake applications (e.g., in military, healthcare or finance), explanations are **required** to inform the end-user (e.g. a doctor or a portfolio manager) for them to **make more accurate decisions** [a] (GDPR regulation is one that enforces explanations). This is the objective we study in this paper.
> > >
> > >  2. Explanations can be used for **debugging** to improve the classifier itself [e]. This is an interesting possibility of CHM-Corr and EMD-Corr that we are interested in studying in future work. For example, our nonparametric classifier inherently allows one to find errors in the support images/training images---our kNN classifier found numerous identical images in multiple ImageNet categories see Fig. A45 (page 59) in our manuscript. It is easy for the users of our Corr classifiers to replace errorneous training-set images with better images that yield higher accuracy. In another debugging scenario, the Corr classifiers' correspondence patches are highlighting what model is looking at, which, like heatmaps, can be leveraged to detect spurious correlations in the query image (e.g. classifiers using watermark in photos to classify horses instead of the actual animal's features) [e][f].
> > >
> > >  3. Explanations can be used to **teach humans**, e.g., how to classify birds in CUB [d] or butterflies [c]. This is possible to achieve with our CHM-Corr and EMD-Corr but we leave this test for future work. For example, AFTER teaching humans how XAI models classify birds using visual explanations, one can run a human study to assess how users' knowledge or ability to classify birds _using the image alone_ (no explanations now) has improved [c][d].
> > >
> > >  4. In the CHI community, there are papers that study the **social aspects**, e.g., "trust of users" associated with explanations. Yet, we'd need further research to test whether our visual explanations improve any aspect [g].
> > >
> > >
> > > In sum, in this paper, we only evaluate explanations' quality by how much they improve users' accuracy, which is an objective measure. We find that when users are to make decisions all by themselves on CUB, visual Corr explanations are their best assistance (better than ResNet-50 and kNN) see Table 2. However, when humans make decisions on only some of the inputs (human-AI team; Table 3), on both ImageNet and CUB, ResNet-50 is a strong no-explanation contender.
> > >
> > > We are excited by our results and believe our methods serve as a baseline for future improvements. Intuitively, our Corr explanations might also have other values (point 2, 3, and 4) besides improving user accuracy.
> > >
> > > ---
> > > References
> > >
> > > [a] https://www.law.berkeley.edu/wp-content/uploads/2018/03/XAI-for-UCB-Forum.pptx
> > >
> > > [b] Slide 6 from DARPA XAI slide deck: https://i.imgur.com/tViOVV9.jpeg
> > >
> > > [c] Teaching categories to human learners with visual explanations. CVPR 2018.
> > >
> > > [d] Counterfactual Visual Explanations. ICML 2019.
> > >
> > > [e] Debugging Tests for Model Explanations. NeurIPS 2020
> > >
> > > [f] Analyzing Classifiers: Fisher Vectors and Deep Neural Networks. CVPR 2016
> > >
> > > [g] Expanding explainability: Towards social transparency in ai systems. CHI 2021

---

> ### Author Response · Authors · 2022-08-02
> **2/2 we replied to your questions**
>
> ### **> In most of the case, just using the prediction from plain ResNet-50 achieves best performance in human-AI team performance. It also performs best in human-alone performance in ImageNet.**
>
> In our study, we perform the human study on both ImageNet and the CUB dataset. ImageNet dataset contains everyday objects that most people are familiar with and have substantial prior knowledge about them. Given a strong prior knowledge, users can make a lot of decisions using the input image and the AI decision alone (i.e. without the need for any further side information).
>
> On the other hand, the CUB dataset contains images of birds, which most lay users have no expertise in identification. In this case, the correspondence-based explanations show users examples that help them reject the AI's wrong decisions (Appendix F), leading to higher accuracy (Table 2).
>
> The task in our user study is designed to assess the effectiveness of XAI methods in achieving human-AI team performance for each classifier separately. **For the first time**, we show that it is possible to achieve a complementary human-AI team accuracy, i.e., better than both AI-only and Human-only (Table 3).
> We agree that ResNet-50 achieves the highest performance in this setup. Yet, our work serves as an important, first baseline for future research in this human-AI interaction model 2 (see Fig. 4 in our revised paper). We note that negative results in the XAI literature are quite common [a][b][c] and provide valuable insights for future work.
>
> References
>
> [a] The effectiveness of feature attribution methods and its correlation with automatic evaluation scores. Nguyen et al, NeurIPS 2021
>
> [b] HIVE: Evaluating the Human Interpretability of Visual Explanations. ECCV 2022.
>
> [c] Does the whole exceed its parts? the effect of ai explanations on complementary team performance. CHI 2021
>
>
> ---
> ### **> It might be good to compare against several XAI baselines. For example, grad-CAM [1].**
>
> We agree that it is important to compare against XAI baselines!
>
> On CUB, we compare CHM-Corr and EMD-Corr with a baseline **CHM-Corr+** that uses five user-defined important patches for bird identification (see Appendix N).
> Interestingly, this comparison against CHM-Corr+ baseline finds that the top-5 patches selected by our EMD-Corr and CHM-Corr are indeed more important than the expert-chosen five bird parts.
>
> We chose not to study attribution maps in this paper since prior works [a] have shown that these heatmaps are uninformative to users in image classification.
> That is, GradCAM and EP heatmaps always highlight the body or the face of a dog regardless of whether the top-1 predicted label of the classifier is correct or not [a]. This result is later confirmed by [b].
> Furthermore, comparing our Corr classifiers with heatmaps would be less scientifically informative than comparing with the more direct baseline of kNN (Sec. 3.2), which is also well-known XAI method in the literature [a][c][d][e].
>
> References
>
> [a] The effectiveness of feature attribution methods and its correlation with automatic evaluation scores. Nguyen et al, NeurIPS 2021
>
> [b] HIVE: Evaluating the Human Interpretability of Visual Explanations. ECCV 2022.
>
> [c] Interpreting Neural Networks with Nearest Neighbors
>
> [d] Advancing Nearest Neighbor Explanation-By-Example With Critical Classification Regions
>
> [e] How Can I Explain This to You? An Empirical Study of Deep Neural Network Explanation Methods
>
> ---
> ### **> It might be great to include more mathematical descriptions for the method, especially for CHM-Corr.**
>
> We agree!
> Per your request, we've updated the paper include more in-depth descriptions of all classifiers: kNN, EMD-Corr, and CHM-Corr. We added mathematical descriptions of the Earth Mover's Distance, optimization objective, and how we use it in EMD-Corr, in the Sec. A3. We also gave a detailed description of different stages of CHMNet for obtaining correspondence keypoints between two images in the Appendix A.4 of the paper.
>
> ---
>
> ### **> In figure 4a, there are 4 stars but without explanation in the caption. **
>
> **** denotes *p values* below 0.0001 which is considered as super significant. We updated the figure to include the explanation (Figure. A46).

---

### Official Review · Reviewer_6EjV · 2022-07-11

**Rating:** 6
**Confidence:** 3
**Soundness:** 3 good
**Presentation:** 3 good
**Contribution:** 3 good

**Summary:**

In this paper, the authors designed two "nearest neighbor"-based classifiers based on visual correspondences, namely, EMD-Corr and CHM-Corr. The EMD-Corr is similar to Zhao et al. (2021), and solves an optimal transport problem to match image patches from the query image to those from a training image. The difference from Zhao et al. (2021) is that the authors only used the top 5 patch matchings for classifications and explanations. The CHM-Corr is based on the Convolutional Hough Matching (CHM) network from Min and Cho (2021) -- it first solves visual correspondences between the query image and a training image using CHM (thereby removing the need for solving the optimal transport problem), and use the visual correspondences obtained from CHM to find top 5 patch matchings for classifications and explanations. In addition to designing the classifiers, the authors did an extensive user study to test the usefulness and effectiveness of visual correspondence-based explanations for improving human accuracy and human-AI team accuracy on solving challenging classification problems.

**Questions:**

The paper is very clearly written, and I do not have additional questions that need authors to clarify.

**Limitations:**

The authors have adequately addressed the limitations of their work associated with collecting data from human subjects online.

**Strengths And Weaknesses:**

Strengths:

+ The user study is comprehensive and great, and demonstrates the utility of visual correspondence-based explanations.

+ The proposed methods (EMD-Corr and CHM-Corr) are simple to implement, and can achieve reasonably high accuracy on complex datasets such as ImageNet.

Weaknesses:

- The proposed methods (EMD-Corr and CHM-Corr) are very incremental in design from prior works (Zhao et al., 2021 and Min and Cho, 2021).

---

> ### Author Response · Authors · 2022-08-02
> **thank you! we clarified our novel contributions**
>
> Thank you for your positive feedback and constructive comments that strengthen our paper!
> On OpenReview, our paper has been revised in light of your valuable comments. We substantially improved the clarity of the writing and presentation of the results.
>
> Please find our replies and clarification below.
>
>
> ### **> The proposed methods (EMD-Corr and CHM-Corr) are  incremental in design from prior works (Zhao et al., 2021 and Min and Cho, 2021).**
>
> We agree that we take the existing correspondence methods in [b] and Min and Cho 2021 but use them in _novel_ XAI applications where these methods were _not_ used before.
>
> Our main contributions are (1) adoption of these established correspondence methods into XAI; (2) the first evaluation of patch-based XAI classifiers on ImageNet and OOD datasets; (3) the first human-study that shows that patch-based XAI classifiers help users improve human's image-classification accuracy and the human-AI team's classification accuracy.
>
>  - (2) That is, despite the popularity of deep kNN [a], and recent EMD-based retrieval methods [b][c], it is still **unknown** how they perform on the standard, large-scale **ImageNet and out-of-distribution datasets at the ImageNet scale** (e.g., ImageNet-R, ImageNet-Sketch, ...etc).
> We are the first to perform this test and find that, interestingly, kNN and our patch-based classifiers outperform standard ResNet-50 classifiers given the same ResNet-50 backbones.
>
> - (3) In explainable AI, many patch-based XAI methods have been proposed (see Table 3 in [d] for a list of methods) but **none of them have been tested** on humans. We are among the first to evaluate patch-based XAI classifiers on humans (there is a concurrent work recently accepted to ECCV 2022 [e]). Compared to all prior work and concurrent work [e], our paper is the **first to report the effectiveness of patch-based XAI classifiers in improving human image-classification accuracy**.
> In contrast, all prior works did not test on humans [d] or tested but did not find patch-based XAI classifiers to be useful to humans [e].
>
>
> References
>
> [a] Deep k-Nearest Neighbors: Towards Confident, Interpretable and Robust Deep Learning. Papernot and McDaniel 2018.
>
> [b] Towards Interpretable Deep Metric Learning with Structural Matching. Zhao et al. ICCV 2021
>
> [c] DeepEMD: Few-Shot Image Classification with Differentiable Earth Mover's Distance and Structured Classifiers. Zhang et al. CVPR 2020
>
> [d] Deformable ProtoPNet: An Interpretable Image Classifier Using Deformable Prototypes. Donnelly et al. CVPR 2022
>
> [e] HIVE: Evaluating the Human Interpretability of Visual Explanations. Kim et al. ECCV 2022
>
> ---
> ### **> Manuscript improvements**
>
> Thanks to the reviewers' feedback, we have substantially improved the manuscript with the below major changes:
>
> - For reproducibility, we've released [an anonymized github repo](https://github.com/visualcorr/anonymous-repo) with detailed instructions for how to run our classifiers and to download our generated Adversarial Patch dataset. In the repo, we also provide a [walkthrough video](https://drive.google.com/file/d/14yBI5TIaD6s-6YkWwa4yEzSBGmI-nw8d/view) showcasing our Gorilla setup for the community's reproduction of our human studies.
>
> - We added diagrams and clarified description (Fig. 4 and Sec. 2.3) of our two human-AI interaction models assessed in this paper.
>
> - We added a runtime analysis of all four main classifiers (Appendix C). We note that using FAISS library, running kNN is fairly tractable. That is, on CUB, kNN takes 9.7s for 1,000 queries on one Nvidia V100 while that takes ResNet-50 8.81s. These numbers are 17.35s and 9.17s on ImageNet.
>
> - We revised the entire paper, fixing many typos, improving the clarity of the writing, and and re-organizing the results into a single storyline: We first propose building and testing kNN and visual correspondence-based XAI classifiers on i.i.d. and OOD data (Secs. 3.1 and 3.2) and then we test them on assisting users in two human-AI interaction models (Secs. 3.3, 3.4, and Appendix M). Our work extensively assessed the classifiers on **three binary-decision-maker settings: AI-only, human-only, and human-AI teams.**

---

> > ### Comment · Reviewer_6EjV · 2022-08-09
> > **Thank you for the response**
> >
> > Thank you for the response! I acknowledge that the paper has provided novel contributions in terms of performing evaluations of patch-based XAI classifiers on large-scale datasets such as ImageNet and conducting human studies on these classifiers, but I am hesitant to increase my rating because the technical contributions (in terms of developing new techniques for XAI) are still incremental.

---

### Meta-Review · Area_Chair_U7dX · 2022-08-28

**Recommendation:** Accept
**Confidence:** Less certain

**Metareview:**

This paper proposes a classifier based on different refining strategies of the closest nearest neighbors using intermediate features, providing explanations in the form of example images. The authors demonstrate that it is possible to achieve complementary human-AI team accuracy in image classification. In general, the paper is clearly written by addressing an interesting problem. The reviewers point out the limitations in terms of the novelty, experiments design, and clarity. Afte the rebuttal and extensive discussion, most of the reviewers agree that the concerns are properly addressed although there still exist some. In general, the paper is interesting by providing some insights, but the authors are expected to make a through revision by considering the reviewers' commets.

**Award:**

No

---

### Decision · Program_Chairs · 2022-09-14

Accept